# REVISITING OUT-OF-DISTRIBUTION DETECTION: ANGULAR SEPARATION LEARNING AS A POWERFUL AND SIMPLE BASELINE

## ABSTRACT

Out-of-Distribution (OOD) detection is a critical safety requirement for deploying deep neural networks in open-world environments. While recent advances increasingly rely on more computationally intensive training methods involving synthetic outliers, contrastive objectives, or specialized loss functions, their gains often come with substantial computational overhead and implementation complexity. In this work, we revisit the fundamentals of OOD detection and uncover a key flaw in common distance-based detectors: sensitivity to feature magnitude. We show that low-norm OOD samples can appear closer to in-distribution (ID) class centroids than actual ID samples, evading detection. To address this, we introduce **Angular Separation Learning (ASL)**, a simple and highly effective strategy that applies $\ell_2$-normalization to features before the final classification layer. This modification compels the network to optimize for angular separation, achieving robust feature learning without additional regularization mechanisms, synthetic samples, or costly negative mining. Through extensive experiments on diverse benchmarks, we demonstrate that ASL not only matches but often surpasses state-of-the-art methods, especially in challenging near-OOD scenarios, while maintaining training efficiency. Our results indicate that a minimalist rethink of standard training can achieve superior OOD performance, prompting a re-evaluation of the complexity-to-performance trade-off in OOD detection. Code is available at `https://anonymous.4open.science/r/ASL-56FF`.

## 1 INTRODUCTION

The deployment of deep learning models into real-world, safety-critical applications represents a major frontier for artificial intelligence. From autonomous vehicles navigating unpredictable roadways (Huang et al., 2020) to AI-powered systems aiding in medical diagnosis (Zimmerer et al., 2022), models are increasingly tasked with making high-stakes decisions in dynamic, open-world environments (Rabe et al., 2021; Henriksson et al., 2023). However, a fundamental gap exists between the controlled conditions of training and the unbounded nature of reality. Models trained under a "closed-world" assumption often fail catastrophically when encountering inputs that deviate from their training distribution, known as out-of-distribution (OOD) samples (Amodei et al., 2016). A self-driving car may misinterpret an unfamiliar object on the road, or a medical diagnostic tool may produce a dangerously overconfident prediction for a rare disease. This failure to recognize and appropriately handle novel inputs is a primary barrier to safe and reliable deployment. Consequently, the ability to detect OOD samples is not merely a desirable property but a critical safety requirement for building trustworthy AI systems that can robustly "know when they don't know".

The core challenge of OOD detection is to distinguish between in-distribution (ID) and OOD inputs using a model trained on ID data. Initial and ongoing research has produced a rich landscape of post-hoc methods that operate on pre-trained models without altering their weights. These approaches leverage various signals from the network, from simple uncertainty scores derived from the model's output logits like MSP (Hendrycks & Gimpel, 2017) and Energy scores (Liu et al., 2020), to more sophisticated techniques that analyze the network's internal state. Researchers have successfully utilized backpropagation gradients (Huang et al., 2021) and manipulated intermediate activation patterns (Sun et al., 2021; Djurisic et al., 2023) to amplify the distinction between ID and OOD

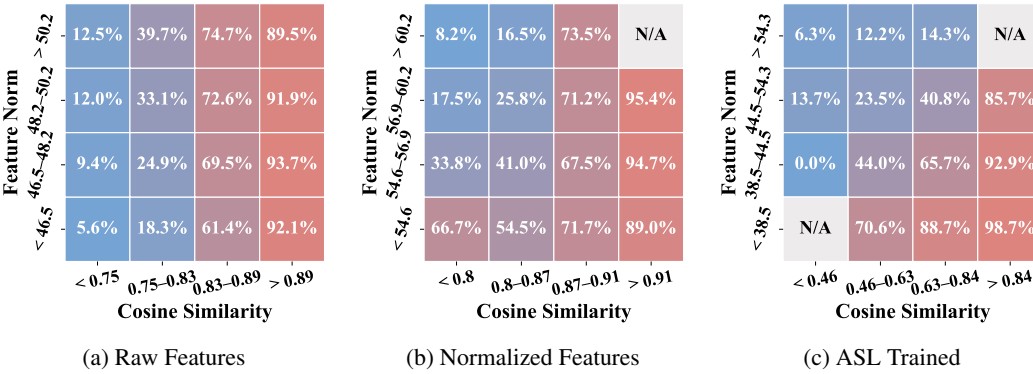

(a) Raw Features      (b) Normalized Features      (c) ASL Trained

Figure 1: Visualization of ID vs. OOD separability in the Mahalanobis space. The two-dimensional space is defined by the feature's $\ell_2$-norm and its cosine similarity to the nearest class mean. The space is partitioned into $4 \times 4$ regions using independent thresholds computed for each dimension, and each region indicates the proportion of in-distribution samples within that area. ASL enhances angular separation and concentration, resulting in more discriminative ID/OOD distributions and improved detection performance.

samples. Among this diverse array of techniques, methods that analyze the geometric properties of the feature space have proven particularly powerful. Specifically, distance-based detectors, such as the Mahalanobis Distance Score (MDS) (Lee et al., 2018), have demonstrated strong empirical performance, particularly under the closed-world assumption or with far-OOD samples, establishing a prominent direction of inquiry in the field.

While these post-hoc methods are valuable, their effectiveness is fundamentally capped by the representations they inherit. A standard cross-entropy objective trains a model to be discriminative, learning only the minimal features necessary to separate the known ID classes rather than modeling the underlying data distribution (Vaze et al., 2022). This can result in a feature space where ID classes are not well-separated and OOD samples are incorrectly projected into regions of high confidence (Ming et al., 2023). This realization has motivated a crucial shift towards training-based approaches that explicitly sculpt the feature manifold for OOD robustness. The core advantage of these methods is the ability to learn representations with superior geometric properties, such as increased intra-class compactness and inter-class separability, thereby creating a more structured and reliable feature space (Winkens et al., 2020). While effective, these methods often incur substantial computational overhead, hyperparameter sensitivity, and implementation intricacy. More importantly, Figure 2 suggests that this complexity does not always reliably translate into superior performance, especially for the challenging near-OOD case, raising a fundamental question: is such complexity always justified?

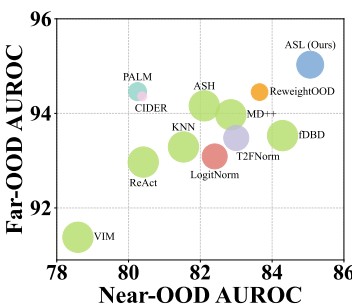

Figure 2: OOD detection performance on ImageNet-200. Larger point size represents higher training speed, and green points indicate post-hoc methods using model trained on cross-entropy.

In stark contrast to this trend, we analyze a key vulnerability in Mahalanobis distance-based detectors: their extreme sensitivity to the scale ($\ell_2$-norm) of features. The Mahalanobis distance calculation conflates a feature's magnitude with its directional information. Our analysis reveals that because ID and OOD samples often exhibit distinct norm distributions, the scoring mechanism is disrupted. This creates a pathological failure mode: a low-norm OOD sample, despite its potentially large angular deviation, can yield a smaller Mahalanobis distance than a genuine high-norm ID sample, thereby evading detection. To address this fundamental problem, we introduce Angular Separation Learning (ASL), which forces the model to learn representations optimized for angular separation, rather than relying on feature magnitude. Surprisingly, this simple modification alone is sufficient to significantly enhance intra-class compactness, which is evidenced by the tighter clustering of ID samples in regions of high cosine similarity, and this angular concentration leaves

well-defined low-density regions between classes, as shown in Fig. 1. In summary, our contributions are as follows:

- We provide a detailed analysis of a specific flaw in Mahalanobis-based detection in Mahalanobis-based OOD detection, revealing that its root cause lies in the feature space learned by standard cross-entropy training.

- We propose ASL, a training framework that applies $\ell_2$-normalization to the feature representations before the cross-entropy loss. This simple modification optimizes the feature space for angular discriminability, creating robust representations for OOD detection.

- We conduct extensive experiments on multiple OOD benchmarks, demonstrating that our method significantly outperforms existing state-of-the-art approaches, particularly in challenging near-OOD scenarios.

## 2 RELATED WORK

**Distance-Based Post-Hoc OOD Detection.** These methods analyze the geometric properties of a pre-trained model's feature space. A prominent line of work builds upon the Mahalanobis Distance Score (MDS) (Lee et al., 2018), which models classes as conditional Gaussians. Refinements include VIM (Wang et al., 2022), which incorporates a residual vector, and RMDS (Ren et al., 2021), which mitigates the influence of non-discriminative features by subtracting a global background distance from the class-conditional one. fDBD (Liu & Qin, 2024) approximates the distance to the decision boundary. MD++ (Müller & Hein, 2025) addresses sensitivity to feature norms directly by applying normalization to features before distance computation. Non-parametric alternatives avoid distributional assumptions, such as the k-Nearest Neighbors (KNN) detector (Sun et al., 2022), and its extension NNGuide (Park et al., 2023b) that uses k-NN to guide confidence scores.

**Angular and Neural Collapse Based Methods.** Recent methods continue to explore richer geometric structures beyond Euclidean distance. A prominent line of work builds upon the phenomenon of neural collapse and angular geometry for post-hoc detection. Demirel et al. (2025) propose ORA, a scale-invariant angular score between features and decision boundaries relative to ID mean. In parallel, the phenomenon of neural collapse has inspired new detectors: NCI (Liu & Qin, 2025) combines centered angular proximity to weight vectors with feature norm filtering, whereas NECO (Ammar et al., 2024) leverages Neural Collapse, computing PCA-projected norm ratio with MaxLogit calibration. In parallel, another thread of research investigates training-based angular learning. SphereFace (Liu et al., 2017) introduces a multiplicative angular margin loss for face recognition to enforce larger inter-class separations on a hypersphere, while Techapanurak et al. (2020) propose a hyperparameter-free cosine-similarity classifier for OOD detection that learns a scaling parameter via an additional network branch.

**OOD Detection via Contrastive Learning.** Training-based methods for OOD detection often leverage contrastive learning to learn more separable feature representations by enhancing intra-class compactness and inter-class separation. These approaches can be categorized based on their source of negative samples for the contrastive objective. A major line of work uses other ID samples as negatives, often building on frameworks like SimCLR (Chen et al., 2020) and Supervised Contrastive Learning (SupCon) (Khosla et al., 2020). For instance, SSD (Sehwag et al., 2021) uses self-supervised contrastive pre-training followed by fine-tuning, while CIDER (Ming et al., 2023) explicitly optimizes mutual information to tighten class clusters. PALM (Lu et al., 2024) represents each class with a mixture of prototypes to capture complex intra-class structures. More recently, ReweightOOD (Regmi et al., 2024b) dynamically reweights negative pairs based on embedding similarity to further reduce intra-class variance. Another strategy explicitly defines the decision boundary by generating pseudo-OOD samples to serve as negative examples. These methods contrast in-distribution (positive) samples against the synthesized (negative) outliers. VOS (Du et al., 2022) synthesizes virtual outliers from low-likelihood regions of class-conditional feature distributions, while NPOS (Tao et al., 2023) adopts a non-parametric approach by perturbing boundary ID samples. HamOS (Li & Zhang, 2025) uses Hamiltonian Monte Carlo to efficiently generate diverse synthetic outliers.

**Normalization.** A growing body of work has highlighted the benefits of normalization for out-of-distribution (OOD) detection. These efforts can be broadly grouped into two main categories. The

first leverages the norm itself as a discriminative signal: for instance, Zhang & Xiang (2023) incorporate feature norms directly into their scoring function, while Park et al. (2023a) identify the feature norm as a class-agnostic confidence measure. However, under standard cross-entropy training, we observe that this separability is often compromised. Models tend to exploit optimization shortcuts by inflating feature magnitudes, leading to significant overlap between ID and OOD norm distributions (Fig. 15, 16). This indicates that reliable norm separability is not an intrinsic property of unconstrained training, but rather necessitates explicit geometric constraints to be effectively realized. The second category applies normalization during training primarily as a form of regularization, early approaches like LogitNorm (Wei et al., 2022) focused on regularizing the model's output space to mitigate overconfidence. Building on this, T2FNorm (Regmi et al., 2024a) demonstrated that normalizing features during training could improve logit separation at test time. Concurrently, from a theoretical standpoint, Haas et al. (2023) established a crucial link between $\ell_2$-normalization, earlier neural collapse, and improved OOD performance. Although these studies confirm the utility of normalization, which acts either as a scoring cue or a regularizer, they largely treat it as an auxiliary mechanism. In contrast, we propose a unified perspective that elevates $\ell_2$-normalization to the core of the learning objective. We demonstrate that it performs contrastive learning in angular space, offering a streamlined and highly effective alternative to complex OOD detection approaches.

## 3 METHOD

### 3.1 PRELIMINARIES

**OOD Detection Setup.** Let $\mathcal{X}$ be the input space and $\mathcal{Y} = \{1, \ldots, K\}$ be the label space for a $K$-class classification task. We are given an in-distribution (ID) training dataset $\mathcal{D}_{in} = \{(\boldsymbol{x}_i, y_i)\}_{i=1}^{N}$ sampled from a joint distribution $P_{in}(\boldsymbol{x}, y)$. A deep neural network $f(\cdot; \theta)$ is trained on $\mathcal{D}_{in}$, which can be decomposed into an encoder $f(\cdot) : \mathcal{X} \to \mathbb{R}^d$ that maps an input $\boldsymbol{x}$ to a $d$-dimensional feature embedding, and a classifier that outputs a class prediction.

The goal of out-of-distribution (OOD) detection is to distinguish ID samples (from $P_{in}$) from OOD samples (from an unknown $P_{out}(\boldsymbol{x})$ with disjoint label space). An ideal scoring function should yield higher scores for ID inputs than for OOD inputs, i.e., $S(\boldsymbol{x}_{in}) > S(\boldsymbol{x}_{out})$ for $\boldsymbol{x}_{in} \sim P_{in}$ and $\boldsymbol{x}_{out} \sim P_{out}$.

Table 1: Comparison of OOD detection performance (FPR95↓ and AUROC↑). FeatCos refers to using only the cosine similarity between features and class means as the detection score. FeatCos[†] uses features from an ASL trained model.

| ID Dataset | Method | Near-OOD | | Far-OOD | |
|---|---|---|---|---|---|
| | | FP↓ | AU↑ | FP↓ | AU↑ |
| CIFAR-10 | MSP | 45.8 | 88.3 | 30.0 | 91.4 |
| | MDS | 49.2 | 85.2 | 33.3 | 89.0 |
| | MD++ | 40.2 | 88.8 | 30.6 | 91.0 |
| | FeatCos | 38.9 | 89.4 | 29.6 | 91.3 |
| | FeatCos[†] | 30.5 | 92.0 | 16.0 | 96.4 |
| CIFAR-100 | MSP | 55.9 | 80.0 | 58.5 | 78.1 |
| | MDS | 83.0 | 59.4 | 73.5 | 68.3 |
| | MD++ | 67.8 | 74.8 | 51.9 | 82.6 |
| | FeatCos | 59.0 | 80.3 | 54.6 | 81.8 |
| | FeatCos[†] | 57.1 | 80.3 | 50.4 | 82.7 |

A widely used baseline is the **Mahalanobis Distance Score (MDS)** (Lee et al., 2018). It models the feature distribution of each class $c$ as a Gaussian $\mathcal{N}(\boldsymbol{\mu}_c, \boldsymbol{\Sigma})$, where $\boldsymbol{\mu}_c$ is the class mean and $\boldsymbol{\Sigma}$ is a shared covariance matrix, both estimated from $\mathcal{D}_{in}$. The OOD score for a test sample $\boldsymbol{x}$ and its feature $f(\boldsymbol{x})$ is defined as:

$$S_{MDS}(\boldsymbol{x}) = -\min_{c \in \mathcal{Y}} \left( (f(\boldsymbol{x}) - \boldsymbol{\mu}_c)^T \boldsymbol{\Sigma}^{-1} (f(\boldsymbol{x}) - \boldsymbol{\mu}_c) \right),$$

this score assumes that OOD samples lie farther from all class centroids than ID samples.

### 3.2 MOTIVATING OBSERVATIONS

**The Mechanics of MDS Failure.** As shown in Table 1, MDS and its variant MD++ (Müller & Hein, 2025) are often outperformed by the simple cosine similarity score.

To understand why MDS fails to deliver satisfactory performance, we can decompose the Mahalanobis distance calculation. For a given feature vector $f(x)$, the distance to a class-c prototype $\hat{\mu}_c$ is:

$$dist_c(\boldsymbol{x}) = \underbrace{f(\boldsymbol{x})^{\top} \hat{\Sigma}^{-1} f(\boldsymbol{x})}_{\text{Sample Quadratic Term (SQT)}} - 2 \underbrace{f(\boldsymbol{x})^{\top} \hat{\Sigma}^{-1} \hat{\mu}_c}_{\text{Cross-Sample Term (CST)}} + \underbrace{\hat{\mu}_c^{\top} \hat{\Sigma}^{-1} \hat{\mu}_c}_{\text{Class Bias Term (CBT)}}. \tag{1}$$

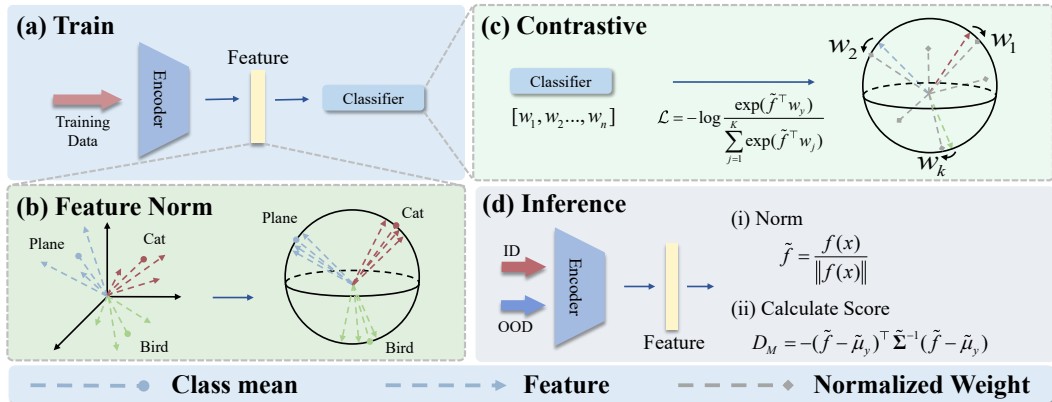

Figure 3: Pipeline of the proposed ASL framework. During training, features are $\ell_2$-normalized before cross-entropy loss. This encourages angular separation between classes without explicit contrastive learning.

Theoretically, minimizing the cross-entropy loss should align with minimizing equation 1 by simultaneously reducing the feature norm and increasing the alignment with the class mean. However, in practice, we observe a significant optimization bias during standard training. The model tends to exploit a 'lazy' shortcut by inflating feature norms to rapidly minimize the loss, rather than investing in the more difficult task of enforcing strict intra-class compactness. This unbalanced training dynamic results in a feature space where large variance in magnitude acts as a confounding factor, making the final MDS score highly sensitive to feature norms rather than reflecting true semantic distance.

To formalize this analysis, we can simplify the expression by performing a whitening transformation. Let $g(x) = \hat{\Sigma}^{-1/2} f(x)$ and $\hat{u}_c = \hat{\Sigma}^{-1/2} \hat{\mu}_c$ be the feature and class mean in the whitened space (also known as the Mahalanobis space), respectively. Ignoring the constant term CBT and letting $\theta(x, y)$ represent the angle between two vectors, then for category $c$ the Mahalanobis distance score can be written as:

$$S_c(\boldsymbol{x}) \approx -\|g(\boldsymbol{x})\|_2^2 + 2\|g(\boldsymbol{x})\|_2\|\hat{u}_c\|_2 \cos(\theta(g(\boldsymbol{x}), \hat{u}_c)). \tag{2}$$

Empirically, the norm of a typical ID feature, $\|g(\boldsymbol{x})\|_{ID}$, is close to the norm of its corresponding class mean, $\|\hat{u}_c\|_2$. Since $\cos(\theta) \leq 1$, the score is maximized at a norm value that is less than or equal to the typical ID feature norm. This implies that for any feature $g(x)$ whose norm is greater than the optimal value (i.e., $\|g(x)\|_2 > \|\hat{u}_c\|_2 \cos\theta$), increasing its norm will decrease the score, correctly identifying it as an OOD sample. However, it also exposes a critical failure mode: an OOD sample with an anomalously small norm can achieve a deceptively high score.

We demonstrate this pathology with an empirical example (see Figure 1a), where the whitened class mean norm is $\|\hat{u}_c\|_2 = 48$: 1) A **typical ID sample** with high similarity ($\cos\theta = 0.825$) and representative norm ($\|g(x)\|_2 = 49$) attains a score of **1480**. 2) A **near OOD sample** with equally high similarity ($\cos\theta = 0.82$) but smaller norm ($\|g(x)\|_2 = 47$) achieves a higher score of **1491**. 3) A more **dissimilar OOD sample** ($\cos\theta = 0.81$) with an even smaller norm ($\|g(x)\|_2 = 40$) receives the highest score (**1510**). Notably, despite lower alignment with the class prototype, its significantly reduced norm yields a falsely confident score.

Work such as MD++ has observed that feature normalization can mitigate the interference of low-norm OOD samples when using MDS. However, as shown in Figure 1b, while normalization increases the proportion of ID samples in lower-norm regions to some extent, it fails to improve angular distribution. As a result, the normalized model still exhibits near-OOD performance comparable to, or even worse than, the simple cosine similarity score.

This analysis uncovers a counter-intuitive reality: MDS can be easily fooled by OOD samples with small feature norms. This confounding effect allows samples with lower semantic similarity to appear more in-distribution than genuine ID samples. This motivates our central research question: *Can we go further to learn a feature geometry that is intrinsically invariant to norm variations and emphasizes angular separation for optimal OOD performance?*

### 3.3 ASL: Training for Optimal Angular Separation

Inspired by the observations in the previous section, particularly the experimental results in Table 1, we reveal a critical issue and a promising direction for current OOD detection methods. On one hand, the performance of feature norm-sensitive MDS is significantly disrupted by variations in feature norms, leading to unstable results. On the other hand, a striking finding is that using a purely angular metric as the scoring criterion achieves highly competitive performance, especially in challenging near-OOD scenarios.

These observations directly inspire our core idea: given the criticality of angular information for distinguishing OOD samples, rather than passively dealing with the interference caused by norm variations during OOD detection, we should proactively enforce the model during training to learn a feature space optimized purely for angular separation, thereby generating feature representations that are insensitive to norm variations and inherently robust.

To achieve this goal, we propose **Angular Separation Learning (ASL)**. ASL is built upon a key insight: when the standard cross-entropy loss is applied to $\ell_2$-normalized features, it performs contrastive learning. This simple operation forces the model to focus exclusively on optimizing the angular separation between classes. We will show that this method is equivalent to a form of contrastive learning while avoiding the computational overhead of explicit pair mining or complex loss functions, and this simplicity is also key to the high training efficiency of ASL. Specifically, it only adds a normalization step after feature extraction, along with a further simplification of removing the bias term from the linear layer. This results in a computational cost that is nearly on par with standard cross-entropy training, while achieving lower training overhead compared to other methods.

Let the feature extractor be $f(\cdot)$ and the final linear classifier have weights $\{w_j\}_{j=1}^K$, which we treat as learnable prototypes for each class. We first $\ell_2$-normalize the feature vector: $\tilde{f} = s \cdot f(x)/||f(x)||_2$. Here, $s$ is a scaling factor introduced to counteract the reduced gradient magnitudes resulting from $\ell_2$-normalization, and it is typically set to 10. The logit for class $j$ then becomes $w_j^\top \tilde{f} = ||w_j||_2 \cos(\theta_j)$, where $\theta_j$ is the angle between the feature vector $\tilde{f}$ and the class prototype $w_j$. To prevent the model from simply increasing the weight norms $||w_j||_2$ to minimize the loss, we employ standard weight decay. The final training objective for ASL is:

$$\mathcal{L}_{\text{ASL}} = \mathcal{L}_{\text{CE}}(\mathbf{W}^\top \tilde{f}, y) + \lambda ||\mathbf{W}||_F^2,$$

where $\tilde{f} = s \cdot f(x)/||f(x)||_2$ is the normalized feature vector, $y$ is the class label of sample $x$, $\mathbf{W}$ is the weight matrix of the final classifier, $\lambda$ is the weight decay hyperparameter, and $||\cdot||_F$ denotes the Frobenius norm. This simple yet effective objective compels the model to learn a feature space where classes are distinguished purely by their angular position on the hypersphere.

The following proposition formalizes the properties of this approach.

**Proposition 1** (Angular Separation Learning). *Let $\tilde{f} = s \cdot f(x)/\|f(x)\|_2$ be $\ell_2$-normalized features and $\mathbf{W} = [w_1, \ldots, w_K] \in \mathbb{R}^{d \times K}$ a prototype matrix. Minimizing the loss*

$$\mathcal{L}_{ASL} = -\log \frac{\exp(\tilde{f}^\top w_y)}{\sum_{j=1}^K \exp(\tilde{f}^\top w_j)} + \lambda \|\mathbf{W}\|_F^2, \quad \lambda > 0, \tag{3}$$

*induces a hyperspherical feature geometry with the following properties: (i) It minimizes the angular distance $\theta_y$ between $\tilde{f}$ and $w_y$ while maximizing the angular distances $\theta_j$ ($j \neq y$) between $\tilde{f}$ and all other prototypes $w_j$; (ii) At convergence, $|w_j|_2 \leq \frac{s}{2\lambda}$ for all $j$.*

As formally established in Proposition 1, the optimization process performs prototype-based contrastive learning in the angular space, simultaneously minimizing the angle to the correct class prototype while maximizing the angles to all incorrect ones, and the weight decay $\lambda$ imposes an upper bound on the norms of the classifier weights, which in turn function as an inverse temperature controlling the sharpness of the softmax distribution, while this temperature regulation is dynamic (unlike fixed-temperature in contrastive learning), allowing it to adapt from coarse-grained early training to fine-grained late-stage optimization. See Appendix E.2 for a more complete discussion.

Table 2: OOD detection performance on ImageNet-200 using a ResNet-18 model trained from scratch. CE represents model trained with cross-entropy. **Best results** are presented in bold, and second-best results are presented in underline.

| | Near-OOD | | | | | | Far-OOD | | | | | | | | |
| | SSB-hard | | NINCO | | Avg. | | iNaturalist | | Textures | | OpenImage-O | | Avg. | | Acc.↑ |
| Method | FP↓ | AU↑ | FP↓ | AU↑ | FP↓ | AU↑ | FP↓ | AU↑ | FP↓ | AU↑ | FP↓ | AU↑ | FP↓ | AU↑ | |
|---|---|---|---|---|---|---|---|---|---|---|---|---|---|---|---|
| *CE* | 72.5 | 78.8 | 46.1 | 86.9 | 59.3 | 82.8 | 19.1 | 95.9 | 24.3 | 94.6 | 29.8 | 91.4 | 24.4 | 94.0 | 86.3 |
| CIDER | 71.0 | 75.5 | 45.3 | 85.3 | 58.1 | 80.4 | 20.0 | 94.4 | 16.8 | 97.1 | 27.7 | 91.6 | 21.5 | 94.4 | 84.2 |
| LogitNorm | 66.7 | 78.4 | 46.6 | 86.4 | 56.7 | 82.4 | 15.7 | 96.1 | 31.7 | 92.3 | 31.0 | 90.9 | 26.1 | 93.1 | 86.4 |
| PALM | 70.1 | 75.7 | 50.3 | 84.8 | 60.2 | 80.2 | 21.9 | 94.3 | **12.7** | **97.8** | 29.3 | 91.3 | 21.3 | 94.5 | 83.8 |
| ReweightOOD | 67.2 | 80.5 | 41.8 | 86.8 | 54.5 | 83.7 | 18.2 | 95.0 | 19.5 | 96.4 | 27.8 | 91.9 | 21.9 | 94.5 | 84.0 |
| T2FNorm | 66.1 | 79.2 | 46.2 | 86.8 | 56.1 | 83.0 | **13.7** | **96.8** | 33.7 | 91.8 | 28.5 | 91.8 | 25.3 | 93.5 | 86.5 |
| **ASL (Ours)** | **65.7** | **81.5** | **39.1** | **88.6** | **52.4** | **85.1** | 15.2 | 96.5 | 21.6 | 95.4 | **24.4** | **93.2** | **20.4** | **95.0** | **86.8** |

Table 3: OOD detection performance on CIFAR-10 and CIFAR-100.

| | CIFAR-10 | | | | | | | | CIFAR-100 | | | | | | |
| | Near-OOD | | Far-OOD | | Avg. | | | | Near-OOD | | Far-OOD | | Avg. | | |
| Method | FP↓ | AU↑ | FP↓ | AU↑ | FP↓ | AU↑ | Acc.↑ | Method | FP↓ | AU↑ | FP↓ | AU↑ | FP↓ | AU↑ | Acc.↑ |
|---|---|---|---|---|---|---|---|---|---|---|---|---|---|---|---|
| *CE* | 40.2 | 88.8 | 30.6 | 91.0 | 35.4 | 89.9 | **95.2** | *CE* | 67.8 | 74.8 | 51.9 | 82.6 | 59.8 | 78.7 | **77.4** |
| NPOS | 32.3 | 90.3 | 21.1 | 94.6 | 26.7 | 92.4 | 92.1 | NPOS | 67.1 | 77.7 | 50.2 | 83.6 | 58.6 | 80.7 | 72.9 |
| CIDER | 31.4 | 90.1 | 22.5 | 93.4 | 26.9 | 91.7 | 92.8 | CIDER | 68.4 | 74.4 | 61.5 | 76.9 | 64.9 | 75.7 | 68.1 |
| LogitNorm | 28.9 | 92.5 | 15.8 | 96.3 | 22.4 | 94.4 | 94.5 | LogitNorm | 62.4 | 78.5 | 50.4 | 82.1 | 56.4 | 80.3 | 76.1 |
| PALM | 30.4 | 90.7 | 18.5 | 95.6 | 24.5 | 93.1 | 93.5 | PALM | 64.7 | 77.2 | **37.8** | **86.7** | **51.2** | 82.0 | 75.7 |
| ReweightOOD | 30.3 | 91.8 | **10.2** | **97.6** | 20.2 | 94.7 | 92.9 | ReweightOOD | 65.8 | 76.5 | 42.0 | **88.1** | 53.9 | 82.3 | 71.5 |
| T2FNorm | 25.8 | 93.0 | 15.2 | 96.5 | 20.5 | 94.8 | 94.7 | T2FNorm | 58.2 | 79.9 | 49.5 | 83.3 | 53.8 | 81.6 | 76.3 |
| HamOS | 28.5 | 91.5 | 16.2 | 95.6 | 22.3 | 93.5 | 93.8 | HamOS | 62.7 | 79.1 | 54.2 | 80.6 | 58.5 | 79.9 | 74.9 |
| **ASL (Ours)** | **25.1** | **93.3** | 12.0 | 97.2 | **18.6** | **95.2** | 94.8 | **ASL (Ours)** | **56.2** | **80.9** | 47.8 | 85.0 | 52.0 | **83.0** | 76.7 |

**From "good enough" to "as good as possible".** Without normalization, a model can achieve low classification loss by simply scaling up the magnitude of features, even if the angular separation is poor—a "lazy" shortcut to optimization. Feature normalization blocks this shortcut, thereby compelling the model to learn more essential and finer-grained features that enable optimal angular separation.

## 4 EXPERIMENTS

### 4.1 EXPERIMENTAL SETUP

**Datasets and Models.** Our experimental setup adheres to the standard benchmarks established in OpenOOD (Zhang et al., 2024) to ensure fair and rigorous comparisons. We conduct experiments on four in-distribution (ID) datasets: CIFAR-10, CIFAR-100 (Krizhevsky et al., 2009), ImageNet-200, and the full ImageNet-1K (Deng et al., 2009). For each ID setting, we evaluate on a comprehensive suite of near-OOD (semantically similar) and far-OOD (semantically distinct) datasets. Additionally, we evaluate model performance under covariate shift by incorporating corrupted samples (Hendrycks & Dietterich, 2019) into the ID test set. We further include experiments on the BIMCV medical imaging dataset (Vayá et al., 2020), and the detailed setup is provided in the Appendix B.1.

We employ a ResNet-18 (He et al., 2016) for experiments on CIFAR-10, CIFAR-100, ImageNet-200, and BIMCV. For ImageNet-1K, we utilize standard pre-trained ResNet-50 (He et al., 2016) and ViT-B/16 (Dosovitskiy et al., 2021) models from Torchvision (maintainers & contributors, 2016). Detailed dataset setup and training configuration are deferred to Appendix B.

**Evaluation Metrics.** We adopt two standard metrics for OOD detection: the Area Under the Receiver Operating Characteristic curve (AUROC) and the False Positive Rate at 95% True Positive Rate (FPR95). Higher AUROC and lower FPR95 values indicate better performance.

**Baselines.** We benchmark our method against a wide range of state-of-the-art approaches, which can be categorized as follows: 1) Post-hoc methods: MSP (Hendrycks & Gimpel, 2017), MLS (Hendrycks et al., 2022), MDS (Lee et al., 2018), RMDS (Ren et al., 2021), ReAct (Sun et al.,

Table 4: OOD detection performance under covariate shift on ImageNet-200 using a ResNet-18 model trained from scratch.

| | Near-OOD | | | | | | Far-OOD | | | | | | | |
| | SSB-hard | | NINCO | | Avg. | | iNaturalist | | Textures | | OpenImage-O | | Avg. | |
| Method | FP↓ | AU↑ | FP↓ | AU↑ | FP↓ | AU↑ | FP↓ | AU↑ | FP↓ | AU↑ | FP↓ | AU↑ | FP↓ | AU↑ |
|---|---|---|---|---|---|---|---|---|---|---|---|---|---|---|
| *CE* | 78.1 | 70.2 | 55.6 | 79.2 | 66.9 | 74.7 | 28.4 | 90.0 | **22.1** | **95.1** | 39.9 | 86.0 | 30.1 | 90.3 |
| CIDER | 75.2 | 68.0 | 53.0 | 78.0 | 64.1 | 73.0 | 29.1 | 88.9 | 26.0 | 94.0 | 36.4 | 85.2 | 30.5 | 89.4 |
| LogitNorm | 71.7 | 71.3 | 54.1 | 79.9 | 62.9 | 75.6 | 24.5 | 92.0 | 40.3 | 87.4 | 39.7 | 85.0 | 34.8 | 88.1 |
| PALM | 74.8 | 67.8 | 57.5 | 77.1 | 66.2 | 72.4 | 31.8 | 88.2 | 22.2 | 94.5 | 38.7 | 84.5 | 30.9 | 89.1 |
| ReweightOOD | 72.0 | 72.9 | 49.7 | 79.3 | 60.9 | 76.1 | 27.6 | 89.0 | 28.8 | 92.3 | 36.8 | 85.2 | 31.1 | 88.8 |
| T2FNorm | 71.5 | 72.8 | 54.2 | 80.1 | 62.9 | 76.5 | 23.6 | 92.4 | 43.8 | 83.8 | 39.1 | 84.7 | 35.5 | 86.9 |
| **ASL (Ours)** | **70.7** | **74.9** | **47.2** | **82.3** | **59.0** | **78.6** | 23.8 | 92.5 | 30.4 | 91.9 | **33.1** | **88.0** | **29.1** | **90.8** |

Table 5: OOD detection performance on ImageNet with fine-tuned ResNet-50 (Left), ViT-B/16 (Right).

| | ResNet-50 | | | | | | | ViT-B/16 | | | | | | |
| | Near-OOD | | Far-OOD | | Avg. | | | Near-OOD | | Far-OOD | | Avg. | | |
| Method | FP↓ | AU↑ | FP↓ | AU↑ | FP↓ | AU↑ | Acc.↑ | Method | FP↓ | AU↑ | FP↓ | AU↑ | FP↓ | AU↑ | Acc.↑ |
|---|---|---|---|---|---|---|---|---|---|---|---|---|---|---|---|
| *CE* | 73.1 | 71.4 | 26.6 | 93.9 | 49.8 | 82.7 | 76.2 | *CE* | 66.7 | 78.5 | 27.3 | 91.9 | 47.0 | 85.2 | **81.1** |
| CIDER | 78.1 | 66.8 | 35.2 | 91.8 | 56.7 | 79.3 | 66.2 | CIDER | 71.9 | 76.1 | 37.2 | 90.7 | 54.6 | 83.4 | 77.6 |
| LogitNorm | 69.5 | 74.0 | 30.8 | 91.8 | 50.2 | 82.9 | 76.8 | LogitNorm | 84.3 | 74.3 | 50.9 | 86.3 | 67.6 | 80.3 | 80.5 |
| PALM | 68.8 | 72.4 | 21.0 | **95.3** | 44.9 | 83.9 | 70.3 | PALM | 70.4 | 79.1 | 31.7 | 91.9 | 51.1 | 85.5 | 77.2 |
| ReweightOOD | 70.7 | 71.3 | **19.7** | 95.2 | 45.2 | 83.3 | 60.5 | ReweightOOD | 72.6 | 77.7 | 34.0 | 90.6 | 53.3 | 84.1 | 71.2 |
| T2FNorm | 68.0 | 75.9 | 34.7 | 90.4 | 51.4 | 83.1 | 76.8 | T2FNorm | 83.2 | 76.1 | 73.7 | 83.4 | 78.5 | 79.8 | 80.5 |
| **ASL (Ours)** | **65.4** | **78.8** | 28.3 | 93.3 | 46.9 | **86.0** | 76.8 | **ASL (Ours)** | **63.3** | **81.9** | **21.2** | **94.0** | **42.2** | **88.0** | 80.9 |

2021), VIM (Wang et al., 2022), ASH (Djurisic et al., 2023), KNN (Sun et al., 2022), and fDBD (Liu & Qin, 2024). 2) Training-based methods: CIDER (Ming et al., 2023), LogitNorm (Wei et al., 2022), PALM (Lu et al., 2024), ReweightOOD (Regmi et al., 2024b), and T2FNorm (Regmi et al., 2024a).

## 4.2 MAIN RESULT

**ASL achieves state-of-the-art performance from scratch on ImageNet-200 and CIFAR benchmarks.** We evaluate ASL under the standard setting of training from scratch on ImageNet-200 and CIFAR datasets. As shown in Table 2, ASL establishes a new state-of-the-art among training-based methods, outperforming the next-best method by 1.4 percentage points in average AU-ROC on near-OOD tasks. On CIFAR-10 and CIFAR-100 (Table 3), ASL consistently outperforms all competing methods. These results demonstrate ASL's effectiveness in learning a highly discriminative feature space without complex training objectives.

**ASL sustains robust OOD detection performance under covariate shift.** We further evaluate ASL under the covariate shift setting, where its robustness is particularly evident. As shown in Table 4, ASL demonstrates a significant advantage in detecting near-OOD samples on ImageNet-200. More strikingly, for far-OOD detection, ASL is the only training-based method that improves

Table 6: OOD Detection Performance of Post-hoc Methods with/without ASL on CIFAR-10.

| | Near-OOD | | Far-OOD | |
| Method | FP↓ | AU↑ | FP↓ | AU↑ |
|---|---|---|---|---|
| MSP | 45.8 | 88.3 | 30.0 | 91.4 |
| MLS | 60.0 | 88.1 | 35.4 | 92.2 |
| ReAct | 62.0 | 86.7 | 42.0 | 91.1 |
| ASH | 89.2 | 73.8 | 76.9 | 77.9 |
| MDS | 49.2 | 85.2 | 33.3 | 89.0 |
| +ASL | **25.1** | **93.3** | **12.0** | **97.2** |
| RMDS | 38.6 | 89.8 | 25.9 | 92.4 |
| +ASL | 33.7 | 90.6 | 18.9 | 94.4 |
| VIM | 45.6 | 88.4 | 28.1 | 92.4 |
| +ASL | 26.2 | 93.0 | 13.1 | 96.9 |
| KNN | 34.4 | 90.7 | 24.4 | 93.1 |
| +ASL | 26.6 | 92.9 | 15.6 | 96.5 |
| fDBD | 35.5 | 90.6 | 24.3 | 93.4 |
| +ASL | 30.3 | 92.1 | 15.5 | 96.5 |
| MD++ | 40.2 | 88.8 | 30.6 | 91.0 |
| +ASL | **25.1** | **93.3** | **12.0** | **97.2** |

upon the standard cross-entropy baseline. This further validates its practicality in potential for real-world application.

**ASL significantly enhances OOD detection when fine-tuning pre-trained models on ImageNet.** To assess ASL in a practical fine-tuning scenario, we apply it to pre-trained ResNet-50 and ViT-B/16

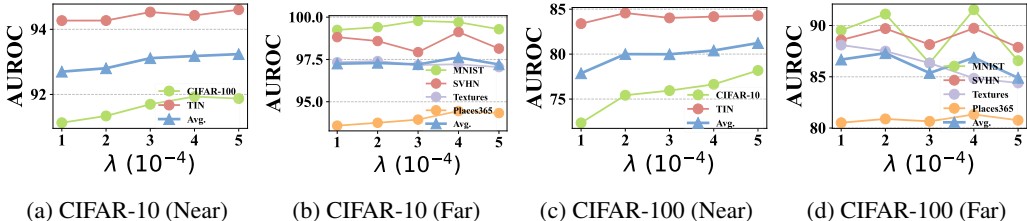

(a) CIFAR-10 (Near)    (b) CIFAR-10 (Far)    (c) CIFAR-100 (Near)    (d) CIFAR-100 (Far)

Figure 4: Ablation study on the weight decay hyperparameter $\lambda$. Performance of ASL across a wide range of $\lambda$ values (from $1 \times 10^{-4}$ to $5 \times 10^{-4}$) on CIFAR-10 and CIFAR-100.

models on the full ImageNet-1K dataset. The results, presented in Table 5, demonstrate substantial improvements. For ResNet-50, fine-tuning with ASL significantly boosts the near-OOD AUROC from a baseline of 71.4% to 78.8%. For ViT-B/16, ASL concurrently improves near-OOD AUROC from 78.5% to 81.9% and Far-OOD AUROC from 91.9% to a leading 94.0%. More experimental results and discussions are provided in Appendix E.4 and E.5. Furthermore, we validate ASL's effectiveness in medical applications, with detailed results provided in Appendix C.

## 4.3 ANALYSIS

**Combine with Distance-Based Methods.** As demonstrated in Table 6, applying various distance-based post-hoc methods to features extracted from an ASL-trained model consistently and significantly enhances their performance compared to using features from a standard cross-entropy model. For example, when paired with ASL features, VIM's near-OOD FPR95 plummets from 45.6% to 26.2%, and its Far-OOD FPR95 is more than halved, decreasing from 28.1% to 13.1%.

**Impact of Weight Decay Hyperparameter $\lambda$.** As shown in Proposition 1, the weight decay hyperparameter $\lambda$ constrains the norms of the classifier weights. We ablate $\lambda$ in Fig. 4. Higher $\lambda$ prompts the model to learn finer features, improving near-OOD detection, while Far-OOD performance remains robust. Thus, $\lambda$ effectively boosts near-OOD detection without harming Far-OOD performance.

**Effectiveness Progressively Increases Across Network Layers.** To understand how ASL influences representation learning throughout the network, we evaluate OOD detection performance using features from different blocks of ResNet-18 on CIFAR-10. Fig. 5 clearly illustrates that ASL consistently learns more discriminative features than both standard cross-entropy and PALM at every network depth. Notably, in Block 4, ASL's performance shows a dramatic leap to an AUROC of 81.3%. This demonstrates that ASL is already learning a highly effective and separable feature space before the final layer.

**Training Efficiency.** ASL offers significant practical advantages in training efficiency. As shown in Table 7, ASL maintains competitive training time per epoch (277s) on ImageNet-200, remaining close to the standard cross-entropy baseline while being substantially faster than other training methods. This efficiency makes our approach not only effective but also highly practical for large-scale applications.

**Comparison of Norm-Based Normalization Schemes.** We investigated the impact of different normalization schemes on out-of-distribution (OOD) detection

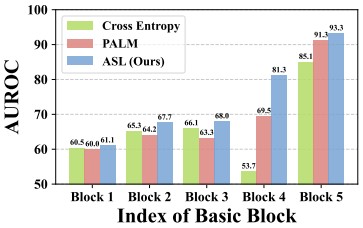

Figure 5: OOD detection performance (AUROC) across different layers.

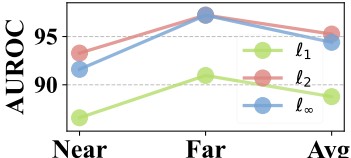

Figure 6: Different Normalization.

Table 7: Training time per epoch (in seconds) on ImageNet-200 for training-based methods.

| Method | Time↓ |
|---|---|
| CE | **252**±1.63 |
| CIDER | 657±4.55 |
| NPOS | 874±2.16 |
| HamOS | 3399±7.76 |
| LogitNorm | 294±1.25 |
| PALM | 381±2.36 |
| ReweightOOD | 406±2.36 |
| T2FNorm | 293±2.05 |
| **ASL (Ours)** | 277±1.70 |

performance, and compared $\ell_1$, $\ell_2$, and $\ell_\infty$ normalization on CIFAR-10. As shown in Fig. 6, $\ell_2$ normalization achieves the best performance. This is consistent with our core motivation of optimizing features on the unit hypersphere. This geometric constraint forces the model to rely solely on angular information for discrimination, thereby reducing sensitivity to feature norms.

## 5   CONCLUSION

In this work, we propose ASL, a training framework that addresses a key challenge in OOD detection by applying $\ell_2$-normalization to features before the cross-entropy loss. This simple yet powerful modification forces the model to learn optimal angular separation, significantly improving its ability to distinguish challenging near-OOD samples. Extensive experiments demonstrate that ASL achieves state-of-the-art performance on major benchmarks. It delivers these results with high training efficiency and proves effective in fine-tuning on large-scale datasets. We hope this work serves as a compelling baseline and encourages a re-evaluation of the complexity-performance Pareto frontier in future research.

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

# APPENDIX

## CONTENTS

## A    PROOF OF PROPOSITION 1

*Proof.* **Proof of i: Angular Contrastive Dynamics.**

Let $z_j = \tilde{f}^\top w_j = s\|w_j\|_2 \cos\theta_j$, where $\theta_j$ is the angle between $\tilde{f}$ and $w_j$. The gradient of the cross-entropy loss with respect to $\tilde{f}$ is:

$$\frac{\partial \mathcal{L}_{\text{CE}}}{\partial \tilde{f}} = \sum_{j=1}^{K}(p_j - \mathbf{1}_{j=y})w_j, \quad \text{where} \quad p_j = \frac{\exp(z_j)}{\sum_k \exp(z_k)}.$$

Let $f = f(x)$. The Jacobian of the normalization $\tilde{f} = s \cdot f/\|f\|_2$ is:

$$\frac{\partial \tilde{f}}{\partial f} = \frac{s}{\|f\|_2}\left(\mathbf{I} - \frac{\tilde{f}\tilde{f}^\top}{s^2}\right).$$

By the chain rule:

$$\frac{\partial \mathcal{L}_{\text{CE}}}{\partial f} = \left(\frac{\partial \tilde{f}}{\partial f}\right)^\top \frac{\partial \mathcal{L}_{\text{CE}}}{\partial \tilde{f}} = \frac{s}{\|f\|_2}\left(\mathbf{I} - \frac{\tilde{f}\tilde{f}^\top}{s^2}\right)\left[\sum_j (p_j - \mathbf{1}_{j=y})w_j\right].$$

The matrix $\mathbf{I} - \tilde{f}\tilde{f}^\top$ projects vectors onto the space orthogonal to $\tilde{f}$. Thus, the radial component (parallel to $\tilde{f}$) of the gradient is:

$$\tilde{f}^\top\left(\frac{\partial \mathcal{L}_{\text{CE}}}{\partial f}\right) = \frac{s}{\|f\|_2}\tilde{f}^\top\left(\mathbf{I} - \frac{\tilde{f}\tilde{f}^\top}{s^2}\right)\left[\sum_j (p_j - \mathbf{1}_{j=y})w_j\right] = 0,$$

This shows that the gradient does not alter $\|f\|_2$ during optimization, as it has no radial component. Only the direction of $f$ (and thus $\tilde{f}$) is updated. The gradient of $z_j$ with respect to the angle $\theta_j$ is:

$$\frac{\partial z_j}{\partial \theta_j} = -s\|w_j\|_2 \sin\theta_j.$$

Using the chain rule, we can express the influence of the loss on the angle $\theta_j$:

$$\frac{\partial \mathcal{L}_{\text{CE}}}{\partial \theta_j} = (p_j - \mathbf{1}_{j=y}) \cdot (-s\|w_j\|_2 \sin\theta_j).$$

For the target class ($j = y$), the term $(p_y - 1)$ is negative, exerting a force to minimize $\theta_y$. For any non-target class ($j \neq y$), $(p_j - 0)$ is positive, exerting a force to maximize $\theta_j$. The magnitudes of these forces are scaled by $\|w_j\|\sin\theta_j$, ensuring active optimization. The projection property of the feature gradient ($\frac{\partial \mathcal{L}_{\text{CE}}}{\partial f}$ has no radial component) confines all updates to the angular domain, thus the optimization inherently minimizes intra-class angles while maximizing inter-class angles.

**Proof of ii: Norm Bound at Convergence.**

At convergence, $\mathbb{E}_{x,y}\left[\frac{\partial \mathcal{L}}{\partial w_j}\right] = 0$. Substituting the full gradient:

$$\mathbb{E}_{x,y}\left[(p_j - \mathbf{1}_{j=y})\tilde{f}\right] + 2\lambda w_j = 0.$$

Taking norms and applying Jensen's inequality:

$$2\lambda\|w_j\|_2 = \left\|\mathbb{E}_{x,y}\left[(\mathbf{1}_{j=y} - p_j)\tilde{f}\right]\right\|_2$$
$$\leq \mathbb{E}_{x,y}\left[\left\|(\mathbf{1}_{j=y} - p_j)\tilde{f}\right\|_2\right]$$
$$= \mathbb{E}_{x,y}\left[|\mathbf{1}_{j=y} - p_j| \cdot s\right] \leq s,$$

where $\|\tilde{f}\|_2 = s$ and $|\mathbf{1}_{j=y} - p_j| \leq 1$. Thus, $\|w_j\|_2 \leq \frac{s}{2\lambda}$. This proves (ii).

$\square$

## B  More Experiments Setup

### B.1  Datasets

We adhere to the benchmark standards established in OpenOOD v1.5 (Zhang et al., 2024) for a rigorous and fair evaluation. Below we detail the in-distribution (ID) and out-of-distribution (OOD) datasets used in our experiments.

**CIFAR-10.** We use the official CIFAR-10 training set with 50,000 samples as $\mathcal{D}_{\text{ID}}^{\text{train}}$. From the test set, we hold out 1,000 samples for validation ($\mathcal{D}_{\text{ID}}^{\text{val}}$) and use the remaining 9,000 as the ID test set ($\mathcal{D}_{\text{ID}}^{\text{test}}$). **Near-OOD** datasets include CIFAR-100 (Krizhevsky et al., 2009) and TinyImageNet (TIN) (Le & Yang, 2015). **Far-OOD** datasets comprise MNIST (Deng, 2012), SVHN (Netzer et al., 2011), Texture (Cimpoi et al., 2014), and Places365 (Zhou et al., 2017).

**CIFAR-100.** We similarly use 50,000 training samples, hold out 1,000 validation samples, and use the remaining 9,000 test samples. **Near-OOD** datasets are CIFAR-10 and TIN. **Far-OOD** datasets are identical to those used for CIFAR-10.

**ImageNet-1K.** We use the standard ImageNet-1K training set (1.2M images) for training. From the validation set, 45,000 images are used as the ID test set ($\mathcal{D}_{\text{ID}}^{\text{test}}$), and the remaining 5,000 serve as the ID validation set ($\mathcal{D}_{\text{ID}}^{\text{val}}$). **Near-OOD** evaluation employs the Semantic Shift Benchmark (SSB-hard) (Vaze et al., 2022) and NINCO (Bitterwolf et al., 2023). **Far-OOD** datasets consist of iNaturalist (Van Horn et al., 2018), Texture (Cimpoi et al., 2014), and OpenImage-O (Wang et al., 2022).

**ImageNet-200.** To reduce the computational burden of training from scratch on the full ImageNet-1K, we use the ImageNet-200 subset, containing the same 200 categories as ImageNet-R (Hendrycks et al., 2021). The near-OOD and far-OOD datasets are consistent with those used for the ImageNet-1K benchmark.

**Covariate Shift Evaluation.** To simulate real-world scenarios where models are required to handle both semantic and covariate shifts, we refer to the evaluation protocol proposed by Yang et al. (2023). Specifically, we expand the in-distribution (ID) test set to include covariate-shifted samples from corrupted datasets such as CIFAR-10-C, CIFAR-100-C, and ImageNet-C (Hendrycks & Dietterich, 2019), while the near-OOD and far-OOD test sets remain consistent with the original benchmark. For ImageNet-200, the ID data comprise subsets of the same 200 categories (consistent with those in ImageNet-R, as mentioned earlier).

**Medical Dataset.** To validate the effectiveness and practical value of ASL in safety-critical, real-world applications, we further experiment on medical images. We refer to the evaluation protocol proposed by Yang et al. (2023). The dataset composition is as follows: **ID** set consists of chest X-ray images from the BIMCV COVID-19+ dataset (Vayá et al., 2020), a large collection of images from the Valencian Region Medical Image Bank, and the model is trained to diagnose COVID-19 based on these images. **Covariate-Shifted ID** set uses chest X-ray images from the ACTUALMED (ActMed) dataset (Wang et al., 2020) to simulate a realistic deployment scenario where a model encounters data from different hospitals or equipment, and these images represent the same diagnostic task (COVID-19 detection) but originate from a different source than the training data. **Near-OOD** category includes medically related but semantically distinct samples from two sources: the RSNA Bone Age dataset (Halabi et al., 2019) (containing bone X-rays, representing an anatomical region shift) and a COVID-19 CT scan dataset (Yang et al., 2020) (representing an imaging modality shift). **Far-OOD** category, which represents samples completely unrelated to the medical domain, consists of standard natural image datasets: MNIST, CIFAR-10, Texture, and TinyImageNet.

### B.2  Training Details

We have listed in Table 8 the training details corresponding to the different data sets. Due to the substantial GPU memory consumption required for training NPOS and HamOS, fine-tuning these models on ImageNet using a single GPU is computationally prohibitive and also entails prohibitively high time costs, we only experimented with them on CIFAR. For experiments on ImageNet-200 and full ImageNet, we employed mixed-precision training to accelerate the process (Batch Size = 128), whereas standard full-precision training was used on CIFAR-10 and CIFAR-100. For the

Table 8: Training hyperparameters for each dataset and architecture used in the experiments. Weight Decay is set to $10^{-4}$ for contrastive learning based methods (CIDER, PALM, ReweightOOD, etc.), and $5 \times 10^{-4}$ for cross-entropy-based methods.

| | CIFAR-10 | CIFAR-100 | ImageNet-200 | ImageNet | ImageNet | BIMCV |
|---|---|---|---|---|---|---|
| Backbone | ResNet-18 | ResNet-18 | ResNet-18 | ResNet-50 | ViT-B/16 | ResNet-18 |
| Training epochs | 100 | 100 | 90 | 30 | 30 | 200 |
| Optimizer | SGD | SGD | SGD | SGD | SGD | SGD |
| Momentum | 0.9 | 0.9 | 0.9 | 0.9 | 0.9 | 0.9 |
| Weight Decay | $(5 \times 10^{-4})/(10^{-4})$ | $(5 \times 10^{-4})/(10^{-4})$ | $(5 \times 10^{-4})/(10^{-4})$ | $(5 \times 10^{-4})/(10^{-4})$ | $(5 \times 10^{-4})/(10^{-4})$ | $(5 \times 10^{-4})/(10^{-4})$ |
| LR schedule | Cos. Anneal | Cos. Anneal | Cos. Anneal | Cos. Anneal | Cos. Anneal | Cos. Anneal |
| Initial LR | $10^{-1}$ | $10^{-1}$ | $10^{-1}$ | $10^{-3}$ | $10^{-5}$ | $10^{-3}$ |
| Min LR | $10^{-6}$ | $10^{-6}$ | $10^{-6}$ | $10^{-6}$ | $10^{-8}$ | $10^{-6}$ |

medical dataset, the temperature was set to 0.05 and scaling factor was set to 20, while other training parameters were kept consistent with the CIFAR-10 experiments.

**Experiment environment.** All experimental results reported in this paper were obtained from three runs performed on an NVIDIA GeForce RTX 3090 GPU (24GB VRAM). The system operated under Ubuntu 20.04. Our deep learning framework was built on Python 3.8.10, PyTorch 1.12.1, and Torchvision 0.13.1, utilizing CUDA 11.3 for accelerated computing.

We strictly adhere to the training parameter settings in OpenOOD for fair comparison, and the detailed descriptions of some training settings are provided below.

**NPOS (Tao et al., 2023).** When using the standard setting in NPOS, we use $k = 300$ neighbors to identify boundary samples, sample 600 candidate outliers per boundary point from a Gaussian distribution with covariance $\sigma^2 = 0.1 \times I$, retain 200 boundary samples per class. Outlier synthesis begins at epoch 40. On CIFAR-10 and CIFAR-100, the batch size is set to 256.

**CIDER (Ming et al., 2023).** The prototype moving average coefficient $\alpha$ is set to 0.5 on CIFAR-100, 0.95 on other datasets. On CIFAR-10 and CIFAR-100, the batch size is set to 512.

**LogitNorm (Wei et al., 2022).** On CIFAR-10 and CIFAR-100, the batch size is set to 128.

**PALM (Lu et al., 2024).** Each ID class is set to have 6 prototypes by default, and the prototype update adopts the Exponential Moving Average (EMA) technique, with the momentum $\alpha$ set to 0.9 on CIFAR-10 and 0.999 on other datasets. On CIFAR-10 and CIFAR-100, the batch size is set to 256.

**ReweightOOD (Regmi et al., 2024b).** On CIFAR-10 and CIFAR-100, the batch size is set to 512.

**T2FNorm (Regmi et al., 2024a).** On CIFAR-10 and CIFAR-100, the batch size is set to 128.

**HamOS (Li & Zhang, 2025).** When we use the default parameters in HamOS, the pre-trained model is fine-tuned for 20 epochs (with an initial learning rate of 0.01 that decays to 0 via cosine annealing), and the batch size is set to 128.

**ASL (ours).** We use the parameters in Table 8. Specifically, we use a linear classification head without bias and set the batch size to 128.

### B.3 BASELINE METHODS

**MDS (Lee et al., 2018).** Based on the multivariate Gaussian assumption, this method models the feature distribution of each class as a Gaussian and uses the Mahalanobis distance to measure the discrepancy between a test sample and class centroids. The detection score is defined as:

$$\text{score}(\boldsymbol{x}) = \max_c -(\boldsymbol{z_x} - \hat{\boldsymbol{\mu}}_c)^\top \hat{\Sigma}^{-1}(\boldsymbol{z_x} - \hat{\boldsymbol{\mu}}_c),$$

where $\hat{\Sigma}$ denotes the shared covariance matrix estimated from all training features.

**RMDS (Ren et al., 2021).** To address the issue that non-discriminative features may hinder the detection of near-OOD samples in traditional MDS, RMDS incorporates a class-agnostic global background Gaussian distribution. The influence of non-discriminative features is mitigated by subtracting the global Mahalanobis distance from the class-conditional one. Specifically, the global background Gaussian is modeled as $\mathcal{N}(\hat{\boldsymbol{\mu}}_0, \hat{\Sigma}_0)$, where $\hat{\boldsymbol{\mu}}_0 = \frac{1}{N} \sum_{i=1}^N \boldsymbol{z}_i$ is the global mean of all training features and $\hat{\Sigma}_0 = \frac{1}{N} \sum_{i=1}^N (\boldsymbol{z}_i - \hat{\boldsymbol{\mu}}_0)(\boldsymbol{z}_i - \hat{\boldsymbol{\mu}}_0)^\top$ is the global covariance matrix.

**VIM (Wang et al., 2022).** This approach combines the energy score from the output space with the norm of the residual in the feature space. A virtual logit term is introduced to enhance the separation between ID and OOD samples:

$$\text{score}(\boldsymbol{x}) = \text{Energy}(\boldsymbol{x}) + \alpha \cdot \|\boldsymbol{R}^\top \boldsymbol{z_x}\|_2,$$

where $\boldsymbol{R}$ represents the basis vectors spanning the residual space of the training feature matrix.

**KNN (Sun et al., 2022).** Detection is performed based on the distance between the test sample feature and its $k$-th nearest neighbor in the normalized training feature set. The scoring function is:

$$\text{score}(\boldsymbol{x}) = -\|\boldsymbol{z_x} - \boldsymbol{z}_{(k)}\|_2,$$

where $\boldsymbol{z}_{(k)}$ denotes the feature of the $k$-th nearest training sample.

**fDBD (Liu & Qin, 2024).** This method uses a closed-form estimate of the distance from the sample feature to each class decision boundary. The score is regularized by the deviation of the feature from the training feature mean, resulting in the following formulation:

$$\text{score}(\boldsymbol{x}) = \frac{1}{|\mathcal{C}| - 1} \sum_{c \neq f(\boldsymbol{x})} \frac{\tilde{D}_f(\boldsymbol{z_x}, c)}{\|\boldsymbol{z_x} - \boldsymbol{\mu}_{\text{train}}\|_2}.$$

**MD++ (Müller & Hein, 2025).** This approach improves traditional Mahalanobis distance by incorporating $\ell_2$-normalization of features. It aims to address core limitations of the original method, namely large intra- and inter-class variation in feature norms and violation of the shared covariance Gaussian assumption, thereby aligning the feature distribution more closely with the theoretical prerequisites of the Mahalanobis distance.

Table 9: OOD detection performance on BIMCV under standard setting.

| | Near-OOD | | | | | | Far-OOD | | | | | | | | | |
| | CT-SCAN | | XRayBone | | Avg. | | MNIST | | Texture | | CIFAR-10 | | TIN | | Avg. | |
| Method | FP↓ | AU↑ | FP↓ | AU↑ | FP↓ | AU↑ | FP↓ | AU↑ | FP↓ | AU↑ | FP↓ | AU↑ | FP↓ | AU↑ | FP↓ | AU↑ |
|---|---|---|---|---|---|---|---|---|---|---|---|---|---|---|---|---|
| *Post-hoc Methods* | | | | | | | | | | | | | | | | |
| MSP | 95.9±3.25 | 33.8±6.25 | 90.1±10.3 | 59.4±8.71 | 93.0±6.79 | 46.6±7.48 | 64.1±1.03 | 69.3±3.35 | 100.0±0.00 | 34.5±1.80 | 97.8±0.48 | 48.6±0.98 | 98.0±0.79 | 47.6±4.03 | 90.0±0.34 | 50.0±2.54 |
| MLS | 95.1±2.94 | 36.2±4.00 | 90.1±11.0 | 60.3±12.3 | 92.6±6.94 | 48.3±8.13 | 61.5±2.54 | 73.0±5.96 | 100.0±0.00 | 34.3±0.61 | 98.0±0.63 | 48.6±0.21 | 98.0±0.87 | 47.6±3.00 | 89.4±0.69 | 50.8±2.44 |
| ReAct | 91.4±0.71 | 47.8±1.13 | 91.9±11.0 | 60.1±14.6 | 91.7±5.83 | 54.0±6.73 | 42.4±1.51 | 85.7±1.93 | 100.0±0.00 | 37.2±0.99 | 98.1±0.79 | 49.7±1.16 | 98.2±0.16 | 48.5±1.21 | 84.7±0.62 | 55.3±0.24 |
| ASH | 91.3±4.52 | 42.8±2.38 | 78.9±19.4 | 70.1±10.8 | 85.1±11.9 | 56.4±6.59 | 51.2±5.32 | 77.1±8.17 | 99.9±0.00 | 42.3±0.08 | 95.5±3.33 | 52.0±0.32 | 96.3±0.32 | 52.0±2.49 | 85.7±0.42 | 55.9±2.57 |
| MDS | 16.1±0.87 | 96.9±0.08 | 28.5±15.8 | 94.9±2.30 | 22.3±8.33 | 95.9±1.11 | 0.11±0.00 | 100.0±0.02 | 9.43±3.65 | 98.1±0.57 | 19.0±3.10 | 95.0±1.10 | 18.5±5.00 | 95.0±1.68 | 11.8±2.94 | 97.0±0.84 |
| +ASL | 21.1±1.90 | 94.8±0.15 | 10.4±4.76 | 97.7±0.94 | 15.8±3.33 | 96.3±0.55 | 0.06±0.08 | 100.0±0.01 | 11.6±3.41 | 97.7±0.32 | 11.8±3.73 | 97.3±0.30 | 12.5±4.13 | 97.3±0.29 | 8.99±2.84 | 98.1±0.23 |
| RMDS | 93.9±1.19 | 45.1±5.75 | 89.3±7.22 | 56.0±8.28 | 91.6±4.21 | 50.5±7.01 | 89.7±2.38 | 54.4±4.76 | 95.4±0.32 | 55.9±0.30 | 96.2±0.32 | 51.7±1.70 | 96.1±0.56 | 51.6±1.87 | 94.3±0.30 | 53.4±0.22 |
| +ASL | 92.7±0.16 | 48.8±1.00 | 94.4±0.40 | 51.3±12.6 | 93.6±0.28 | 50.0±5.80 | 83.2±18.9 | 61.3±16.4 | 94.4±0.79 | 53.7±3.68 | 95.0±0.40 | 51.1±2.09 | 95.5±0.40 | 51.0±1.98 | 92.0±5.12 | 54.3±6.03 |
| VIM | 90.1±5.08 | 57.5±5.74 | 85.1±15.0 | 76.0±8.07 | 87.6±10.0 | 66.7±6.90 | 3.59±2.54 | 99.0±0.71 | 99.6±0.48 | 51.3±1.01 | 94.9±2.46 | 63.5±1.53 | 95.4±0.63 | 62.6±1.25 | 73.4±1.21 | 69.1±0.50 |
| +ASL | 78.4±2.14 | 74.4±3.45 | 74.5±19.7 | 80.1±9.61 | 76.5±10.9 | 77.2±6.53 | 20.4±28.6 | 97.2±3.81 | 98.2±0.79 | 47.8±2.44 | 94.2±0.24 | 62.4±1.09 | 94.8±1.75 | 59.9±3.96 | 76.9±6.45 | 66.8±0.92 |
| KNN | 46.0±3.57 | 87.9±1.92 | 5.33±1.83 | 98.9±0.00 | 25.6±2.70 | 93.4±0.96 | 0.28±0.24 | 100.0±0.00 | 29.1±1.59 | 95.0±0.05 | 40.5±0.63 | 91.7±0.13 | 38.7±0.56 | 92.6±0.04 | 27.1±0.75 | 94.8±0.05 |
| +ASL | 36.9±7.30 | 90.5±1.62 | 7.86±0.00 | 98.3±0.02 | 22.4±3.65 | 94.4±0.82 | 0.51±0.40 | 99.9±0.08 | 10.4±2.78 | 97.9±0.31 | 17.8±4.21 | 95.3±0.38 | 17.1±3.97 | 95.8±0.30 | 11.4±2.84 | 97.2±0.26 |
| fDBD | 88.0±0.87 | 49.2±2.21 | 74.8±8.33 | 70.4±6.40 | 81.4±4.60 | 59.8±4.31 | 35.2±3.17 | 84.3±0.27 | 92.2±0.71 | 47.2±0.86 | 90.7±0.24 | 54.3±0.80 | 90.6±0.24 | 54.1±3.20 | 77.2±0.97 | 60.0±1.15 |
| +ASL | 90.2±1.83 | 51.9±4.13 | 80.2±4.60 | 66.5±3.40 | 85.2±3.21 | 59.2±3.77 | 57.1±14.8 | 71.4±5.96 | 92.8±1.59 | 48.5±0.44 | 88.1±0.32 | 55.5±0.91 | 90.0±0.63 | 54.5±0.67 | 82.0±3.08 | 57.5±1.44 |
| MD++ | 38.3±12.1 | 91.2±3.06 | 3.87±2.30 | 98.8±0.68 | 21.1±7.22 | 95.0±1.87 | 0.22±0.32 | 100.0±0.06 | 42.3±9.44 | 93.9±0.85 | 28.9±4.05 | 94.8±0.60 | 28.2±0.16 | 95.1±0.12 | 24.9±3.49 | 95.9±0.41 |
| +ASL | 21.1±1.90 | 94.8±0.15 | 10.4±4.76 | 97.7±0.94 | 15.8±3.33 | 96.3±0.55 | 0.06±0.08 | 100.0±0.01 | 11.6±3.41 | 97.7±0.32 | 11.8±3.73 | 97.3±0.30 | 12.5±4.13 | 97.3±0.29 | 8.99±2.84 | 98.1±0.23 |
| *Training-based Methods* | | | | | | | | | | | | | | | | |
| CE | 38.3±12.1 | 91.2±3.06 | 3.87±2.30 | 98.8±0.68 | 21.1±7.22 | 95.0±1.87 | 0.22±0.32 | 100.0±0.06 | 42.3±9.44 | 93.9±0.85 | 28.9±4.05 | 94.8±0.60 | 28.2±0.16 | 95.1±0.12 | 24.9±3.49 | 95.9±0.41 |
| NPOS | 72.5±6.03 | 66.0±13.9 | 54.1±46.0 | 79.7±24.6 | 63.3±26.0 | 72.8±19.3 | 1.74±1.03 | 99.4±0.26 | 73.1±26.5 | 65.0±26.5 | 77.0±22.1 | 61.4±14.6 | 75.9±23.8 | 63.4±19.7 | 56.9±18.4 | 72.3±15.2 |
| CIDER | 55.7±29.8 | 70.8±22.5 | 37.7±8.89 | 78.7±4.44 | 46.7±10.4 | 74.7±9.03 | 20.6±5.32 | 91.3±5.02 | 48.3±12.0 | 79.6±4.45 | 45.9±10.5 | 80.3±5.63 | 47.5±11.7 | 78.3±8.28 | 40.6±9.88 | 82.4±5.94 |
| LogitNorm | 99.9±0.16 | 31.0±17.4 | 94.4±7.86 | 36.3±20.9 | 97.2±3.85 | 33.7±1.75 | 100.0±0.00 | 12.5±16.8 | 99.9±0.16 | 40.8±5.46 | 96.0±5.71 | 49.4±8.20 | 98.0±2.86 | 48.6±6.72 | 98.5±2.18 | 37.8±0.90 |
| PALM | 96.2±0.63 | 13.8±0.43 | 76.2±1.59 | 85.3±1.18 | 86.2±0.48 | 49.6±0.37 | 92.7±0.00 | 12.6±0.77 | 85.6±1.43 | 71.8±0.78 | 88.6±1.19 | 54.0±1.27 | 88.5±1.03 | 52.1±1.55 | 88.9±0.97 | 47.6±1.09 |
| ReweightOOD | 44.9±13.2 | 86.7±5.24 | 32.6±3.41 | 95.1±0.10 | 38.7±8.29 | 90.9±2.67 | 9.09±10.8 | 97.6±2.61 | 8.02±1.83 | 98.5±0.13 | 19.0±5.32 | 96.0±0.97 | 17.0±3.57 | 96.4±0.68 | 13.3±0.02 | 97.1±0.21 |
| T2FNorm | 86.9±6.51 | 54.7±9.37 | 84.1±17.5 | 63.0±12.8 | 85.5±12.0 | 58.8±11.1 | 55.9±26.5 | 75.6±12.9 | 100.0±0.00 | 31.4±1.87 | 98.3±0.87 | 48.1±0.63 | 98.8±1.67 | 45.0±2.91 | 88.2±5.99 | 50.0±1.88 |
| HamOS | 87.3±1.27 | 55.5±1.11 | 88.9±12.9 | 49.0±24.5 | 88.1±5.79 | 52.2±12.8 | 60.0±35.2 | 51.0±23.8 | 84.9±4.05 | 70.8±3.17 | 89.1±6.59 | 62.6±1.00 | 86.6±8.41 | 64.6±1.89 | 80.2±13.6 | 62.2±5.89 |
| **ASL (Ours)** | 21.1±1.90 | 94.8±0.15 | 10.4±4.76 | 97.7±0.94 | **15.8±3.33** | **96.3±0.55** | 0.06±0.08 | 100.0±0.01 | 11.6±3.41 | 97.7±0.32 | 11.8±3.73 | 97.3±0.30 | 12.5±4.13 | 97.3±0.29 | **8.99±2.84** | **98.1±0.23** |

## C EXPERIMENT ON MEDICAL DATASET BIMCV

To further verify the validity and practical value in safety-critical applications, further experiments were performed on the BIMCV Medical Image Dataset. This benchmark also involves detecting out-of-distribution samples under both standard and covariate shift conditions, simulating realistic deployment scenarios where a model encounters data from different hospitals or equipment.

**Under the standard setting** (Table 9), ASL demonstrates strong performance among training-based methods. Notably, it reduces the average FPR95 on near-OOD tasks to 15.8% from the standard cross-entropy baseline's 21.1%. The improvement is even more significant on far-OOD tasks, where

Table 10: OOD detection performance on BIMCV under covariate shift.

| | Near-OOD | | | | | | Far-OOD | | | | | | | | | |
| | CT-SCAN | | XRayBone | | Avg. | | MNIST | | Texture | | CIFAR-10 | | TIN | | Avg. | |
| Method | FP↓ | AU↑ | FP↓ | AU↑ | FP↓ | AU↑ | FP↓ | AU↑ | FP↓ | AU↑ | FP↓ | AU↑ | FP↓ | AU↑ | FP↓ | AU↑ |
|---|---|---|---|---|---|---|---|---|---|---|---|---|---|---|---|---|
| *Post-hoc Methods* | | | | | | | | | | | | | | | | |
| MSP | 96.2±3.11 | 33.3±6.63 | 90.8±9.82 | 59.1±9.34 | 93.5±6.46 | 46.2±7.99 | 64.8±0.35 | 69.3±2.70 | 100.0±0.00 | 34.1±1.44 | 98.0±0.41 | 48.2±0.48 | 98.2±0.69 | 47.2±3.56 | 90.3±0.16 | 49.7±2.04 [t] |
| MLS | 95.5±2.90 | 35.7±4.42 | 90.8±10.4 | 60.2±12.9 | 93.2±6.64 | 47.9±8.67 | 62.1±1.80 | 73.1±5.35 | 100.0±0.00 | 33.9±0.25 | 98.2±0.55 | 48.3±0.30 | 98.3±0.76 | 47.3±2.53 | 89.7±0.50 | 50.6±1.96 |
| ReAct | 92.0±1.04 | 47.5±0.50 | 92.3±10.4 | 60.0±15.3 | 92.2±5.74 | 53.8±7.41 | 42.4±2.70 | 86.0±1.35 | 100.0±0.00 | 36.9±1.48 | 98.3±0.76 | 49.5±1.79 | 98.4±0.21 | 48.3±0.61 | 84.8±0.92 | 55.2±0.33 |
| ASH | 91.9±4.35 | 42.5±2.87 | 79.5±19.6 | 70.2±11.4 | 85.7±12.0 | 56.4±7.15 | 51.2±4.77 | 77.6±7.48 | 99.9±0.00 | 42.2±0.47 | 95.9±3.11 | 51.9±0.82 | 96.7±0.28 | 51.9±2.03 | 85.9±0.35 | 55.9±2.06 |
| MDS | 14.9±0.90 | 97.2±0.09 | 27.1±15.7 | 95.3±2.24 | 21.0±8.29 | 96.2±1.07 | 0.10±0.00 | 100.0±0.02 | 8.41±3.32 | 98.3±0.51 | 17.6±3.11 | 95.5±1.00 | 17.2±4.84 | 95.5±1.54 | 10.8±2.52 | 97.3±0.77 |
| +ASL | 20.7±2.14 | 95.0±0.30 | 9.82±4.91 | 97.8±1.00 | 15.2±3.53 | 96.4±0.65 | 0.05±0.07 | 100.0±0.01 | 11.0±3.73 | 97.8±0.39 | 11.3±4.08 | 97.4±0.40 | 11.9±4.56 | 97.4±0.39 | 8.58±3.11 | 98.2±0.30 |
| RMDS | 93.6±1.04 | 45.3±4.92 | 89.1±6.77 | 56.1±7.42 | 91.3±3.91 | 50.7±6.17 | 89.3±2.90 | 54.6±5.59 | 95.1±0.14 | 56.0±0.39 | 95.9±0.21 | 51.9±0.94 | 95.9±0.55 | 51.8±1.11 | 94.1±0.50 | 53.6±0.98 |
| +ASL | 92.7±0.14 | 48.4±0.09 | 94.3±0.14 | 51.0±13.3 | 93.5±0.00 | 49.7±6.62 | 83.6±18.1 | 60.9±15.4 | 94.4±0.62 | 53.3±2.86 | 94.9±0.28 | 50.8±1.20 | 95.4±0.14 | 50.7±1.10 | 92.1±4.79 | 53.9±5.14 |
| VIM | 90.9±4.98 | 57.2±6.21 | 85.7±14.9 | 76.0±8.30 | 88.3±9.95 | 66.6±7.26 | 3.27±2.42 | 99.0±0.71 | 99.6±0.41 | 51.1±1.25 | 95.3±2.49 | 63.4±1.85 | 95.9±0.55 | 62.4±0.97 | 73.5±1.19 | 69.0±0.71 |
| +ASL | 80.0±2.35 | 73.7±3.43 | 76.1±19.4 | 79.7±9.66 | 78.0±10.3 | 76.7±6.55 | 21.1±29.6 | 97.1±3.90 | 98.4±0.69 | 47.0±2.52 | 94.9±0.28 | 61.7±1.19 | 95.5±1.59 | 59.1±4.09 | 77.5±6.76 | 66.2±0.93 |
| KNN | 46.4±4.22 | 88.3±1.96 | 4.79±1.80 | 99.0±0.01 | 25.6±3.01 | 93.7±0.98 | 0.24±0.21 | 100.0±0.00 | 28.4±1.87 | 95.2±0.04 | 40.6±0.35 | 92.0±0.07 | 38.4±0.41 | 92.9±0.00 | 26.9±0.71 | 95.0±0.03 |
| +ASL | 37.0±7.74 | 90.9±1.68 | 6.99±0.07 | 98.5±0.07 | 22.0±3.91 | 94.7±0.87 | 0.44±0.35 | 99.9±0.07 | 9.38±2.76 | 98.1±0.32 | 16.5±4.22 | 95.7±0.44 | 15.8±4.08 | 96.1±0.36 | 10.5±2.85 | 97.4±0.30 |
| fDBD | 88.6±0.90 | 49.2±2.89 | 75.2±9.19 | 70.5±7.01 | 81.9±5.05 | 59.8±4.95 | 35.0±4.42 | 84.6±0.68 | 92.5±0.83 | 47.1±0.36 | 91.2±0.07 | 54.2±0.23 | 91.1±0.35 | 54.1±2.68 | 77.4±1.38 | 60.0±0.65 |
| +ASL | 90.9±1.45 | 51.4±4.43 | 81.2±4.35 | 66.1±3.81 | 86.0±2.90 | 58.7±4.12 | 58.0±14.9 | 71.1±5.48 | 93.4±1.24 | 48.0±0.69 | 89.0±0.00 | 55.0±0.58 | 90.8±0.35 | 54.1±0.99 | 82.8±3.34 | 57.0±1.10 |
| MD++ | 38.4±12.2 | 91.4±2.86 | 3.47±2.00 | 98.9±0.61 | 20.9±7.12 | 95.2±1.74 | 0.20±0.28 | 100.0±0.06 | 42.8±9.12 | 93.9±0.78 | 28.5±3.53 | 94.9±0.51 | 27.6±0.35 | 95.2±0.04 | 24.8±3.14 | 96.0±0.35 |
| +ASL | 20.7±2.14 | 95.0±0.30 | 9.82±4.91 | 97.8±1.00 | 15.2±3.53 | 96.4±0.65 | 0.05±0.07 | 100.0±0.01 | 11.0±3.73 | 97.8±0.39 | 11.3±4.08 | 97.4±0.40 | 11.9±4.56 | 97.4±0.39 | 8.58±3.11 | 98.2±0.30 |
| *Training-based Methods* | | | | | | | | | | | | | | | | |
| CE | 38.4±12.2 | 91.4±2.86 | 3.47±2.00 | 98.9±0.61 | 20.9±7.12 | 95.2±1.74 | 0.20±0.28 | 100.0±0.06 | 42.8±9.12 | 93.9±0.78 | 28.5±3.53 | 94.9±0.51 | 27.6±0.35 | 95.2±0.04 | 24.8±3.14 | 96.0±0.35 |
| NPOS | 72.7±7.88 | 66.1±15.2 | 53.8±47.6 | 79.5±25.2 | 63.3±27.8 | 72.8±20.2 | 1.71±1.18 | 99.4±0.31 | 72.0±28.5 | 65.0±27.6 | 76.5±23.6 | 61.7±16.0 | 75.1±25.6 | 63.6±20.9 | 56.3±19.7 | 72.4±16.2 |
| CIDER | 57.2±27.1 | 70.9±21.7 | 39.3±11.6 | 77.9±5.86 | 48.2±7.74 | 74.4±7.90 | 21.7±7.19 | 91.1±5.51 | 49.3±14.7 | 78.9±6.18 | 46.8±13.4 | 79.6±6.98 | 48.5±14.5 | 77.5±9.86 | 41.6±12.4 | 81.7±7.13 |
| LogitNorm | 99.9±0.14 | 30.5±17.9 | 95.1±6.98 | 35.5±19.7 | 97.5±3.42 | 33.0±0.92 | 100.0±0.00 | 11.8±15.9 | 99.9±0.14 | 40.1±6.38 | 96.4±5.11 | 48.6±9.26 | 98.2±2.56 | 47.9±7.77 | 98.6±1.95 | 37.1±1.88 |
| PALM | 96.5±0.55 | 13.5±0.36 | 75.8±1.94 | 85.4±1.20 | 86.1±0.69 | 49.5±0.42 | 93.4±0.00 | 12.3±0.72 | 85.9±1.45 | 71.8±0.70 | 89.0±1.18 | 54.0±1.12 | 88.9±0.97 | 52.2±1.39 | 89.3±0.90 | 47.6±0.98 |
| ReweightOOD | 45.7±13.8 | 86.7±5.55 | 32.9±3.66 | 95.2±0.15 | 39.3±8.74 | 91.0±2.85 | 8.85±10.7 | 97.7±2.51 | 7.67±2.00 | 98.5±0.15 | 19.3±6.36 | 96.1±1.03 | 17.2±4.42 | 96.5±0.73 | 13.2±0.52 | 97.2±0.15 |
| T2FNorm | 88.0±6.01 | 53.8±9.40 | 85.3±16.4 | 62.3±12.9 | 86.7±11.2 | 58.0±11.2 | 57.2±27.2 | 75.2±13.2 | 100.0±0.00 | 30.7±1.91 | 98.5±0.76 | 47.3±0.73 | 99.0±1.45 | 44.2±2.99 | 88.7±6.26 | 49.3±1.90 |
| HamOS | 87.3±0.62 | 55.9±0.62 | 88.7±13.5 | 49.2±24.4 | 88.0±6.63 | 52.5±12.5 | 60.5±35.0 | 51.0±23.2 | 84.8±4.70 | 71.1±3.51 | 89.0±7.05 | 62.8±0.59 | 86.6±8.92 | 64.9±1.46 | 80.2±13.9 | 62.5±5.43 |
| **ASL (Ours)** | 20.7±2.14 | 95.0±0.30 | 9.82±4.91 | 97.8±1.00 | **15.2±3.53** | **96.4±0.65** | 0.05±0.07 | 100.0±0.01 | 11.0±3.73 | 97.8±0.39 | 11.3±4.08 | 97.4±0.40 | 11.9±4.56 | 97.4±0.39 | **8.58±3.11** | **98.2±0.30** |

the FPR95 drops from 24.9% to 8.99%. This suggests a reduced tendency to misclassify OOD samples, which is critical for medical applications.

**Under the covariate shift setting** (Table 10), ASL continues to show its effectiveness. It maintains a low average FPR95 of 15.2% for near-OOD and 8.58% for far-OOD tasks, compared to the baseline's 20.9% and 24.8%, respectively. This result highlights ASL's potential for robustness against domain shifts, a common issue in medical imaging.

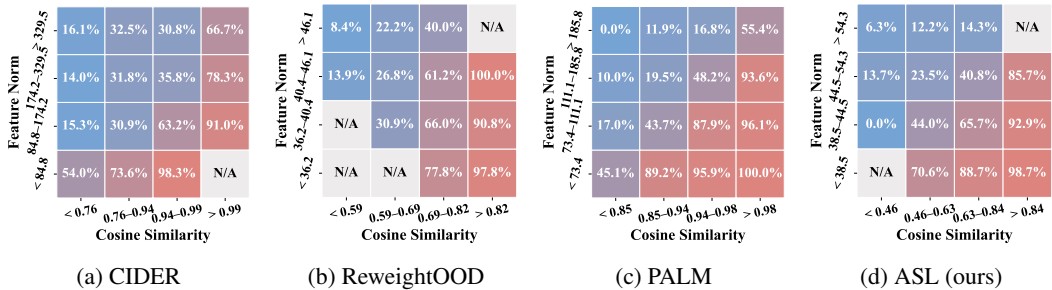

|  (a) CIDER | (b) ReweightOOD | (c) PALM | (d) ASL (ours) |

Figure 7: Heatmap of ID sample distribution in CIFAR-10 Mahalanobis feature space.

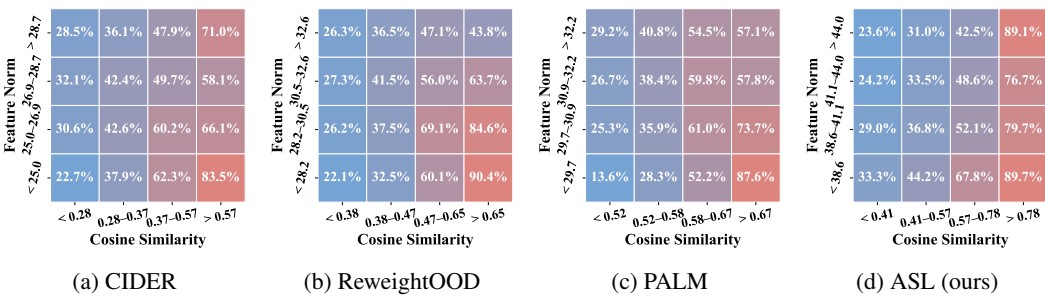

|  (a) CIDER | (b) ReweightOOD | (c) PALM | (d) ASL (ours) |

Figure 8: Heatmap of ID sample distribution in CIFAR-100 Mahalanobis feature space.

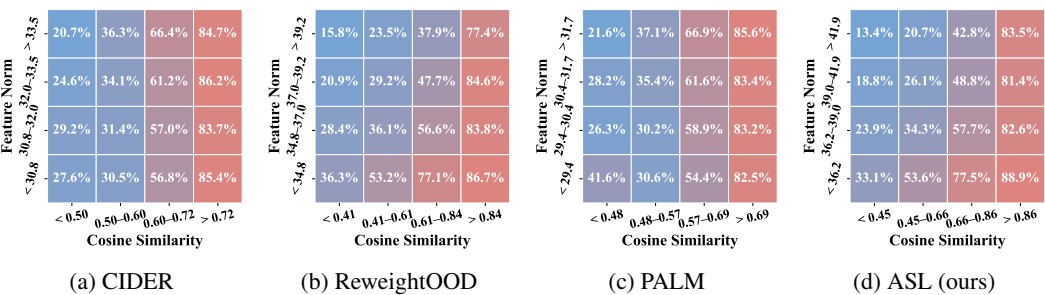

(a) CIDER  (b) ReweightOOD  (c) PALM  (d) ASL (ours)

Figure 9: Heatmap of ID sample distribution in ImageNet-200 Mahalanobis feature space.

# D  MORE VISUALIZATION OF LEARNED FEATURE SPACE

## D.1  HEATMAP ANALYSIS

To visually inspect the geometric structure of the feature space learned by different methods, we further performed a heatmap visualization on CIFAR-10, CIFAR-100, and ImageNet-200 to illustrate the joint distribution of samples within the distribution according to the feature norm and cosine similarity with the nearest ID class prototype. As shown in the Fig. 7, 8, and 9, each cell in the heatmap shows the proportion of ID samples belonging to a certain feature norm and cosine similarity range.

Notably, ASL yields a distinctly more structured feature geometry compared to other methods. Two observations highlight its effectiveness: First, ID samples are highly concentrated in regions with high cosine similarity (right side of each heatmap), indicating stronger angular compactness around the class means. Second, within these high-similarity regions, ID features are further clustered in areas with low feature norms. As shown in equation 2, this results in higher OOD detection scores for ID samples.

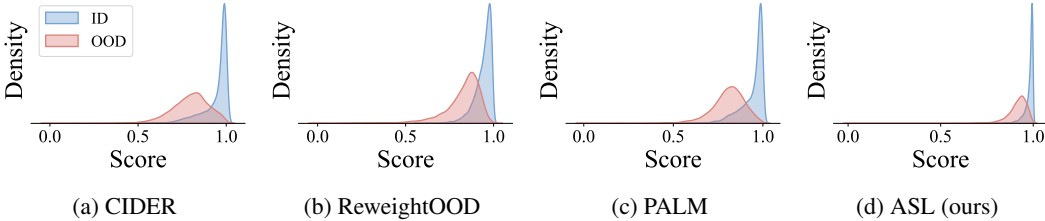

(a) CIDER  (b) ReweightOOD  (c) PALM  (d) ASL (ours)

Figure 10: Kernel Density Estimation (KDE) of OOD detection scores on CIFAR-10.

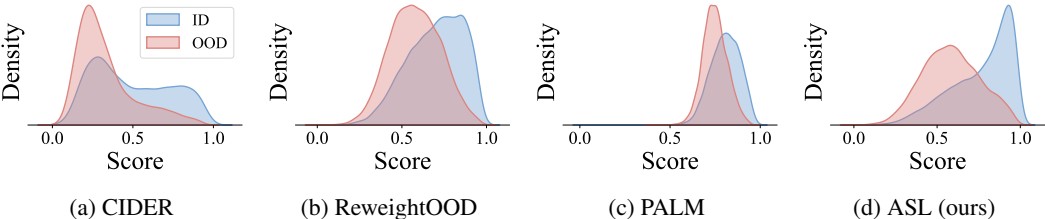

(a) CIDER  (b) ReweightOOD  (c) PALM  (d) ASL (ours)

Figure 11: KDE of OOD detection scores on CIFAR-100.

## D.2  SCORE DISTRIBUTION ANALYSIS

To further evaluate the effectiveness of ASL, we analyze the separability of OOD detection scores using Kernel Density Estimation (KDE) plots. These plots visualize the distribution of scores for ID samples (blue) versus OOD samples (red), with a larger separation and lower overlap between the two distributions indicating better detection performance.

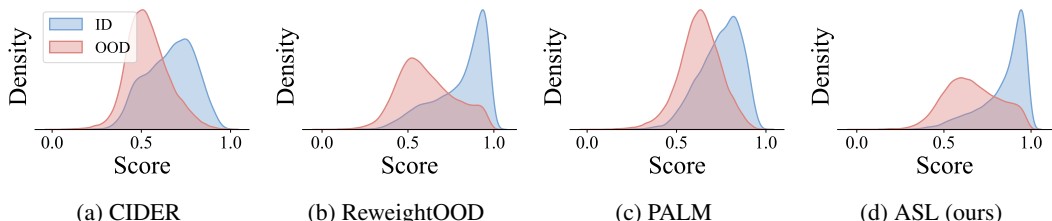

Figure 12: KDE of OOD detection scores on ImageNet-200.

As shown in Fig. 10, 11, and 12, ASL consistently produces a more distinct separation between the ID and OOD score distributions compared to other state-of-the-art methods like CIDER, ReweightOOD, and PALM. For ASL, the ID score distribution forms a tight, sharp peak at the higher end of the score range, while the OOD score distribution is pushed towards the lower end. This clear separation minimizes the ambiguity between ID and OOD samples. In contrast, the score distributions for other methods exhibit significantly more overlap, suggesting a higher rate of confusion and misclassification. These visualizations empirically confirm that the angularly discriminative feature space learned by ASL translates directly into a more reliable and effective scoring mechanism for OOD detection across various benchmarks.

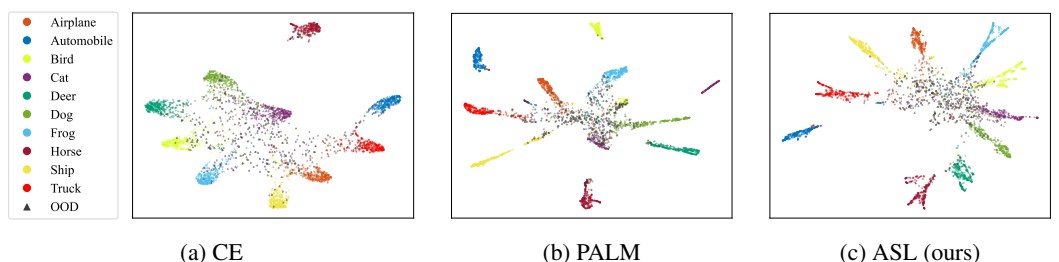

Figure 13: UMAP Visualization of CIFAR-10 Feature Space.

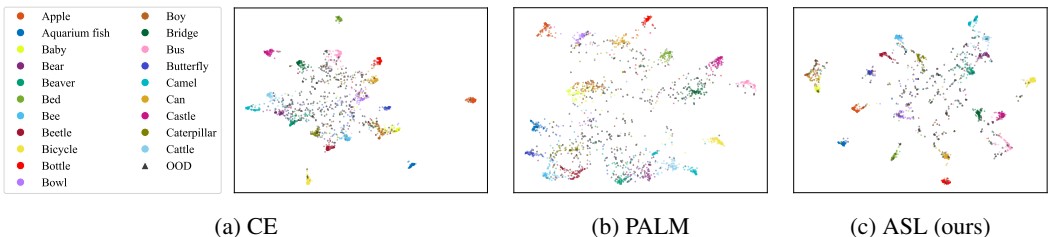

Figure 14: UMAP Visualization of CIFAR-100 Feature Space.

## D.3 UMAP VISUALIZATION ANALYSIS OF FEATURE SPACE

To directly verify the hyperspherical geometry of features learned by ASL and the relative positioning of ID and OOD samples, we conduct UMAP dimensionality reduction visualization on the feature spaces of CIFAR-10 (Fig. 13) and CIFAR-100 (Fig. 14, first 20 subclasses).

On the CIFAR-10 dataset, ASL exhibits a structured feature geometry highly similar to that of PALM (contrastive learning method). More notably, a distinct distribution pattern emerges: OOD samples (marked in gray in Figure 13) are predominantly concentrated in the central region of the UMAP projection, while ID samples of different classes form compact clusters that surround this central area. On the CIFAR-100 dataset, ASL still maintains intra-class clustering performance. Concurrently, OOD samples in ASL's feature space tend to aggregate in the gaps between adjacent ID clusters rather than infiltrating the ID clusters themselves.

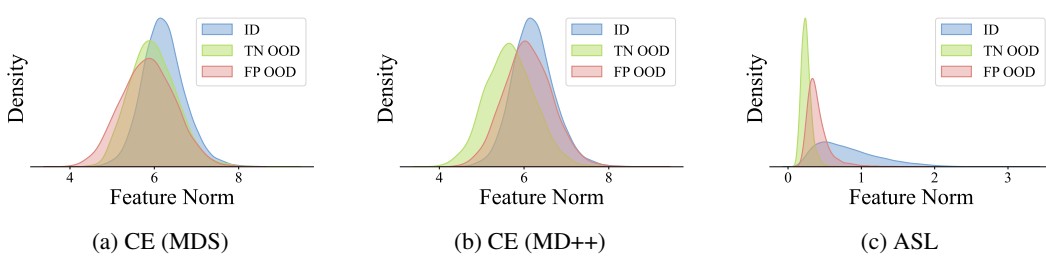

Figure 15: Feature norm distributions on CIFAR-10.

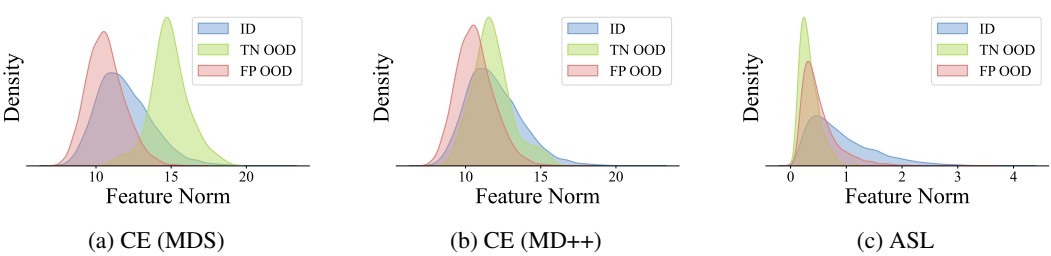

Figure 16: Feature norm distributions on CIFAR-100.

### D.4 VISUALIZATION OF LOW-NORM FALSE-POSITIVE

To further demonstrate the impact of low-norm out-of-distribution (OOD) samples on detection performance, we visualize the feature norm distribution of OOD false positives. Specifically, a feature norm threshold is established on the in-distribution (ID) validation set to achieve a 95% true positive rate for ID samples. Based on this threshold, an out-of-distribution (OOD) sample exceeding the threshold is a false positive (FP), while one below it is a true negative (TN). As shown in Fig. 15 (CIFAR-10), 16 (CIFAR-100), and 17 (ImageNet-200), ASL-trained features exhibit a clearer separation between ID and OOD norm distributions. In contrast, CE (MDS) and CE (MD++) show substantial overlap between ID and OOD norm distributions, with OOD samples infiltrating the low-norm region where ID samples reside. Especially on CIFAR-100 and ImageNet-200, a considerable portion of false positives for MDS and MD++ are concentrated in the low-norm tail, while ASL mitigates this risk.

Table 11: Comparison of different normalization methods at a conceptual level. ASL is distinguished by its direct focus on optimizing angular separation and its bias-free, prototype-based formulation.

| Method | Normalization | Loss Function | Bias Term | Primary Motivation |
|---|---|---|---|---|
| Standard CE | None | Cross-Entropy | Yes | Maximize classification margin |
| LogitNorm | Logits | Modified CE | Yes | Mitigate overconfidence on OOD samples |
| T2FNorm | Features ($l_2$) | Cross-Entropy | Yes | Improve logit-space separability at test time |
| Haas et al. (2023) | Features ($l_2$) | Cross-Entropy | Yes | Analyze link to and acceleration of Neural Collapse |
| **ASL (ours)** | **Features ($l_2$)** | **Cross-Entropy** | **No** | **Optimize angular separation for OOD robustness** |

## E FURTHER DISCUSSION

### E.1 COMPARISON OF DIFFERENT NORMALIZATION METHODS

To situate our work more clearly within the landscape of normalization-based approaches for improving model robustness and OOD detection, we provide a detailed conceptual and empirical comparison with closely related methods here.

**Conceptual Distinctions.** We list the specifics of the different normalization methods in Table 11, **LogitNorm**, for instance, operates on the network's final outputs (logits) to mitigate overconfi-

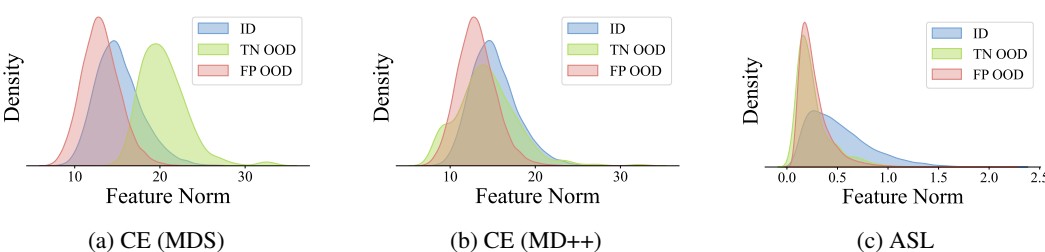

Figure 17: Feature norm distributions on ImageNet-200.

Table 12: OOD detection performance of different normalization methods on CIFAR-10 and CIFAR-100. ASL demonstrates superior performance, especially on the more challenging near-OOD and CIFAR-100 benchmarks.

| Method | CIFAR-10 | | | | | | CIFAR-100 | | | | | |
|---|---|---|---|---|---|---|---|---|---|---|---|---|
| | Near-OOD | | Far-OOD | | Avg. | | Near-OOD | | Far-OOD | | Avg. | |
| | FP↓ | AU↑ | FP↓ | AU↑ | FP↓ | AU↑ | FP↓ | AU↑ | FP↓ | AU↑ | FP↓ | AU↑ |
| CE | 40.2 | 88.8 | 30.6 | 91.0 | 35.4 | 89.9 | 74.8 | 67.8 | 51.9 | 82.6 | 59.8 | 78.7 |
| LogitNorm | 28.9 | 92.5 | 15.8 | 96.3 | 22.4 | 94.4 | 62.4 | 78.5 | 50.4 | 82.1 | 56.4 | 80.3 |
| T2FNorm | 25.8 | 93.0 | 15.2 | 96.5 | 20.5 | 94.8 | 58.2 | 79.9 | 49.5 | 83.3 | 53.8 | 81.6 |
| Haas et al. (2023) | 27.1 | 92.5 | 19.4 | 94.7 | 23.2 | 93.6 | 60.8 | 78.5 | 62.6 | 77.5 | 61.7 | 78.0 |
| **ASL (ours)** | **25.1** | **93.3** | **12.0** | **97.2** | **18.6** | **95.3** | **56.2** | **80.9** | **47.8** | **85.0** | **52.0** | **83.0** |

dence, rather than directly shaping the feature geometry. **T2FNorm** does normalize features, but its motivation is more general—to improve logit separation at test time. While Haas et al. (2023) also explored feature normalization, they focused on presenting a direct improvement relative to Mukhoti et al. (2021), with the goal of accelerating neural collapse. While these methods use feature normalization as a beneficial regularizer, ASL elevates it to be the core mechanism of its training objective, creating a pure, bias-free hyperspherical geometry designed explicitly for angular discrimination.

**Empirical Performance.** These conceptual differences translate directly into tangible performance gains, as detailed in Table 12. We compare ASL against other normalization methods on the CIFAR benchmarks. The results clearly show that ASL consistently outperforms its counterparts.

### E.2 FURTHER DISCUSSION OF ANGULAR SEPARATION LEARNING

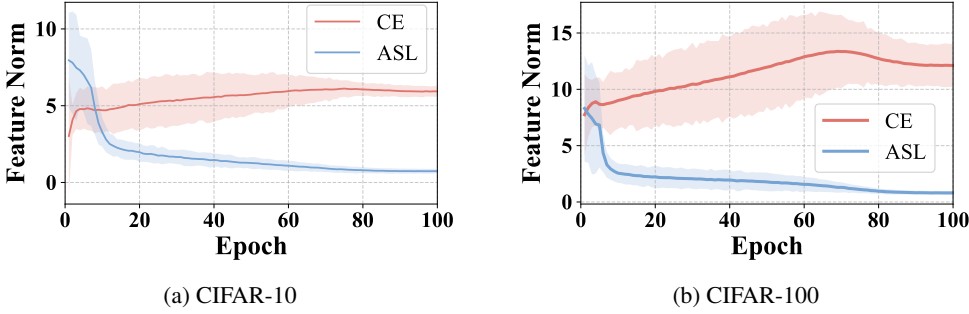

(a) CIFAR-10      (b) CIFAR-100

Figure 18: Evolution of feature norms during training. The line represents the mean feature norm, and the shaded areas depict the variance.

**Analysis of Feature Norm Evolution during Training.** We visualize the evolution of feature norms during training to further demonstrate the effectiveness of ASL. As shown in Fig. 18, standard cross-entropy training tends to continually increase the magnitude of features, which serves as a "lazy" shortcut for classification. This behavior is particularly evident on the more complex CIFAR-100 dataset, where the CE model learns features with both a larger mean norm and significantly higher variance (indicated by the wider shaded area). In stark contrast, ASL rapidly reduces and stabilizes

feature norms to a consistent value early in training. The remarkably narrow variance band for ASL confirms that it produces features with highly consistent magnitudes across samples. This constraint prevents the model from relying on feature scaling and instead forces it to learn more discriminative angular representations, leading to a more robust feature space for OOD detection.

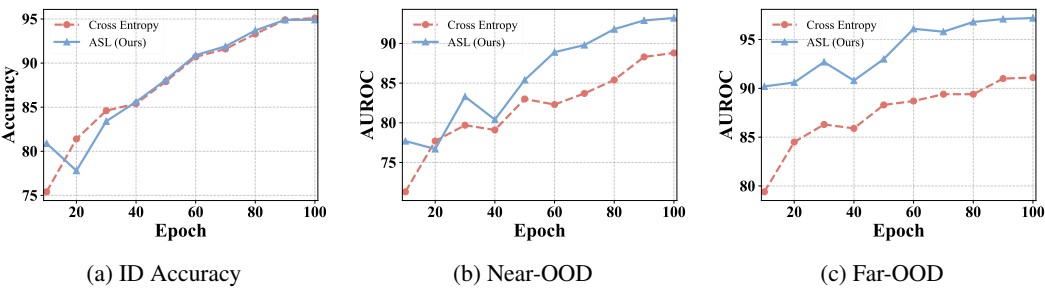

(a) ID Accuracy       (b) Near-OOD       (c) Far-OOD

Figure 19: Training Dynamics Analysis on CIFAR-10.

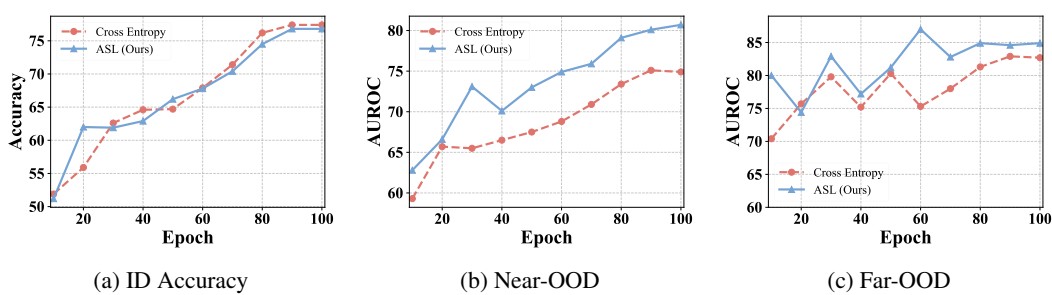

(a) ID Accuracy       (b) Near-OOD       (c) Far-OOD

Figure 20: Training Dynamics Analysis on CIFAR-100.

As illustrated in Fig. 19 and 20, ASL exhibits superior optimization characteristics compared to Cross-Entropy (CE). First, regarding far-OOD, ASL rapidly establishes strong separability, reaching a high performance level at a very early stage of training (e.g., epoch 10). Second, on the more challenging near-OOD, ASL demonstrates a robust optimization trajectory. Despite transient fluctuations in the early training phase, ASL quickly stabilizes and maintains a clear performance advantage over CE in subsequent epochs.

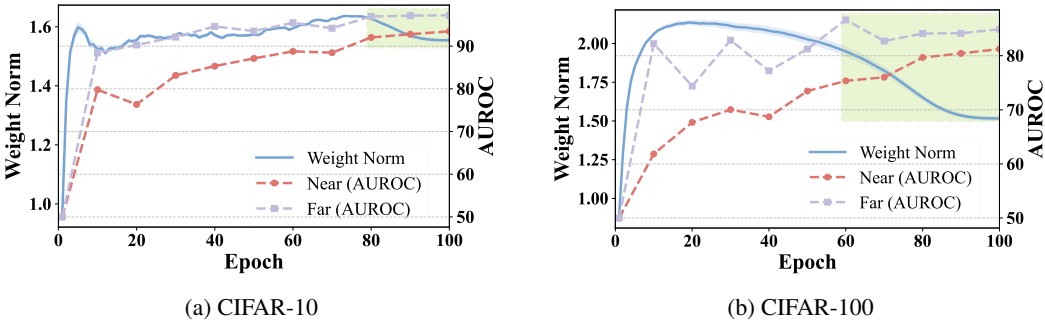

(a) CIFAR-10          (b) CIFAR-100

Figure 21: Evolution of weight norms and AUROC during training.

**Analysis of Temperature.** As established in Proposition 1, $\lambda$ constrains the norm of the classifier weights such that $\|w_j\|_2 \leq s/(2\lambda)$ at convergence. In this formulation, the term $\|w_j\|_2$ functions as an inverse temperature parameter. This inverse temperature has two parts: A fixed component ($s$): A scaling factor, typically set to 10, used to counteract the smaller gradients caused by $\ell_2$-normalization. A dynamic component ($\|w_j\|_2$): The norm of the classifier weights is learned during training and is controlled by the weight decay. By adjusting $\lambda$, we effectively control $\|w_j\|_2$ and thereby tune this temperature. A larger $\lambda$ leads to smaller weight norms, which softens the softmax

distribution. As shown in Fig. 4, this encourages the model to learn finer-grained features for inter-class discrimination—a behavior particularly beneficial for challenging near-OOD detection.

Notably, unlike typical contrastive learning methods that use a fixed temperature, the temperature in ASL is dynamically regulated through the weight norms, as illustrated in Fig. 21. Specifically, larger weight norms in the early stages of training result in a sharper probability distribution (larger inter-class probability gaps), facilitating coarse-grained discrimination and accelerating initial optimization. In later stages (green region), smaller weight norms smooth the probability distribution (smaller inter-class gaps), directing the model's focus toward finer-grained details and thereby improving near-OOD performance. In contrast, contrastive learning methods with a fixed temperature tend to remain in a coarse-grained training regime and fail to transition into a fine-grained phase, limiting their ability to improve near-OOD detection.

Table 13: PALM with Different Weight Decay

|  | Near-OOD | | | | | | Far-OOD | | | | | | | | |
|---|---|---|---|---|---|---|---|---|---|---|---|---|---|---|---|
|  | CIFAR-100 | | TIN | | Avg. | | MNIST | | SVHN | | Texture | | Places365 | | Avg. |
|  | FP↓ | AU↑ | FP↓ | AU↑ | FP↓ | AU↑ | FP↓ | AU↑ | FP↓ | AU↑ | FP↓ | AU↑ | FP↓ | AU↑ | FP↓ AU↑ |
| $1 \times 10^{-4}$ | 33.7 | 89.4 | 27.1 | 91.9 | 30.4 | 90.7 | 16.7 | 95.2 | 6.67 | 98.8 | 19.8 | 95.4 | 31.0 | 92.9 | 18.5 95.6 |
| $5 \times 10^{-4}$ | 73.4 | 79.3 | 59.8 | 84.1 | 66.6 | 81.7 | 14.2 | 94.9 | 2.53 | 99.5 | 67.1 | 83.5 | 46.0 | 88.7 | 32.5 91.7 |

To ensure that ASL's superior performance is not merely an artifact of using a larger weight decay value, we conducted a comparative experiment with PALM. We increased PALM's weight decay from its default value of $1 \times 10^{-4}$ to $5 \times 10^{-4}$, matching the value used for ASL. As shown in Table 13, simply increasing the weight decay for PALM led to a significant degradation in its performance. Specifically, on CIFAR-10, the average near-OOD AUROC dropped from 90.7% to 81.7%, and the far-OOD AUROC decreased from 95.6% to 91.7%. This result demonstrates that ASL's effectiveness stems from its core mechanism—applying cross-entropy loss to normalized features—rather than the specific choice of the weight decay hyperparameter.

Table 14: ID classification accuracy.

| CIFAR-10 | | | | | | | | |
|---|---|---|---|---|---|---|---|---|
| CE | NPOS | CIDER | LogitNorm | PALM | ReweightOOD | T2FNorm | HamOS | ASL |
| **95.2** | 92.1 | 92.8 | 94.5 | 93.5 | 92.9 | 94.7 | 93.8 | 94.8 |
| CIFAR-100 | | | | | | | | |
| CE | NPOS | CIDER | LogitNorm | PALM | ReweightOOD | T2FNorm | HamOS | ASL |
| **77.4** | 72.9 | 68.1 | 76.1 | 75.7 | 71.5 | 76.3 | 74.9 | 76.7 |

## E.3 ID ACCURACY

We list the ID classification accuracy in Table 14. For contrastive learning-based methods, since the linear layer is integrated as part of the feature during training and is not directly used for classification, we follow the approach in Khosla et al. (2020) by adding a second-stage training phase to fine-tune the linear layer separately for classification, using the same training parameters as in the first stage. We observe that ASL results in a slight decrease in classification accuracy compared to standard cross-entropy training. This suggests that the norm information discarded by $\ell_2$-normalization may contribute marginally to the discriminative power of the features learned by cross-entropy. In contrast, ASL focuses solely on angular optimization through $\ell_2$-normalization, which may compromise some discriminative power from the norm space. Nevertheless, ASL still delivers highly competitive performance, indicating that angular separation alone serves as a strong and sufficient supervisory signal for ID classification.

## E.4 FULL EXPERIMENTAL RESULTS

Due to space constraints in the main paper, we present here the full experimental results. This section provides a comprehensive performance comparison of our proposed method, ASL, against

Table 15: Detailed OOD detection performance on CIFAR-10 under standard setting.

| | Near-OOD | | | | | | Far-OOD | | | | | | | | | |
| | CIFAR-100 | | TIN | | Avg. | | MNIST | | SVHN | | Texture | | Places365 | | Avg. | |
| Method | FP↓ | AU↑ | FP↓ | AU↑ | FP↓ | AU↑ | FP↓ | AU↑ | FP↓ | AU↑ | FP↓ | AU↑ | FP↓ | AU↑ | FP↓ | AU↑ |
|---|---|---|---|---|---|---|---|---|---|---|---|---|---|---|---|---|
| *Post-hoc Methods* | | | | | | | | | | | | | | | | |
| MSP | 51.6±0.76 | 87.3±0.09 | 39.9±0.14 | 89.4±0.37 | 45.8±0.31 | 88.3±0.14 | 26.7±4.79 | 91.5±1.43 | 20.0±4.19 | 93.6±1.35 | 25.9±6.88 | 91.8±1.48 | 47.3±1.85 | 88.5±0.09 | 30.0±1.10 | 91.3±0.37 |
| MLS | 66.2±2.20 | 86.5±0.11 | 53.8±1.48 | 89.6±0.52 | 60.0±1.84 | 88.1±0.20 | 30.0±9.63 | 92.8±2.07 | 20.8±9.13 | 94.8±2.04 | 35.3±10.5 | 91.9±1.84 | 55.7±3.02 | 89.3±0.09 | 35.4±1.75 | 92.2±0.48 |
| ReAct | 66.0±4.26 | 85.4±1.27 | 57.9±4.50 | 88.1±0.66 | 62.0±4.38 | 86.8±0.97 | 46.5±19.1 | 90.0±3.59 | 33.7±1.52 | 93.3±0.46 | 44.8±1.81 | 90.2±0.15 | 43.0±6.76 | 90.8±1.38 | 42.0±3.91 | 91.0±0.48 |
| ASH | 90.4±4.26 | 72.2±2.72 | 87.9±2.62 | 75.5±0.98 | 89.2±3.44 | 73.8±1.85 | 71.1±8.66 | 82.7±3.89 | 83.1±4.46 | 70.4±7.80 | 86.1±2.01 | 74.0±3.41 | 67.3±12.2 | 84.7±5.66 | 76.9±0.74 | 77.9±2.36 |
| MDS | 52.7±3.51 | 84.5±2.69 | 45.7±5.30 | 85.8±2.89 | 49.2±4.40 | 85.2±2.79 | 25.2±4.02 | 90.1±1.36 | 30.0±1.19 | 87.8±2.41 | 26.8±4.54 | 93.1±1.21 | 51.1±4.19 | 84.9±2.83 | 33.3±2.89 | 89.0±0.75 |
| +ASL | 29.5±0.32 | 92.0±0.06 | 20.7±0.55 | 94.6±0.00 | 25.1±0.44 | 93.3±0.03 | 2.68±0.58 | 99.5±0.15 | 7.86±0.31 | 98.3±0.08 | 11.4±1.32 | 97.5±0.32 | 26.2±1.33 | 93.6±0.49 | 12.0±0.06 | 97.2±0.02 |
| RMDS | 44.2±1.96 | 88.7±0.13 | 33.0±0.32 | 91.0±0.29 | 38.6±0.82 | 89.8±0.08 | 25.0±2.80 | 91.7±1.09 | 19.3±3.71 | 94.0±1.52 | 21.8±2.63 | 93.4±0.69 | 37.3±0.23 | 90.5±0.05 | 25.9±0.83 | 92.4±0.27 |
| +ASL | 39.6±0.27 | 89.5±0.22 | 27.7±2.49 | 91.8±0.14 | 33.6±1.38 | 90.7±0.18 | 9.58±0.34 | 96.6±0.19 | 15.3±0.19 | 95.0±0.70 | 22.1±0.25 | 93.5±0.22 | 28.5±0.22 | 92.5±0.09 | 18.9±0.03 | 94.4±0.16 |
| VIM | 50.2±2.28 | 87.4±0.24 | 41.0±0.49 | 89.5±0.07 | 45.6±1.38 | 88.4±0.08 | 23.5±4.69 | 92.4±1.95 | 20.4±0.41 | 93.3±1.29 | 20.8±1.25 | 95.2±0.04 | 47.7±2.05 | 88.5±0.10 | 28.1±1.47 | 92.4±0.82 |
| +ASL | 30.1±0.02 | 91.8±0.02 | 22.3±0.05 | 94.3±0.02 | 26.2±0.03 | 93.0±0.00 | 3.61±0.06 | 99.3±0.19 | 8.16±0.38 | 98.0±0.58 | 14.5±0.09 | 96.8±0.03 | 26.1±0.06 | 93.6±0.07 | 13.1±0.12 | 96.9±0.28 |
| KNN | 38.2±0.83 | 89.7±0.08 | 30.5±0.08 | 91.8±0.35 | 34.4±0.46 | 90.7±0.22 | 22.2±2.96 | 93.1±1.14 | 19.1±2.92 | 94.5±1.27 | 21.3±2.14 | 93.8±0.60 | 34.9±0.19 | 91.1±0.21 | 24.4±0.57 | 93.1±0.24 |
| +ASL | 30.0±1.37 | 91.8±0.51 | 23.1±2.32 | 94.0±0.79 | 26.6±1.84 | 92.9±0.65 | 5.26±3.51 | 99.1±1.09 | 11.8±7.42 | 97.5±1.54 | 15.9±1.08 | 96.7±0.18 | 29.5±4.56 | 92.7±1.33 | 15.6±2.39 | 96.5±0.49 |
| fDBD | 40.2±2.34 | 89.4±0.21 | 30.9±0.88 | 91.7±0.16 | 35.5±1.61 | 90.6±0.02 | 22.3±4.11 | 93.6±1.51 | 19.4±1.05 | 94.7±0.64 | 21.0±2.53 | 94.1±0.09 | 34.7±0.69 | 91.2±0.18 | 24.3±0.31 | 93.4±0.00 |
| +ASL | 35.1±0.07 | 91.0±0.11 | 25.5±0.17 | 93.3±0.13 | 30.3±0.05 | 92.1±0.01 | 7.22±0.55 | 98.4±0.63 | 9.21±0.29 | 97.9±0.09 | 14.8±0.30 | 96.6±0.83 | 30.8±0.07 | 92.4±0.03 | 15.5±0.03 | 96.3±0.07 |
| MD++ | 44.2±1.15 | 87.9±0.65 | 36.2±1.12 | 89.7±1.11 | 40.2±0.02 | 88.8±0.88 | 25.8±1.49 | 92.7±0.04 | 24.2±1.63 | 92.1±0.10 | 25.6±3.11 | 92.1±0.19 | 47.0±0.98 | 87.3±0.53 | 30.6±0.57 | 91.0±0.15 |
| +ASL | 29.5±0.32 | 92.0±0.06 | 20.7±0.55 | 94.6±0.00 | 25.1±0.44 | 93.3±0.03 | 2.68±0.58 | 99.5±0.15 | 7.86±0.31 | 98.3±0.08 | 11.4±1.32 | 97.5±0.32 | 26.2±1.33 | 93.6±0.49 | 12.0±0.06 | 97.2±0.02 |
| *Training-based Methods* | | | | | | | | | | | | | | | | |
| NPOS | 35.3±0.20 | 89.0±0.08 | 29.3±2.09 | 91.5±0.66 | 32.3±0.94 | 90.3±0.37 | 26.0±5.71 | 93.3±1.39 | 3.50±1.49 | 99.3±0.14 | 24.0±4.16 | 94.4±0.76 | 31.1±1.72 | 91.3±0.33 | 21.1±2.41 | 94.6±0.49 |
| CIDER | 35.0±0.22 | 89.0±0.02 | 27.8±0.03 | 91.2±0.06 | 31.4±0.09 | 90.1±0.02 | 22.7±0.09 | 93.2±0.08 | 11.8±0.60 | 97.1±0.07 | 25.6±0.13 | 90.8±0.22 | 29.7±0.06 | 92.3±0.03 | 22.5±0.16 | 93.4±0.01 |
| LogitNorm | 33.5±0.25 | 91.1±0.07 | 24.3±0.12 | 93.8±0.07 | 28.9±0.18 | 92.5±0.07 | 1.48±0.05 | 99.7±0.01 | 12.3±0.64 | 96.7±0.21 | 25.0±0.79 | 94.4±0.15 | 24.3±0.02 | 94.2±0.00 | 15.8±0.34 | 96.3±0.09 |
| PALM | 33.7±1.07 | 89.4±0.09 | 27.1±1.74 | 91.9±0.15 | 30.4±1.41 | 90.7±0.12 | 16.7±1.03 | 95.2±0.05 | 6.67±0.98 | 98.8±0.11 | 19.8±0.35 | 95.4±0.37 | 31.0±3.39 | 92.9±0.42 | 18.5±1.26 | 95.6±0.05 |
| ReweightOOD | 36.9±0.31 | 89.5±0.03 | 23.7±0.18 | 94.1±0.02 | 30.3±0.24 | 91.8±0.03 | 8.06±0.19 | 97.7±0.05 | 0.18±0.02 | 99.9±0.00 | 11.6±0.16 | 97.9±0.03 | 20.8±0.48 | 94.9±0.08 | 10.2±0.04 | 97.6±0.00 |
| T2FNorm | 30.0±0.58 | 91.9±0.09 | 21.6±0.59 | 94.2±0.04 | 25.8±0.59 | 93.0±0.07 | 5.79±0.29 | 98.8±0.03 | 9.19±0.66 | 98.2±0.34 | 19.5±0.93 | 95.5±0.18 | 26.3±2.11 | 93.6±0.65 | 15.2±0.20 | 96.5±0.02 |
| HamOS | 33.0±0.21 | 89.9±0.02 | 24.0±0.77 | 93.0±0.11 | 28.5±0.28 | 91.5±0.06 | 14.2±2.11 | 95.8±0.05 | 3.30±0.14 | 99.3±0.22 | 21.4±0.64 | 94.5±0.08 | 25.8±0.62 | 92.7±0.80 | 16.2±0.92 | 95.6±0.22 |
| **ASL (Ours)** | 29.5±0.32 | 92.0±0.06 | 20.7±0.55 | 94.6±0.00 | **25.1±0.44** | **93.3±0.03** | 2.68±0.58 | 99.5±0.15 | 7.86±0.31 | 98.3±0.08 | 11.4±1.32 | 97.5±0.32 | 26.2±1.33 | 93.6±0.49 | 12.0±0.06 | 97.2±0.02 |

Table 16: Detailed OOD detection performance on CIFAR-100 under standard setting.

| | Near-OOD | | | | | | Far-OOD | | | | | | | | | |
| | CIFAR-100 | | TIN | | Avg. | | MNIST | | SVHN | | Texture | | Places365 | | Avg. | |
| Method | FP↓ | AU↑ | FP↓ | AU↑ | FP↓ | AU↑ | FP↓ | AU↑ | FP↓ | AU↑ | FP↓ | AU↑ | FP↓ | AU↑ | FP↓ | AU↑ |
|---|---|---|---|---|---|---|---|---|---|---|---|---|---|---|---|---|
| *Post-hoc Methods* | | | | | | | | | | | | | | | | |
| MSP | 60.6±2.08 | 78.2±0.22 | 51.1±0.11 | 81.8±0.06 | 55.9±1.10 | 80.0±0.14 | 55.1±2.28 | 77.6±0.26 | 61.2±2.26 | 77.1±1.15 | 61.0±0.36 | 77.7±0.13 | 56.8±0.37 | 79.4±0.34 | 58.5±1.08 | 78.1±0.24 |
| MLS | 61.6±2.24 | 78.7±0.45 | 53.1±1.01 | 82.4±0.33 | 57.4±1.62 | 80.5±0.39 | 50.2±1.14 | 80.2±0.12 | 54.0±2.00 | 81.3±1.59 | 62.3±0.52 | 78.6±0.08 | 56.9±1.11 | 79.8±0.23 | 55.9±0.64 | 80.0±0.35 |
| ReAct | 64.2±2.49 | 78.0±0.51 | 53.3±1.58 | 82.4±0.29 | 58.8±2.03 | 80.2±0.40 | 52.1±1.80 | 80.0±0.06 | 49.9±0.13 | 82.5±1.50 | 55.9±0.64 | 80.1±0.16 | 56.4±0.87 | 79.8±0.07 | 53.6±0.86 | 80.6±0.36 |
| ASH | 67.3±3.51 | 77.1±0.65 | 63.1±0.71 | 79.8±0.21 | 65.2±0.09 | 78.4±0.22 | 60.6±1.25 | 78.5±0.83 | 46.0±2.38 | 82.9±2.48 | 66.8±4.48 | 78.6±1.35 | 68.1±4.56 | 77.8±0.75 | 60.4±2.54 | 79.5±0.94 |
| MDS | 88.2±0.35 | 55.9±0.17 | 77.8±0.42 | 62.9±0.83 | 83.0±0.39 | 59.4±0.50 | 72.9±1.01 | 64.3±2.86 | 69.9±4.21 | 69.0±5.34 | 71.8±3.11 | 76.0±0.71 | 79.5±0.07 | 63.7±0.86 | 73.5±0.54 | 68.3±0.66 |
| +ASL | 63.7±0.26 | 77.6±0.42 | 48.6±0.79 | 84.2±0.07 | 56.2±0.53 | 80.9±0.24 | 38.9±1.74 | 86.7±0.07 | 46.8±1.30 | 87.5±0.25 | 47.8±0.24 | 85.0±0.41 | 57.8±0.35 | 81.0±0.15 | 47.8±0.14 | 85.0±0.09 |
| RMDS | 63.7±1.91 | 77.7±0.15 | 49.3±0.41 | 82.7±0.11 | 56.5±0.75 | 80.2±0.13 | 48.3±0.60 | 81.6±0.25 | 60.0±3.17 | 82.5±0.79 | 53.1±0.12 | 84.3±0.56 | 53.4±0.38 | 83.7±0.67 | 53.7±0.82 | 83.0±0.18 |
| +ASL | 64.8±0.08 | 75.6±0.15 | 54.6±0.25 | 79.7±0.05 | 59.7±0.09 | 77.7±0.05 | 41.5±0.08 | 81.7±0.37 | 44.3±0.13 | 84.5±0.24 | 57.5±0.28 | 80.2±0.21 | 58.9±0.10 | 79.9±0.02 | 50.6±0.04 | 81.6±0.10 |
| VIM | 72.5±1.47 | 71.6±0.39 | 55.0±0.22 | 78.3±0.30 | 63.8±1.06 | 75.0±0.05 | 52.6±2.78 | 77.6±3.93 | 48.3±3.82 | 81.5±2.53 | 46.9±2.06 | 85.8±0.56 | 61.9±0.15 | 76.1±0.56 | 52.4±0.22 | 80.3±0.35 |
| +ASL | 62.1±0.08 | 77.7±0.17 | 47.1±0.20 | 84.7±0.15 | 54.6±0.06 | 81.2±0.16 | 31.0±0.18 | 88.4±0.10 | 42.0±0.15 | 87.3±0.20 | 43.9±0.19 | 86.2±0.15 | 56.2±0.17 | 81.2±0.10 | 43.2±0.09 | 85.8±0.04 |
| KNN | 75.4±2.14 | 76.4±0.24 | 50.4±0.07 | 83.2±0.05 | 62.9±1.33 | 79.8±0.10 | 47.6±2.41 | 82.8±0.61 | 57.6±1.11 | 82.4±0.27 | 54.7±1.95 | 84.0±0.11 | 61.3±0.11 | 79.5±0.53 | 55.3±1.34 | 82.2±0.06 |
| +ASL | 65.9±0.10 | 77.2±0.09 | 50.1±0.05 | 83.3±0.04 | 58.0±0.07 | 80.3±0.07 | 44.9±0.14 | 83.0±0.14 | 49.8±0.20 | 84.1±0.06 | 50.5±0.16 | 83.0±0.02 | 61.1±0.09 | 79.0±0.19 | 51.6±0.10 | 82.3±0.01 |
| fDBD | 66.1±2.92 | 77.9±0.40 | 48.5±0.53 | 83.6±0.11 | 57.3±1.73 | 80.8±0.26 | 49.7±2.72 | 80.7±0.71 | 60.1±2.35 | 78.9±0.83 | 55.2±2.04 | 81.1±0.31 | 57.1±0.22 | 79.7±0.01 | 55.5±1.72 | 80.1±0.46 |
| +ASL | 64.6±0.09 | 76.9±0.12 | 47.5±0.10 | 84.2±0.15 | 56.0±0.09 | 80.5±0.13 | 32.7±0.18 | 89.4±0.06 | 43.1±0.05 | 87.2±0.11 | 49.1±0.08 | 83.1±0.04 | 56.2±0.16 | 80.6±0.17 | 45.3±0.08 | 85.1±0.01 |
| MD++ | 76.6±0.64 | 71.8±0.09 | 58.9±0.93 | 77.9±0.52 | 67.8±0.79 | 74.8±0.30 | 51.4±0.85 | 79.5±1.71 | 46.0±5.23 | 85.8±2.19 | 45.6±0.46 | 87.5±0.19 | 64.5±0.43 | 77.7±1.41 | 51.9±1.51 | 82.6±0.42 |
| +ASL | 63.7±0.26 | 77.6±0.42 | 48.6±0.79 | 84.2±0.07 | 56.2±0.53 | 80.9±0.24 | 38.9±1.74 | 86.7±0.07 | 46.8±1.30 | 87.5±0.25 | 47.8±0.24 | 85.0±0.41 | 57.8±0.35 | 81.0±0.15 | 47.8±0.14 | 85.0±0.09 |
| *Training-based Methods* | | | | | | | | | | | | | | | | |
| NPOS | 76.8±3.63 | 74.5±0.75 | 57.4±2.41 | 81.0±0.57 | 67.1±3.02 | 77.7±0.66 | 68.4±2.29 | 75.9±1.35 | 29.7±0.30 | 92.5±0.15 | 43.7±0.78 | 87.4±0.29 | 58.8±0.27 | 78.5±0.17 | 50.2±0.78 | 83.6±0.33 |
| CIDER | 77.2±0.13 | 70.4±0.12 | 59.5±0.07 | 78.4±0.14 | 68.4±0.03 | 74.4±0.13 | 77.8±0.18 | 60.2±0.04 | 40.4±0.05 | 92.8±0.08 | 60.3±0.10 | 78.5±0.14 | 67.5±0.19 | 76.2±0.09 | 61.5±0.10 | 76.9±0.05 |
| LogitNorm | 73.0±0.13 | 74.5±0.09 | 51.8±0.03 | 82.6±0.01 | 62.4±0.05 | 78.5±0.04 | 26.8±0.20 | 91.7±0.12 | 41.9±0.18 | 87.3±0.20 | 78.2±0.17 | 71.9±0.14 | 54.7±0.09 | 81.2±0.07 | 50.4±0.03 | 82.1±0.07 |
| PALM | 76.7±0.16 | 71.0±0.06 | 52.6±0.07 | 83.5±0.01 | 64.7±0.11 | 77.3±0.04 | 54.7±0.49 | 76.1±0.21 | 3.18±0.09 | 99.3±0.09 | 40.1±0.02 | 89.5±0.10 | 53.1±0.07 | 82.0±0.17 | **37.8±0.11** | **86.7±0.10** |
| ReweightOOD | 73.7±0.20 | 71.1±0.08 | 57.9±0.15 | 81.8±0.04 | 65.8±0.17 | 76.5±0.06 | 67.2±0.14 | 77.9±0.01 | 11.5±0.22 | 98.0±0.11 | 34.2±0.13 | 92.4±0.06 | 55.0±0.08 | 84.2±0.09 | 42.0±0.10 | 88.1±0.02 |
| T2FNorm | 68.0±0.15 | 76.0±0.01 | 48.4±0.19 | 83.9±0.07 | 58.2±0.02 | 79.9±0.03 | 29.0±0.28 | 90.8±0.05 | 46.3±0.05 | 84.2±0.07 | 68.4±0.06 | 77.1±0.04 | 54.1±0.03 | 81.1±0.09 | 49.5±0.09 | 83.3±0.06 |
| HamOS | 71.8±0.45 | 76.6±0.06 | 53.6±0.26 | 81.7±0.34 | 62.7±0.09 | 79.1±0.20 | 75.4±1.02 | 69.6±0.31 | 35.5±0.42 | 90.1±0.07 | 50.6±0.25 | 82.2±0.19 | 55.3±0.11 | 80.5±0.14 | 54.2±0.45 | 80.6±0.18 |
| **ASL (Ours)** | 63.7±0.26 | 77.6±0.42 | 48.6±0.79 | 84.2±0.07 | **56.2±0.53** | **80.9±0.24** | 38.9±1.74 | 86.7±0.07 | 46.8±1.30 | 87.5±0.25 | 47.8±0.24 | 85.0±0.41 | 57.8±0.35 | 81.0±0.15 | 47.8±0.14 | 85.0±0.09 |

Table 17: Detailed OOD detection performance on ImageNet-200 under standard setting.

| | Near-OOD | | | | | | Far-OOD | | | | | | | |
| | SSB-hard | | NINCO | | Avg. | | iNaturalist | | Textures | | OpenImage-O | | Avg. | |
| Method | FP↓ | AU↑ | FP↓ | AU↑ | FP↓ | AU↑ | FP↓ | AU↑ | FP↓ | AU↑ | FP↓ | AU↑ | FP↓ | AU↑ |
|---|---|---|---|---|---|---|---|---|---|---|---|---|---|---|
| *Post-hoc Methods* | | | | | | | | | | | | | | |
| MSP | $65.9_{\pm0.11}$ | $80.4_{\pm0.04}$ | $44.2_{\pm0.89}$ | $86.2_{\pm0.13}$ | $55.0_{\pm0.50}$ | $83.3_{\pm0.04}$ | $26.1_{\pm0.20}$ | $93.0_{\pm0.05}$ | $45.9_{\pm1.73}$ | $88.2_{\pm0.34}$ | $35.3_{\pm0.42}$ | $89.1_{\pm0.16}$ | $35.8_{\pm0.78}$ | $90.1_{\pm0.15}$ |
| MLS | $69.4_{\pm0.55}$ | $80.1_{\pm0.07}$ | $48.4_{\pm1.29}$ | $85.8_{\pm0.16}$ | $58.9_{\pm0.92}$ | $83.0_{\pm0.11}$ | $23.8_{\pm0.32}$ | $93.5_{\pm0.31}$ | $43.3_{\pm3.31}$ | $90.4_{\pm0.31}$ | $35.8_{\pm0.30}$ | $89.6_{\pm0.12}$ | $34.3_{\pm1.03}$ | $91.2_{\pm0.04}$ |
| ReAct | $73.9_{\pm1.63}$ | $77.0_{\pm1.87}$ | $56.6_{\pm2.76}$ | $83.8_{\pm0.84}$ | $65.2_{\pm2.19}$ | $80.4_{\pm1.36}$ | $20.6_{\pm1.99}$ | $94.7_{\pm1.10}$ | $28.4_{\pm1.41}$ | $93.5_{\pm0.72}$ | $34.0_{\pm1.20}$ | $90.7_{\pm0.48}$ | $27.7_{\pm0.73}$ | $93.0_{\pm0.77}$ |
| ASH | $73.7_{\pm1.24}$ | $79.0_{\pm0.46}$ | $57.3_{\pm1.17}$ | $85.3_{\pm0.12}$ | $65.5_{\pm0.04}$ | $82.1_{\pm0.17}$ | $21.4_{\pm0.05}$ | $95.4_{\pm0.00}$ | $24.2_{\pm1.37}$ | $95.0_{\pm0.32}$ | $33.8_{\pm0.69}$ | $92.1_{\pm0.25}$ | $26.4_{\pm0.21}$ | $94.2_{\pm0.19}$ |
| MDS | $85.6_{\pm0.89}$ | $57.4_{\pm0.02}$ | $76.7_{\pm0.53}$ | $64.6_{\pm0.54}$ | $81.1_{\pm0.71}$ | $61.0_{\pm0.28}$ | $62.3_{\pm0.62}$ | $73.1_{\pm0.91}$ | $60.8_{\pm1.26}$ | $78.5_{\pm0.90}$ | $70.0_{\pm0.53}$ | $69.0_{\pm0.25}$ | $64.4_{\pm0.80}$ | $73.5_{\pm0.69}$ |
| +ASL | $65.7_{\pm0.09}$ | $81.5_{\pm0.05}$ | $39.1_{\pm0.09}$ | $88.6_{\pm0.07}$ | $52.4_{\pm0.84}$ | $85.1_{\pm0.06}$ | $15.2_{\pm0.13}$ | $96.5_{\pm0.06}$ | $21.6_{\pm0.26}$ | $95.4_{\pm0.43}$ | $24.4_{\pm0.43}$ | $93.2_{\pm0.36}$ | $20.4_{\pm0.19}$ | $95.0_{\pm0.25}$ |
| RMDS | $65.6_{\pm0.47}$ | $79.6_{\pm0.10}$ | $43.7_{\pm1.42}$ | $84.2_{\pm0.28}$ | $54.6_{\pm0.48}$ | $81.9_{\pm0.19}$ | $25.0_{\pm0.02}$ | $89.5_{\pm0.55}$ | $37.0_{\pm0.04}$ | $86.3_{\pm0.54}$ | $34.5_{\pm0.39}$ | $86.4_{\pm0.23}$ | $32.1_{\pm0.15}$ | $87.4_{\pm0.44}$ |
| +ASL | $64.0_{\pm0.18}$ | $81.5_{\pm0.22}$ | $42.1_{\pm0.10}$ | $86.3_{\pm0.16}$ | $53.0_{\pm0.14}$ | $83.9_{\pm0.19}$ | $27.8_{\pm0.16}$ | $90.5_{\pm0.10}$ | $37.3_{\pm0.18}$ | $87.8_{\pm0.10}$ | $34.2_{\pm0.14}$ | $88.2_{\pm0.27}$ | $33.1_{\pm0.16}$ | $88.8_{\pm0.15}$ |
| VIM | $72.2_{\pm0.90}$ | $73.8_{\pm0.24}$ | $48.4_{\pm1.96}$ | $83.4_{\pm0.12}$ | $60.3_{\pm1.43}$ | $78.6_{\pm0.06}$ | $27.1_{\pm0.35}$ | $91.2_{\pm0.44}$ | $20.1_{\pm0.16}$ | $94.7_{\pm0.23}$ | $34.2_{\pm0.65}$ | $88.2_{\pm0.01}$ | $27.1_{\pm0.05}$ | $91.4_{\pm0.22}$ |
| +ASL | $67.7_{\pm0.32}$ | $79.8_{\pm0.15}$ | $42.9_{\pm0.49}$ | $87.7_{\pm0.32}$ | $55.3_{\pm0.41}$ | $83.8_{\pm0.24}$ | $15.0_{\pm0.18}$ | $96.3_{\pm0.25}$ | $19.1_{\pm0.07}$ | $96.0_{\pm0.01}$ | $25.1_{\pm0.06}$ | $93.2_{\pm0.40}$ | $19.7_{\pm0.02}$ | $95.2_{\pm0.22}$ |
| KNN | $73.7_{\pm0.16}$ | $76.9_{\pm0.04}$ | $47.3_{\pm1.30}$ | $86.1_{\pm0.13}$ | $60.5_{\pm0.57}$ | $81.5_{\pm0.08}$ | $24.2_{\pm0.32}$ | $94.1_{\pm0.29}$ | $23.4_{\pm0.69}$ | $95.3_{\pm0.03}$ | $32.8_{\pm1.18}$ | $90.5_{\pm0.16}$ | $26.8_{\pm0.05}$ | $93.3_{\pm0.16}$ |
| +ASL | $71.0_{\pm0.08}$ | $79.3_{\pm0.10}$ | $41.0_{\pm0.05}$ | $87.2_{\pm0.17}$ | $56.0_{\pm0.07}$ | $83.3_{\pm0.14}$ | $19.7_{\pm0.11}$ | $94.9_{\pm0.19}$ | $29.9_{\pm0.09}$ | $92.8_{\pm0.06}$ | $27.8_{\pm0.05}$ | $91.8_{\pm0.06}$ | $25.8_{\pm0.08}$ | $93.2_{\pm0.06}$ |
| fDBD | $66.1_{\pm0.54}$ | $80.6_{\pm0.05}$ | $39.7_{\pm0.58}$ | $88.0_{\pm0.07}$ | $52.9_{\pm0.02}$ | $84.3_{\pm0.01}$ | $16.3_{\pm0.03}$ | $95.8_{\pm0.17}$ | $28.8_{\pm0.29}$ | $93.0_{\pm0.01}$ | $27.4_{\pm0.42}$ | $91.8_{\pm0.08}$ | $24.2_{\pm0.23}$ | $93.5_{\pm0.08}$ |
| +ASL | $72.8_{\pm0.19}$ | $79.4_{\pm0.07}$ | $44.1_{\pm0.07}$ | $87.2_{\pm0.06}$ | $58.4_{\pm0.13}$ | $83.3_{\pm0.07}$ | $14.3_{\pm0.24}$ | $96.6_{\pm0.08}$ | $22.7_{\pm0.09}$ | $94.3_{\pm0.11}$ | $25.2_{\pm0.15}$ | $92.6_{\pm0.11}$ | $20.8_{\pm0.16}$ | $94.5_{\pm0.10}$ |
| MD++ | $72.5_{\pm1.48}$ | $78.8_{\pm0.07}$ | $46.1_{\pm0.47}$ | $86.9_{\pm0.54}$ | $59.3_{\pm0.51}$ | $82.8_{\pm0.21}$ | $19.1_{\pm0.07}$ | $95.9_{\pm0.24}$ | $24.3_{\pm0.21}$ | $94.6_{\pm1.69}$ | $29.8_{\pm0.21}$ | $91.4_{\pm0.06}$ | $24.4_{\pm0.17}$ | $94.0_{\pm0.50}$ |
| +ASL | $65.7_{\pm0.09}$ | $81.5_{\pm0.05}$ | $39.1_{\pm0.09}$ | $88.6_{\pm0.07}$ | $52.4_{\pm0.84}$ | $85.1_{\pm0.06}$ | $15.2_{\pm0.13}$ | $96.5_{\pm0.06}$ | $21.6_{\pm0.26}$ | $95.4_{\pm0.43}$ | $24.4_{\pm0.43}$ | $93.2_{\pm0.36}$ | $20.4_{\pm0.19}$ | $95.0_{\pm0.25}$ |
| *Training-based Methods* | | | | | | | | | | | | | | |
| CIDER | $71.0_{\pm1.41}$ | $75.5_{\pm1.06}$ | $45.3_{\pm0.52}$ | $85.3_{\pm0.18}$ | $58.1_{\pm0.97}$ | $80.4_{\pm0.62}$ | $20.0_{\pm0.50}$ | $94.4_{\pm0.31}$ | $16.8_{\pm0.14}$ | $97.1_{\pm0.16}$ | $27.7_{\pm0.22}$ | $91.6_{\pm0.41}$ | $21.5_{\pm0.29}$ | $94.4_{\pm0.30}$ |
| LogitNorm | $66.7_{\pm0.17}$ | $78.4_{\pm0.15}$ | $46.6_{\pm0.26}$ | $86.4_{\pm0.07}$ | $56.7_{\pm0.22}$ | $82.4_{\pm0.11}$ | $15.7_{\pm0.36}$ | $96.1_{\pm0.12}$ | $31.7_{\pm0.10}$ | $92.3_{\pm0.09}$ | $31.0_{\pm0.13}$ | $90.9_{\pm0.07}$ | $26.1_{\pm0.13}$ | $93.1_{\pm0.10}$ |
| PALM | $70.1_{\pm0.41}$ | $75.7_{\pm0.03}$ | $50.3_{\pm0.52}$ | $84.8_{\pm0.26}$ | $60.2_{\pm0.46}$ | $80.2_{\pm0.14}$ | $21.9_{\pm0.15}$ | $94.3_{\pm0.10}$ | $12.7_{\pm0.16}$ | $97.8_{\pm0.03}$ | $29.3_{\pm0.20}$ | $91.3_{\pm0.13}$ | $21.3_{\pm0.17}$ | $94.5_{\pm0.08}$ |
| ReweightOOD | $67.2_{\pm0.29}$ | $80.5_{\pm0.28}$ | $41.8_{\pm0.26}$ | $86.8_{\pm0.15}$ | $54.5_{\pm0.27}$ | $83.7_{\pm0.21}$ | $18.2_{\pm0.14}$ | $95.0_{\pm0.11}$ | $19.5_{\pm0.20}$ | $96.4_{\pm0.15}$ | $27.8_{\pm0.45}$ | $91.9_{\pm0.28}$ | $21.9_{\pm0.13}$ | $94.5_{\pm0.18}$ |
| T2FNorm | $66.1_{\pm0.20}$ | $79.2_{\pm0.12}$ | $46.2_{\pm0.12}$ | $86.8_{\pm0.24}$ | $56.1_{\pm0.04}$ | $83.0_{\pm0.18}$ | $13.7_{\pm0.18}$ | $96.8_{\pm0.09}$ | $33.7_{\pm0.12}$ | $91.8_{\pm0.10}$ | $28.5_{\pm0.15}$ | $91.8_{\pm0.06}$ | $25.3_{\pm0.15}$ | $93.5_{\pm0.08}$ |
| **ASL (Ours)** | $65.7_{\pm0.09}$ | $81.5_{\pm0.05}$ | $39.1_{\pm0.09}$ | $88.6_{\pm0.07}$ | $52.4_{\pm0.84}$ | $85.1_{\pm0.06}$ | $15.2_{\pm0.13}$ | $96.5_{\pm0.06}$ | $21.6_{\pm0.26}$ | $95.4_{\pm0.43}$ | $24.4_{\pm0.43}$ | $93.2_{\pm0.36}$ | $20.4_{\pm0.19}$ | $95.0_{\pm0.25}$ |

Table 18: OOD detection performance on ImageNet-1K with ResNet-50 under standard setting.

| | Near-OOD | | | | | | Far-OOD | | | | | | | |
| | SSB-hard | | NINCO | | Avg. | | iNaturalist | | Textures | | OpenImage-O | | Avg. | |
| Method | FP↓ | AU↑ | FP↓ | AU↑ | FP↓ | AU↑ | FP↓ | AU↑ | FP↓ | AU↑ | FP↓ | AU↑ | FP↓ | AU↑ |
|---|---|---|---|---|---|---|---|---|---|---|---|---|---|---|
| *Post-hoc Methods* | | | | | | | | | | | | | | |
| MSP | $74.6_{\pm0.00}$ | $72.2_{\pm0.00}$ | $56.8_{\pm0.00}$ | $80.0_{\pm0.00}$ | $65.7_{\pm0.00}$ | $76.1_{\pm0.00}$ | $43.4_{\pm0.00}$ | $88.4_{\pm0.00}$ | $60.9_{\pm0.00}$ | $82.5_{\pm0.00}$ | $50.0_{\pm0.00}$ | $85.0_{\pm0.00}$ | $51.4_{\pm0.00}$ | $85.3_{\pm0.00}$ |
| MLS | $76.2_{\pm0.00}$ | $72.8_{\pm0.00}$ | $59.2_{\pm0.00}$ | $80.4_{\pm0.00}$ | $67.7_{\pm0.00}$ | $76.6_{\pm0.00}$ | $30.6_{\pm0.00}$ | $91.2_{\pm0.00}$ | $46.0_{\pm0.00}$ | $88.4_{\pm0.00}$ | $37.7_{\pm0.00}$ | $89.3_{\pm0.00}$ | $38.1_{\pm0.00}$ | $89.6_{\pm0.00}$ |
| ReAct | $78.0_{\pm0.00}$ | $73.1_{\pm0.00}$ | $55.8_{\pm0.00}$ | $81.7_{\pm0.00}$ | $66.9_{\pm0.00}$ | $77.4_{\pm0.00}$ | $16.8_{\pm0.00}$ | $96.3_{\pm0.00}$ | $29.7_{\pm0.00}$ | $92.8_{\pm0.00}$ | $32.7_{\pm0.00}$ | $91.9_{\pm0.00}$ | $26.4_{\pm0.00}$ | $93.7_{\pm0.00}$ |
| ASH | $73.7_{\pm0.00}$ | $73.1_{\pm0.00}$ | $53.0_{\pm0.00}$ | $83.4_{\pm0.00}$ | $63.3_{\pm0.00}$ | $78.2_{\pm0.00}$ | $14.3_{\pm0.00}$ | $97.0_{\pm0.00}$ | $15.4_{\pm0.00}$ | $96.9_{\pm0.00}$ | $29.2_{\pm0.00}$ | $93.3_{\pm0.00}$ | $19.6_{\pm0.00}$ | $95.8_{\pm0.00}$ |
| MDS | $92.8_{\pm0.00}$ | $47.0_{\pm0.00}$ | $79.0_{\pm0.00}$ | $62.1_{\pm0.00}$ | $85.9_{\pm0.00}$ | $54.6_{\pm0.00}$ | $73.8_{\pm0.00}$ | $63.7_{\pm0.00}$ | $43.0_{\pm0.00}$ | $89.7_{\pm0.00}$ | $72.6_{\pm0.00}$ | $68.7_{\pm0.00}$ | $63.1_{\pm0.00}$ | $74.0_{\pm0.00}$ |
| +ASL | $81.7_{\pm0.56}$ | $72.2_{\pm0.75}$ | $49.1_{\pm0.55}$ | $85.4_{\pm0.00}$ | $65.4_{\pm0.56}$ | $78.8_{\pm0.37}$ | $21.4_{\pm0.32}$ | $95.1_{\pm0.31}$ | $34.3_{\pm0.20}$ | $92.3_{\pm0.24}$ | $29.3_{\pm0.20}$ | $92.3_{\pm0.23}$ | $28.3_{\pm0.24}$ | $93.3_{\pm0.26}$ |
| RMDS | $78.8_{\pm0.00}$ | $70.8_{\pm0.00}$ | $52.2_{\pm0.00}$ | $82.0_{\pm0.00}$ | $65.5_{\pm0.00}$ | $76.4_{\pm0.00}$ | $34.4_{\pm0.00}$ | $87.1_{\pm0.00}$ | $47.0_{\pm0.00}$ | $86.5_{\pm0.00}$ | $41.1_{\pm0.00}$ | $85.6_{\pm0.00}$ | $40.9_{\pm0.00}$ | $86.4_{\pm0.00}$ |
| +ASL | $79.5_{\pm0.88}$ | $74.5_{\pm0.28}$ | $49.5_{\pm0.90}$ | $85.4_{\pm0.12}$ | $64.5_{\pm0.89}$ | $80.0_{\pm0.20}$ | $29.3_{\pm0.57}$ | $93.7_{\pm0.24}$ | $45.1_{\pm0.28}$ | $88.2_{\pm0.13}$ | $36.2_{\pm0.16}$ | $90.4_{\pm0.12}$ | $36.9_{\pm0.34}$ | $90.8_{\pm0.16}$ |
| VIM | $81.0_{\pm0.00}$ | $65.1_{\pm0.00}$ | $62.7_{\pm0.00}$ | $78.5_{\pm0.00}$ | $71.8_{\pm0.00}$ | $71.8_{\pm0.00}$ | $30.5_{\pm0.00}$ | $89.6_{\pm0.00}$ | $10.6_{\pm0.00}$ | $98.0_{\pm0.00}$ | $33.0_{\pm0.00}$ | $90.4_{\pm0.00}$ | $24.7_{\pm0.00}$ | $92.7_{\pm0.00}$ |
| +ASL | $92.2_{\pm0.99}$ | $64.2_{\pm0.42}$ | $76.5_{\pm1.07}$ | $79.5_{\pm0.11}$ | $84.4_{\pm1.03}$ | $71.8_{\pm0.27}$ | $21.0_{\pm0.57}$ | $94.5_{\pm0.39}$ | $32.8_{\pm0.29}$ | $93.0_{\pm0.18}$ | $32.7_{\pm0.63}$ | $91.1_{\pm0.21}$ | $28.8_{\pm0.50}$ | $92.8_{\pm0.26}$ |
| KNN | $86.2_{\pm0.00}$ | $59.1_{\pm0.00}$ | $62.9_{\pm0.00}$ | $77.5_{\pm0.00}$ | $74.5_{\pm0.00}$ | $68.3_{\pm0.00}$ | $40.9_{\pm0.00}$ | $87.1_{\pm0.00}$ | $14.5_{\pm0.00}$ | $97.5_{\pm0.00}$ | $46.4_{\pm0.00}$ | $86.7_{\pm0.00}$ | $33.9_{\pm0.00}$ | $90.4_{\pm0.00}$ |
| +ASL | $81.3_{\pm0.49}$ | $69.9_{\pm0.08}$ | $55.1_{\pm0.27}$ | $82.6_{\pm0.10}$ | $68.2_{\pm0.38}$ | $76.2_{\pm0.09}$ | $33.5_{\pm0.46}$ | $91.5_{\pm0.13}$ | $39.9_{\pm0.28}$ | $89.6_{\pm0.12}$ | $36.7_{\pm0.47}$ | $89.8_{\pm0.14}$ | $36.7_{\pm0.40}$ | $90.3_{\pm0.13}$ |
| fDBD | $78.3_{\pm0.00}$ | $70.2_{\pm0.00}$ | $52.1_{\pm0.00}$ | $82.6_{\pm0.00}$ | $65.2_{\pm0.00}$ | $76.4_{\pm0.00}$ | $22.1_{\pm0.00}$ | $93.7_{\pm0.00}$ | $27.0_{\pm0.00}$ | $93.4_{\pm0.00}$ | $30.2_{\pm0.00}$ | $91.2_{\pm0.00}$ | $26.7_{\pm0.00}$ | $92.8_{\pm0.00}$ |
| +ASL | $81.4_{\pm0.85}$ | $69.8_{\pm0.11}$ | $53.5_{\pm0.66}$ | $82.7_{\pm0.39}$ | $67.5_{\pm0.75}$ | $76.2_{\pm0.25}$ | $21.3_{\pm0.62}$ | $94.7_{\pm0.18}$ | $40.0_{\pm0.54}$ | $88.7_{\pm0.23}$ | $32.1_{\pm0.80}$ | $90.8_{\pm0.16}$ | $31.1_{\pm0.65}$ | $91.4_{\pm0.20}$ |
| MD++ | $85.1_{\pm0.00}$ | $63.1_{\pm0.00}$ | $61.0_{\pm0.00}$ | $79.8_{\pm0.00}$ | $73.1_{\pm0.00}$ | $71.4_{\pm0.00}$ | $26.3_{\pm0.00}$ | $94.1_{\pm0.00}$ | $15.3_{\pm0.00}$ | $97.2_{\pm0.00}$ | $38.2_{\pm0.00}$ | $90.3_{\pm0.00}$ | $26.6_{\pm0.00}$ | $93.9_{\pm0.00}$ |
| +ASL | $81.7_{\pm0.56}$ | $72.2_{\pm0.75}$ | $49.1_{\pm0.55}$ | $85.4_{\pm0.00}$ | $65.4_{\pm0.56}$ | $78.8_{\pm0.37}$ | $21.4_{\pm0.32}$ | $95.1_{\pm0.31}$ | $34.3_{\pm0.20}$ | $92.3_{\pm0.24}$ | $29.3_{\pm0.20}$ | $92.3_{\pm0.23}$ | $28.3_{\pm0.24}$ | $93.3_{\pm0.26}$ |
| *Training-based Methods* | | | | | | | | | | | | | | |
| CIDER | $88.5_{\pm0.34}$ | $58.1_{\pm0.66}$ | $67.8_{\pm0.54}$ | $75.5_{\pm0.48}$ | $78.1_{\pm0.44}$ | $66.8_{\pm0.57}$ | $26.8_{\pm0.23}$ | $92.8_{\pm0.17}$ | $24.9_{\pm0.36}$ | $95.3_{\pm0.21}$ | $54.0_{\pm0.24}$ | $87.4_{\pm0.33}$ | $35.3_{\pm0.27}$ | $91.8_{\pm0.24}$ |
| LogitNorm | $81.8_{\pm0.44}$ | $67.0_{\pm0.41}$ | $57.3_{\pm0.69}$ | $81.0_{\pm0.35}$ | $69.5_{\pm0.56}$ | $74.0_{\pm0.38}$ | $18.8_{\pm0.54}$ | $95.3_{\pm0.52}$ | $41.0_{\pm0.18}$ | $89.2_{\pm0.23}$ | $32.7_{\pm0.11}$ | $90.8_{\pm0.12}$ | $30.8_{\pm0.28}$ | $91.8_{\pm0.29}$ |
| PALM | $84.1_{\pm0.66}$ | $64.1_{\pm0.34}$ | $53.4_{\pm0.83}$ | $80.8_{\pm0.28}$ | $68.8_{\pm0.74}$ | $72.4_{\pm0.31}$ | $16.5_{\pm0.36}$ | $96.1_{\pm0.11}$ | $16.3_{\pm0.17}$ | $97.1_{\pm0.20}$ | $30.1_{\pm0.10}$ | $92.7_{\pm0.28}$ | $21.0_{\pm0.01}$ | $95.3_{\pm0.21}$ |
| ReweightOOD | $83.7_{\pm0.47}$ | $61.9_{\pm0.19}$ | $57.8_{\pm0.43}$ | $80.6_{\pm0.26}$ | $70.7_{\pm0.45}$ | $71.3_{\pm0.23}$ | $18.7_{\pm0.42}$ | $94.5_{\pm0.16}$ | $8.15_{\pm0.18}$ | $98.4_{\pm0.14}$ | $32.1_{\pm0.38}$ | $92.8_{\pm0.11}$ | **$19.7_{\pm0.33}$** | **$95.2_{\pm0.14}$** |
| T2FNorm | $81.1_{\pm0.57}$ | $69.4_{\pm0.32}$ | $54.9_{\pm0.34}$ | $82.5_{\pm0.27}$ | $68.0_{\pm0.45}$ | $75.9_{\pm0.30}$ | $19.1_{\pm0.15}$ | $95.1_{\pm0.10}$ | $50.5_{\pm0.17}$ | $85.9_{\pm0.21}$ | $34.5_{\pm0.11}$ | $90.1_{\pm0.25}$ | $34.7_{\pm0.14}$ | $90.4_{\pm0.18}$ |
| **ASL (Ours)** | $81.7_{\pm0.56}$ | $72.2_{\pm0.75}$ | $49.1_{\pm0.55}$ | $85.4_{\pm0.00}$ | **$65.4_{\pm0.56}$** | **$78.8_{\pm0.37}$** | $21.4_{\pm0.32}$ | $95.1_{\pm0.31}$ | $34.3_{\pm0.20}$ | $92.3_{\pm0.24}$ | $29.3_{\pm0.20}$ | $92.3_{\pm0.23}$ | $28.3_{\pm0.24}$ | $93.3_{\pm0.26}$ |

various post-hoc and training-based OOD detection methods. The results are detailed across several standard benchmarks: Table 15 presents the outcomes on CIFAR-10, while Table 16 covers CIFAR-100. Performance on the subset ImageNet-200 is reported in Table 17. To demonstrate architectural versatility, we also include results on the full ImageNet-1k dataset using a ResNet-50 backbone in Table 18 and a ViT-B/16 backbone in Table 19.

## E.5 COVARIATE SHIFT EXPERIMENT

In complex scenarios involving covariate shift, ASL has demonstrated advanced performance across multiple benchmark tests, with a particularly significant improvement in its ability to detect semantically similar and hard-to-distinguish near-OOD samples. Specifically, on CIFAR-10 (Table 20), CIFAR-100 (Table 21), and ImageNet-200 (Table 22), ASL significantly enhances the performance of the original cross-entropy method and achieves state-of-the-art results in OOD detection. For

Table 19: OOD detection performance on ImageNet-1K with ViT-B/16 under standard setting.

| | Near-OOD | | | | | | Far-OOD | | | | | | | |
| | SSB-hard | | NINCO | | Avg. | | iNaturalist | | Textures | | OpenImage-O | | Avg. | |
| Method | FP↓ | AU↑ | FP↓ | AU↑ | FP↓ | AU↑ | FP↓ | AU↑ | FP↓ | AU↑ | FP↓ | AU↑ | FP↓ | AU↑ |
|---|---|---|---|---|---|---|---|---|---|---|---|---|---|---|
| *Post-hoc Methods* | | | | | | | | | | | | | | |
| MSP | 86.1±0.00 | 69.0±0.00 | 77.1±0.00 | 78.1±0.00 | 81.6±0.00 | 73.6±0.00 | 42.7±0.00 | 88.2±0.00 | 56.3±0.00 | 85.1±0.00 | 56.3±0.00 | 84.8±0.00 | 51.8±0.00 | 86.0±0.00 |
| MLS | 91.2±0.00 | 64.5±0.00 | 92.9±0.00 | 72.4±0.00 | 92.1±0.00 | 68.4±0.00 | 72.9±0.00 | 85.2±0.00 | 78.4±0.00 | 83.8±0.00 | 85.5±0.00 | 81.6±0.00 | 78.9±0.00 | 83.5±0.00 |
| ReAct | 90.2±0.00 | 63.2±0.00 | 78.5±0.00 | 75.5±0.00 | 84.4±0.00 | 69.3±0.00 | 48.3±0.00 | 86.1±0.00 | 55.9±0.00 | 86.7±0.00 | 57.6±0.00 | 84.3±0.00 | 53.9±0.00 | 85.7±0.00 |
| ASH | 93.1±0.00 | 54.1±0.00 | 95.1±0.00 | 53.0±0.00 | 94.1±0.00 | 53.6±0.00 | 96.8±0.00 | 50.5±0.00 | 98.7±0.00 | 47.9±0.00 | 94.5±0.00 | 55.4±0.00 | 96.7±0.00 | 51.3±0.00 |
| MDS | 95.0±0.00 | 47.7±0.00 | 96.7±0.00 | 47.3±0.00 | 95.8±0.00 | 47.5±0.00 | 89.5±0.00 | 64.0±0.00 | 97.2±0.00 | 55.0±0.00 | 94.9±0.00 | 53.7±0.00 | 93.9±0.00 | 57.5±0.00 |
| +ASL | 82.8±0.34 | 74.9±0.18 | 43.7±0.20 | 89.0±0.19 | 63.3±0.27 | 81.9±0.19 | 9.41±0.42 | 97.6±0.23 | 30.8±0.27 | 91.0±0.25 | 23.4±0.42 | 93.5±0.22 | 21.2±0.37 | 94.0±0.23 |
| RMDS | 85.8±0.00 | 72.5±0.00 | 46.8±0.00 | 87.3±0.00 | 66.3±0.00 | 79.9±0.00 | 19.4±0.00 | 96.1±0.00 | 37.2±0.00 | 89.4±0.00 | 29.7±0.00 | 92.3±0.00 | 28.8±0.00 | 92.6±0.00 |
| +ASL | 83.2±0.56 | 75.4±0.18 | 42.3±0.47 | 89.1±0.17 | 62.8±0.36 | 82.2±0.18 | 11.0±0.71 | 97.3±0.19 | 35.5±0.36 | 89.6±0.28 | 25.2±0.55 | 93.0±0.19 | 23.9±0.54 | 93.3±0.22 |
| VIM | 90.3±0.00 | 69.7±0.00 | 62.9±0.00 | 84.1±0.00 | 76.6±0.00 | 76.9±0.00 | 17.2±0.00 | 95.8±0.00 | 40.3±0.00 | 90.7±0.00 | 29.8±0.00 | 92.2±0.00 | 29.1±0.00 | 92.9±0.00 |
| +ASL | 90.6±1.02 | 63.9±0.38 | 81.5±1.07 | 76.2±0.54 | 86.0±1.05 | 70.0±0.46 | 35.0±1.17 | 90.7±0.23 | 62.1±0.61 | 85.2±0.16 | 50.4±1.17 | 87.4±0.31 | 49.2±0.95 | 87.7±0.23 |
| KNN | 85.9±0.00 | 66.0±0.00 | 55.7±0.00 | 82.1±0.00 | 70.8±0.00 | 74.0±0.00 | 27.8±0.00 | 91.6±0.00 | 33.6±0.00 | 91.1±0.00 | 34.8±0.00 | 89.9±0.00 | 32.1±0.00 | 90.9±0.00 |
| +ASL | 84.7±0.69 | 66.9±0.26 | 54.6±0.49 | 82.7±0.23 | 69.7±0.59 | 74.8±0.25 | 23.9±0.52 | 92.3±0.12 | 34.2±0.30 | 91.2±0.15 | 30.7±0.37 | 90.6±0.28 | 29.6±0.40 | 91.4±0.19 |
| fDBD | 87.0±0.00 | 65.7±0.00 | 60.5±0.00 | 80.7±0.00 | 73.8±0.00 | 73.2±0.00 | 32.8±0.00 | 90.5±0.00 | 44.9±0.00 | 88.5±0.00 | 39.6±0.00 | 89.0±0.00 | 39.1±0.00 | 89.3±0.00 |
| +ASL | 84.4±0.68 | 69.3±0.23 | 56.3±0.52 | 83.2±0.07 | 70.3±0.60 | 76.2±0.16 | 24.8±0.60 | 92.8±0.17 | 38.2±0.33 | 89.8±0.16 | 31.1±0.63 | 90.9±0.08 | 31.4±0.52 | 91.1±0.14 |
| MD++ | 83.1±0.00 | 71.1±0.00 | 50.3±0.00 | 86.0±0.00 | 66.7±0.00 | 78.5±0.00 | 20.7±0.00 | 94.4±0.00 | 33.6±0.00 | 89.6±0.00 | 27.6±0.00 | 91.7±0.00 | 27.3±0.00 | 91.9±0.00 |
| +ASL | 82.8±0.34 | 74.9±0.18 | 43.7±0.20 | 89.0±0.19 | 63.3±0.27 | 81.9±0.19 | 9.41±0.42 | 97.6±0.23 | 30.8±0.27 | 91.0±0.25 | 23.4±0.42 | 93.5±0.22 | 21.2±0.37 | 94.0±0.23 |
| *Training-based Methods* | | | | | | | | | | | | | | |
| CIDER | 84.3±0.78 | 69.4±0.40 | 59.5±0.87 | 82.8±0.13 | 71.9±0.83 | 76.1±0.26 | 32.8±1.19 | 92.3±0.22 | 43.0±0.56 | 89.1±0.25 | 35.7±0.66 | 90.8±0.33 | 37.2±0.77 | 90.7±0.27 |
| LogitNorm | 87.3±0.51 | 69.9±0.18 | 81.4±0.41 | 78.6±0.27 | 84.3±0.46 | 74.3±0.22 | 41.1±0.65 | 89.1±0.23 | 56.0±0.39 | 84.6±0.16 | 55.5±0.52 | 85.4±0.22 | 50.9±0.52 | 86.3±0.20 |
| PALM | 86.3±0.95 | 72.3±0.13 | 54.6±0.93 | 85.8±0.12 | 70.4±0.94 | 79.1±0.13 | 23.0±0.71 | 95.2±0.13 | 39.8±0.47 | 88.9±0.11 | 32.3±0.51 | 91.7±0.21 | 31.7±0.56 | 91.9±0.15 |
| ReweightOOD | 87.5±1.03 | 71.4±0.28 | 57.7±0.66 | 83.9±0.20 | 72.6±0.85 | 77.7±0.24 | 22.8±0.51 | 94.5±0.15 | 42.0±0.40 | 87.7±0.12 | 37.2±0.59 | 89.6±0.16 | 34.0±0.50 | 90.6±0.14 |
| T2FNorm | 80.9±0.78 | 74.2±0.18 | 85.5±0.57 | 78.1±0.23 | 83.2±0.68 | 76.1±0.25 | 46.2±0.50 | 91.5±0.20 | 89.0±0.42 | 76.4±0.13 | 86.0±0.52 | 82.4±0.16 | 73.7±0.49 | 83.4±0.16 |
| **ASL (Ours)** | 82.8±0.34 | 74.9±0.18 | 43.7±0.20 | 89.0±0.19 | **63.3±0.27** | **81.9±0.19** | 9.41±0.42 | 97.6±0.23 | 30.8±0.27 | 91.0±0.25 | 23.4±0.42 | 93.5±0.22 | **21.2±0.37** | **94.0±0.23** |

Table 20: OOD detection performance on CIFAR-10 under covariate shift.

| | Near-OOD | | | | | | Far-OOD | | | | | | | | | |
| | CIFAR-100 | | TIN | | Avg. | | MNIST | | SVHN | | Texture | | Places365 | | Avg. | |
| Method | FP↓ | AU↑ | FP↓ | AU↑ | FP↓ | AU↑ | FP↓ | AU↑ | FP↓ | AU↑ | FP↓ | AU↑ | FP↓ | AU↑ | FP↓ | AU↑ |
|---|---|---|---|---|---|---|---|---|---|---|---|---|---|---|---|---|
| *CE* | 69.5±0.24 | 73.3±0.24 | 61.8±0.58 | 75.8±0.60 | 65.7±0.41 | 74.6±0.42 | 40.2±0.43 | 85.5±0.18 | 41.6±0.62 | 83.1±0.12 | 44.6±0.44 | 84.4±0.26 | 68.5±0.12 | 74.7±0.15 | 48.7±0.03 | 81.9±0.12 |
| NPOS | 53.7±0.92 | 76.3±0.22 | 44.9±0.78 | 81.4±0.12 | 49.3±0.85 | 78.9±0.05 | 35.8±0.87 | 86.0±0.27 | 40.2±0.06 | 89.2±0.15 | 54.6±0.51 | 82.5±0.81 | 51.9±0.75 | 79.4±0.41 | 33.0±0.77 | 87.6±0.16 |
| HamOS | 54.1±0.78 | 77.1±1.66 | 45.7±0.97 | 82.3±0.22 | 49.9±0.87 | 79.7±0.72 | 34.1±1.20 | 87.0±1.70 | 12.7±9.08 | 97.7±5.84 | 43.2±2.37 | 85.2±1.01 | 47.5±1.36 | 81.9±0.78 | 34.4±1.64 | 87.9±1.94 |
| CIDER | 53.6±0.13 | 79.1±0.03 | 46.6±0.23 | 82.3±0.15 | 50.1±0.05 | 80.7±0.09 | 41.2±0.04 | 85.5±0.15 | 26.4±1.14 | 93.3±0.10 | 44.4±0.38 | 81.1±0.41 | 48.4±0.08 | 85.1±0.01 | 40.1±0.20 | 86.3±0.04 |
| LogitNorm | 59.5±0.20 | 76.6±0.05 | 51.0±0.07 | 81.5±0.08 | 55.3±0.14 | 79.0±0.07 | 9.09±0.32 | 98.1±0.08 | 36.3±1.06 | 87.4±0.55 | 51.7±0.74 | 83.8±0.25 | 51.0±0.04 | 83.3±0.02 | 37.0±0.36 | 88.2±0.17 |
| PALM | 50.8±0.40 | 80.3±0.08 | 44.0±1.34 | 83.9±0.13 | 47.4±0.87 | 82.1±0.11 | 32.5±0.77 | 89.7±0.16 | 16.3±2.08 | 97.2±0.20 | 36.2±1.00 | 90.4±0.70 | 48.2±2.67 | 86.9±0.51 | 33.3±1.13 | 91.0±0.04 |
| ReweightOOD | 57.0±0.24 | 77.4±0.02 | 44.1±0.03 | 84.3±0.03 | 50.6±0.13 | 80.9±0.03 | 23.6±0.36 | 91.1±0.11 | 2.41±0.07 | 99.5±0.01 | 29.3±0.02 | 93.5±0.06 | 41.0±0.37 | 86.5±0.16 | 24.1±1.03 | **92.7±0.00** |
| T2FNorm | 57.9±0.07 | 76.9±0.43 | 48.5±1.30 | 81.5±0.56 | 53.2±0.69 | 79.2±0.49 | 23.9±0.23 | 93.3±0.01 | 29.4±3.52 | 92.0±1.60 | 46.6±1.25 | 84.8±0.49 | 52.5±0.93 | 82.3±0.41 | 38.1±1.02 | 88.1±0.42 |
| **ASL (Ours)** | 51.0±0.06 | 81.3±0.09 | 38.9±0.06 | 86.5±0.08 | **45.0±0.06** | **83.9±0.08** | 23.3±0.13 | 92.8±0.05 | 1.60±0.07 | 99.7±0.14 | 31.7±0.07 | 90.4±0.08 | 39.4±0.05 | 87.1±0.06 | **24.0±0.08** | 92.5±0.08 |

Table 21: OOD detection performance on CIFAR-100 under covariate shift.

| | Near-OOD | | | | | | Far-OOD | | | | | | | | | |
| | CIFAR-100 | | TIN | | Avg. | | MNIST | | SVHN | | Texture | | Places365 | | Avg. | |
| Method | FP↓ | AU↑ | FP↓ | AU↑ | FP↓ | AU↑ | FP↓ | AU↑ | FP↓ | AU↑ | FP↓ | AU↑ | FP↓ | AU↑ | FP↓ | AU↑ |
|---|---|---|---|---|---|---|---|---|---|---|---|---|---|---|---|---|
| *CE* | 82.5±0.33 | 60.2±0.37 | 72.8±0.16 | 65.8±0.49 | 77.7±0.25 | 63.0±0.43 | 63.6±0.31 | 73.2±0.14 | 66.9±0.07 | 70.5±0.34 | 59.9±0.01 | 80.3±0.73 | 76.6±0.04 | 65.8±0.56 | 66.7±0.09 | 72.5±0.27 |
| NPOS | 82.8±2.81 | 67.7±0.25 | 66.9±1.95 | 74.8±2.38 | 71.1±0.21 | 76.3±1.89 | 69.7±2.32 | 40.2±0.06 | 89.2±0.15 | 54.6±0.51 | 82.5±0.81 | 68.2±0.18 | 71.5±0.21 | 59.8±0.57 | 78.2±0.87 |
| HamOS | 79.1±0.57 | 69.4±0.96 | 64.1±0.97 | 74.8±0.55 | 71.6±0.20 | **72.1±0.75** | 81.9±1.51 | 61.8±1.95 | 47.1±1.18 | 85.6±1.07 | 61.4±0.45 | 75.2±0.14 | 65.6±0.66 | 73.3±0.37 | 64.0±0.36 | 74.0±0.34 |
| CIDER | 83.0±0.30 | 63.4±0.16 | 68.4±0.18 | 71.9±0.36 | 75.7±0.24 | 67.7±0.26 | 83.4±0.19 | 52.4±0.07 | 50.5±0.10 | 90.1±0.02 | 69.2±0.06 | 72.2±0.02 | 75.2±0.14 | 69.8±0.08 | 69.6±0.12 | 71.1±0.05 |
| LogitNorm | 82.5±0.13 | 65.5±0.14 | 65.7±0.12 | 74.7±0.16 | 74.1±0.12 | 70.1±0.15 | 40.1±0.10 | 86.9±0.08 | 56.5±0.21 | 75.4±0.14 | 86.2±0.16 | 62.5±0.18 | 68.1±0.05 | 73.0±0.04 | 62.7±0.11 | 74.4±0.11 |
| PALM | 80.9±0.33 | 59.1±0.14 | 65.5±0.31 | 69.5±0.24 | 73.2±0.32 | 64.3±0.19 | 61.1±0.23 | 65.8±0.16 | 16.3±0.13 | 96.5±0.10 | 52.7±0.13 | 79.0±0.18 | 66.4±0.30 | 68.2±0.21 | 49.1±0.20 | 77.4±0.16 |
| ReweightOOD | 81.0±0.22 | 62.1±0.10 | 68.0±0.16 | 74.1±0.12 | 74.5±0.19 | 68.1±0.11 | 75.6±0.26 | 70.5±0.19 | 20.1±0.26 | 96.5±0.13 | 46.2±0.14 | 88.0±0.08 | 65.6±0.15 | 76.9±0.06 | **51.9±0.20** | **83.0±0.12** |
| T2FNorm | 79.3±0.20 | 65.6±0.13 | 63.6±0.12 | 75.0±0.10 | 71.5±0.16 | 70.3±0.12 | 44.5±0.15 | 84.3±0.16 | 61.8±0.09 | 75.1±0.14 | 79.5±0.18 | 67.1±0.08 | 68.5±0.15 | 71.4±0.08 | 63.6±0.14 | 74.5±0.11 |
| **ASL (Ours)** | 79.0±0.11 | 67.0±0.25 | 62.9±0.15 | 75.5±0.07 | **71.0±0.13** | 71.2±0.16 | 63.2±0.27 | 74.3±0.15 | 31.7±0.16 | 93.0±0.10 | 52.2±0.19 | 81.8±0.12 | 65.2±0.15 | 74.7±0.08 | 53.1±0.19 | 81.0±0.12 |

Table 22: OOD detection performance on ImageNet-200 under covariate shift.

| | Near-OOD | | | | | | Far-OOD | | | | | | | |
| | SSB-hard | | NINCO | | Avg. | | iNaturalist | | Textures | | OpenImage-O | | Avg. | |
| Method | FP↓ | AU↑ | FP↓ | AU↑ | FP↓ | AU↑ | FP↓ | AU↑ | FP↓ | AU↑ | FP↓ | AU↑ | FP↓ | AU↑ |
|---|---|---|---|---|---|---|---|---|---|---|---|---|---|---|
| *CE* | 78.1±0.86 | 70.2±0.06 | 55.6±1.79 | 79.2±0.48 | 66.9±1.32 | 74.7±0.21 | 28.4±1.38 | 90.0±0.64 | 22.1±0.12 | 95.1±0.34 | 39.9±0.32 | 86.0±0.24 | 30.1±0.61 | 90.3±0.41 |
| CIDER | 75.2±1.10 | 68.0±0.20 | 53.0±0.73 | 78.0±0.19 | 64.1±0.92 | 73.0±0.20 | 29.1±1.00 | 88.9±0.27 | 26.0±0.35 | 94.0±0.32 | 36.4±0.46 | 85.2±0.14 | 30.5±0.60 | 89.4±0.24 |
| LogitNorm | 71.7±0.91 | 71.3±0.20 | 54.1±0.63 | 79.9±0.13 | 62.9±0.14 | 75.6±0.03 | 24.5±0.76 | 92.0±0.19 | 40.3±0.40 | 87.4±0.21 | 39.7±0.54 | 85.0±0.17 | 34.8±0.21 | 88.1±0.19 |
| PALM | 74.8±1.56 | 67.8±0.18 | 57.5±0.51 | 77.1±0.33 | 66.2±0.52 | 72.4±0.25 | 31.8±0.30 | 88.2±0.15 | 22.2±0.28 | 94.5±0.20 | 38.7±0.33 | 84.5±0.22 | 30.9±0.30 | 89.1±0.19 |
| ReweightOOD | 72.0±1.38 | 72.9±0.71 | 49.7±0.56 | 79.3±0.31 | 60.9±0.41 | 76.1±0.20 | 27.6±1.81 | 89.0±0.74 | 28.8±0.64 | 92.3±0.46 | 36.8±0.57 | 85.2±0.87 | 31.1±1.00 | 88.8±0.69 |
| T2FNorm | 71.5±1.06 | 72.8±0.27 | 54.2±0.53 | 80.1±0.23 | 62.9±0.26 | 76.5±0.02 | 23.6±0.98 | 92.4±0.47 | 43.8±0.31 | 83.8±0.15 | 39.1±0.40 | 84.7±0.10 | 35.5±0.30 | 86.9±0.17 |
| **ASL (Ours)** | 70.7±0.49 | 74.9±0.34 | 47.2±1.11 | 82.3±0.36 | **59.0±0.80** | **78.6±0.35** | 23.8±0.93 | 92.5±0.65 | 30.4±1.86 | 91.9±0.13 | 33.1±0.64 | 88.0±0.03 | **29.1±0.52** | **90.8±0.16** |

ImageNet, as shown in Table 23 (for the ResNet-50 model) and Table 24 (for the ViT-B/16 model), ASL also exhibits significant effectiveness in fine-tuning for OOD detection.

Table 23: OOD detection performance on ImageNet-1K with ResNet-50 under covariate shift.

| | Near-OOD | | | | | | Far-OOD | | | | | | | |
| | SSB-hard | | NINCO | | Avg. | | iNaturalist | | Textures | | OpenImage-O | | Avg. | |
| Method | FP↓ | AU↑ | FP↓ | AU↑ | FP↓ | AU↑ | FP↓ | AU↑ | FP↓ | AU↑ | FP↓ | AU↑ | FP↓ | AU↑ |
|---|---|---|---|---|---|---|---|---|---|---|---|---|---|---|
| *CE* | $85.3_{\pm0.00}$ | $56.7_{\pm0.00}$ | $68.3_{\pm0.00}$ | $70.6_{\pm0.00}$ | $76.8_{\pm0.00}$ | $63.6_{\pm0.00}$ | $41.5_{\pm0.00}$ | $83.0_{\pm0.00}$ | $13.6_{\pm0.00}$ | $96.4_{\pm0.00}$ | $47.2_{\pm0.00}$ | $83.3_{\pm0.00}$ | $34.1_{\pm0.00}$ | $87.6_{\pm0.00}$ |
| CIDER | $90.5_{\pm1.79}$ | $52.7_{\pm0.51}$ | $71.8_{\pm0.82}$ | $70.0_{\pm0.28}$ | $81.2_{\pm0.49}$ | $61.3_{\pm0.11}$ | $33.6_{\pm1.41}$ | $88.5_{\pm0.35}$ | $31.6_{\pm0.42}$ | $92.6_{\pm0.45}$ | $59.2_{\pm0.68}$ | $82.7_{\pm0.13}$ | $41.5_{\pm0.38}$ | $88.0_{\pm0.22}$ |
| LogitNorm | $84.5_{\pm0.62}$ | $60.7_{\pm0.30}$ | $63.0_{\pm0.25}$ | $74.7_{\pm0.23}$ | $73.8_{\pm0.18}$ | $67.7_{\pm0.04}$ | $26.8_{\pm0.86}$ | $91.1_{\pm0.23}$ | $48.2_{\pm0.29}$ | $83.9_{\pm0.18}$ | $40.4_{\pm1.22}$ | $85.4_{\pm0.15}$ | $38.5_{\pm0.21}$ | $86.8_{\pm0.03}$ |
| PALM | $86.6_{\pm0.74}$ | $57.2_{\pm0.30}$ | $59.9_{\pm0.36}$ | $73.5_{\pm0.28}$ | $73.2_{\pm0.55}$ | $65.4_{\pm0.29}$ | $25.1_{\pm0.47}$ | $91.1_{\pm0.21}$ | $25.0_{\pm0.33}$ | $93.7_{\pm0.19}$ | $38.4_{\pm0.11}$ | $86.9_{\pm0.18}$ | $29.5_{\pm0.08}$ | $\mathbf{90.6}_{\pm0.20}$ |
| ReweightOOD | $86.3_{\pm0.50}$ | $54.9_{\pm0.26}$ | $63.8_{\pm0.26}$ | $72.7_{\pm0.17}$ | $75.1_{\pm0.38}$ | $63.8_{\pm0.22}$ | $27.9_{\pm0.53}$ | $87.2_{\pm0.18}$ | $17.3_{\pm0.45}$ | $95.1_{\pm0.20}$ | $40.5_{\pm0.39}$ | $86.1_{\pm0.15}$ | $\underline{28.6}_{\pm0.45}$ | $\underline{89.5}_{\pm0.17}$ |
| T2FNorm | $83.8_{\pm0.29}$ | $63.7_{\pm0.22}$ | $60.6_{\pm0.98}$ | $77.2_{\pm0.12}$ | $\underline{72.2}_{\pm0.64}$ | $\underline{70.4}_{\pm0.17}$ | $26.4_{\pm0.88}$ | $92.1_{\pm0.23}$ | $56.6_{\pm0.43}$ | $81.1_{\pm0.21}$ | $41.8_{\pm0.42}$ | $85.6_{\pm0.29}$ | $41.6_{\pm0.57}$ | $86.3_{\pm0.24}$ |
| **ASL (Ours)** | $84.4_{\pm0.40}$ | $65.6_{\pm0.20}$ | $56.0_{\pm0.35}$ | $79.1_{\pm0.20}$ | $\mathbf{70.2}_{\pm0.38}$ | $\mathbf{72.3}_{\pm0.20}$ | $30.0_{\pm0.74}$ | $91.0_{\pm0.19}$ | $42.4_{\pm0.30}$ | $87.9_{\pm0.23}$ | $37.6_{\pm0.54}$ | $87.1_{\pm0.09}$ | $36.7_{\pm0.33}$ | $88.7_{\pm0.11}$ |

Table 24: OOD detection performance on ImageNet-1K with ViT-B/16 model under covariate shift.

| | Near-OOD | | | | | | Far-OOD | | | | | | | |
| | SSB-hard | | NINCO | | Avg. | | iNaturalist | | Textures | | OpenImage-O | | Avg. | |
| Method | FP↓ | AU↑ | FP↓ | AU↑ | FP↓ | AU↑ | FP↓ | AU↑ | FP↓ | AU↑ | FP↓ | AU↑ | FP↓ | AU↑ |
|---|---|---|---|---|---|---|---|---|---|---|---|---|---|---|
| *CE* | $86.0_{\pm0.00}$ | $66.9_{\pm0.00}$ | $55.2_{\pm0.00}$ | $82.3_{\pm0.00}$ | $\underline{70.6}_{\pm0.00}$ | $74.6_{\pm0.00}$ | $26.8_{\pm0.00}$ | $93.9_{\pm0.00}$ | $45.4_{\pm0.00}$ | $85.4_{\pm0.00}$ | $36.9_{\pm0.00}$ | $88.9_{\pm0.00}$ | $\underline{36.4}_{\pm0.00}$ | $\underline{89.4}_{\pm0.00}$ |
| CIDER | $86.4_{\pm1.15}$ | $65.2_{\pm0.44}$ | $63.7_{\pm0.87}$ | $78.8_{\pm0.20}$ | $75.0_{\pm0.14}$ | $72.0_{\pm0.12}$ | $38.2_{\pm0.93}$ | $89.5_{\pm0.33}$ | $47.9_{\pm0.60}$ | $85.6_{\pm0.21}$ | $41.7_{\pm0.89}$ | $87.3_{\pm0.58}$ | $42.6_{\pm0.43}$ | $87.5_{\pm0.23}$ |
| LogitNorm | $88.9_{\pm0.76}$ | $66.3_{\pm0.16}$ | $87.0_{\pm0.38}$ | $74.7_{\pm0.27}$ | $88.0_{\pm0.18}$ | $70.5_{\pm0.05}$ | $49.4_{\pm1.01}$ | $86.2_{\pm0.59}$ | $68.8_{\pm0.49}$ | $80.8_{\pm0.18}$ | $69.4_{\pm0.64}$ | $81.8_{\pm0.11}$ | $62.5_{\pm0.29}$ | $82.9_{\pm0.17}$ |
| PALM | $87.9_{\pm0.54}$ | $68.6_{\pm0.12}$ | $59.1_{\pm0.59}$ | $82.1_{\pm0.16}$ | $73.5_{\pm0.23}$ | $\underline{75.4}_{\pm0.14}$ | $28.5_{\pm0.63}$ | $93.0_{\pm0.32}$ | $45.1_{\pm0.36}$ | $85.3_{\pm0.23}$ | $37.8_{\pm0.40}$ | $88.5_{\pm0.33}$ | $37.1_{\pm0.20}$ | $88.9_{\pm0.08}$ |
| ReweightOOD | $89.0_{\pm0.72}$ | $67.5_{\pm0.16}$ | $62.0_{\pm0.30}$ | $79.7_{\pm0.23}$ | $75.5_{\pm0.21}$ | $73.6_{\pm0.19}$ | $28.7_{\pm0.48}$ | $91.7_{\pm0.24}$ | $47.4_{\pm0.45}$ | $83.5_{\pm0.26}$ | $42.9_{\pm0.44}$ | $85.7_{\pm0.38}$ | $39.7_{\pm0.17}$ | $87.0_{\pm0.13}$ |
| T2FNorm | $82.3_{\pm0.61}$ | $71.1_{\pm0.15}$ | $86.5_{\pm0.21}$ | $75.3_{\pm0.10}$ | $84.4_{\pm0.41}$ | $73.2_{\pm0.13}$ | $50.2_{\pm0.79}$ | $89.5_{\pm0.15}$ | $89.8_{\pm0.40}$ | $73.8_{\pm0.26}$ | $87.0_{\pm0.75}$ | $80.0_{\pm0.33}$ | $75.6_{\pm0.12}$ | $81.1_{\pm0.25}$ |
| **ASL (Ours)** | $85.3_{\pm0.36}$ | $70.1_{\pm0.11}$ | $50.0_{\pm0.35}$ | $84.4_{\pm0.15}$ | $\mathbf{67.7}_{\pm0.11}$ | $\mathbf{77.3}_{\pm0.13}$ | $15.9_{\pm0.61}$ | $95.3_{\pm0.20}$ | $37.9_{\pm0.22}$ | $86.1_{\pm0.16}$ | $30.6_{\pm0.25}$ | $89.3_{\pm0.26}$ | $\mathbf{28.1}_{\pm0.21}$ | $\mathbf{90.2}_{\pm0.07}$ |

Table 25: Ablation study on the linear bias term for OOD detection performance on CIFAR-10 and CIFAR-100.

| | CIFAR-10 | | | | | | | CIFAR-100 | | | | | |
| | Near-OOD | | Far-OOD | | Average | | | Near-OOD | | Far-OOD | | Average | |
| | FP↓ | AU↑ | FP↓ | AU↑ | FP↓ | AU↑ | | FP↓ | AU↑ | FP↓ | AU↑ | FP↓ | AU↑ |
|---|---|---|---|---|---|---|---|---|---|---|---|---|---|
| With Bias | 26.3 | 93.0 | 13.5 | 97.0 | 19.9 | 95.0 | With Bias | 55.5 | 80.9 | 49.7 | 83.9 | 52.6 | 82.4 |
| W/o Bias | 25.1 | 93.3 | 12.0 | 97.2 | 18.6 | 95.3 | W/o Bias | 56.2 | 80.9 | 47.8 | 85.0 | 52.0 | 83.0 |

## E.6 ABLATION OF LINEAR CLASSIFICATION HEAD WITHOUT BIAS

As shown in equation 3, when the linear bias term is removed, the model structure becomes consistent with prototype-based contrastive learning, allowing the model to focus more on angular optimization. To verify this, we conduct an ablation study (Table 25). On CIFAR-10, the model without bias achieves a superior average FPR95 of 18.6% and AUROC of 95.3%, compared to 19.9% and 95.0% for the baseline with bias. A similar trend is observed on the more challenging CIFAR-100 benchmark. The bias-free model variant outperforms its counterpart, achieving an average FPR95 of 52.0% and AUROC of 83.0% versus 52.6% and 82.4%. The results indicate that removing the bias term not only enhances training efficiency (Table 7) but also mitigates the slight perturbations that are unbeneficial to the OOD discrimination task.

Table 26: Comparison of OOD detection performance under different feature normalization schemes on CIFAR-10 and CIFAR-100. $\ell_2$ normalization yields the best results, as it optimally enforces angular discrimination on the unit hypersphere.

| | CIFAR-10 | | | | | | | CIFAR-100 | | | | | |
| | Near-OOD | | Far-OOD | | Average | | | Near-OOD | | Far-OOD | | Average | |
| | FP↓ | AU↑ | FP↓ | AU↑ | FP↓ | AU↑ | | FP↓ | AU↑ | FP↓ | AU↑ | FP↓ | AU↑ |
|---|---|---|---|---|---|---|---|---|---|---|---|---|---|
| $\ell_1$ | 51.4 | 86.6 | 36.9 | 91.0 | 44.1 | 88.8 | $\ell_1$ | 70.3 | 75.2 | 75.7 | 73.8 | 73.0 | 74.5 |
| $\ell_2$ | 25.1 | 93.3 | 12.0 | 97.2 | 18.6 | 95.3 | $\ell_2$ | 56.2 | 80.9 | 47.8 | 85.0 | 52.0 | 83.0 |
| $\ell_\infty$ | 30.6 | 91.6 | 12.7 | 97.2 | 21.7 | 94.4 | $\ell_\infty$ | 68.5 | 73.8 | 33.5 | 90.4 | 51.0 | 82.1 |

## E.7 ABLATION OF DIFFERENT NORM-BASED NORMALIZATION SCHEMES

We investigate the impact of different norm-based normalizations. Since norms higher than $\ell_2$ may lead to numerical instability, we only consider $\ell_1$, $\ell_2$, and $\ell_\infty$ normalizations. The results on CIFAR datasets are presented in Table 26. Among the three, $\ell_2$ normalization consistently achieves the best

performance, which aligns with our motivation to optimize features on the unit hypersphere. This geometric constraint encourages the model to rely solely on angular information for discrimination, thereby mitigating the sensitivity to feature norms that plagues traditional Mahalanobis-based detectors. In contrast, $\ell_1$ normalization projects features onto a diamond-shaped surface, which is not isotropic and may introduce bias in certain directions. Although $\ell_\infty$ normalization also constrains the feature magnitude, it projects onto a hypercube surface, which is less symmetric and geometrically less suitable for near-OOD. These results confirm that $\ell_2$ normalization is the most effective choice for inducing a well-structured feature space conducive to OOD detection.

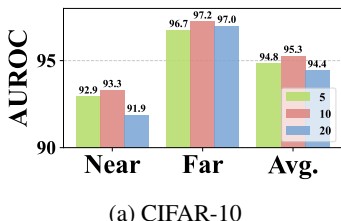
(a) CIFAR-10

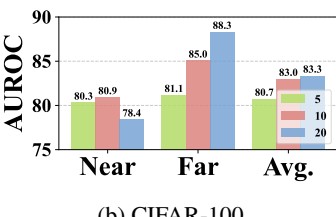
(b) CIFAR-100

Figure 22: Ablation of feature scaling factor $s$.

### E.8 ABLATION OF FEATURE SCALING FACTOR

We conduct an ablation study on the CIFAR-10 and CIFAR-100 datasets to evaluate the impact of the feature scaling factor $s$ on model performance. This factor is applied to features after $\ell_2$-normalization with the aim of regulating gradient magnitudes. We experimented with three different scaling values: 5, 10, and 20.

As shown in Fig. 22, on CIFAR-10, in terms of average performance, $s = 10$ (95.3%) is the optimal choice. On CIFAR-100, the results are consistent with our previous analysis from the temperature ablation study. A higher scaling factor leads the model to learn more coarse-grained representations, which is detrimental to near-OOD performance. A smaller scaling factor of $s = 5$ impacts the model's training speed and, while showing a consistent trend with $s = 10$, results in an overall decrease in performance.

Overall, a scaling factor of $s = 10$ demonstrates the most stable and superior performance across both benchmarks. This indicates that appropriately adjusting the feature scale can further optimize training dynamics, thereby achieving optimal results in challenging OOD detection scenarios.

Table 27: Comparison with angular learning methods on CIFAR-10 and CIFAR-100.

| | CIFAR-10 | | | | CIFAR-100 | | | |
|---|---|---|---|---|---|---|---|---|
| | Near-OOD | | Far-OOD | | Near-OOD | | Far-OOD | |
| Method | FP↓ | AU↑ | FP↓ | AU↑ | FP↓ | AU↑ | FP↓ | AU↑ |
| CE | 40.2 | 88.8 | 30.6 | 91.0 | 67.8 | 74.8 | 51.9 | 82.6 |
| SpereFace | 46.1 | 88.3 | 27.9 | 92.5 | 73.4 | 72.4 | 68.6 | 69.7 |
| Cosine Cls | 26.6 | 93.0 | 15.6 | 96.3 | 57.7 | 80.0 | 54.4 | 81.1 |
| ASL | **25.1** | **93.3** | **12.0** | **97.2** | **56.2** | **80.9** | **47.8** | **85.0** |

### E.9 COMPARISON WITH ANGULAR LEARNING

To more comprehensively evaluate the effectiveness of ASL in angular space learning, we compare it with two related methods here:

**SphereFace (Liu et al., 2017)** introduces an angular margin loss to enhance feature discriminability for face recognition. It employs a multiplicative angular margin in the softmax loss to enforce larger inter-class angular separations.

**Cosine Classifier** applies $\ell_2$-normalization to classifier weights and features.

As shown in Table 27, ASL consistently outperforms SphereFace and Cosine Classifier. SphereFace's angular margin formulation appears detrimental to OOD detection, the margin con-

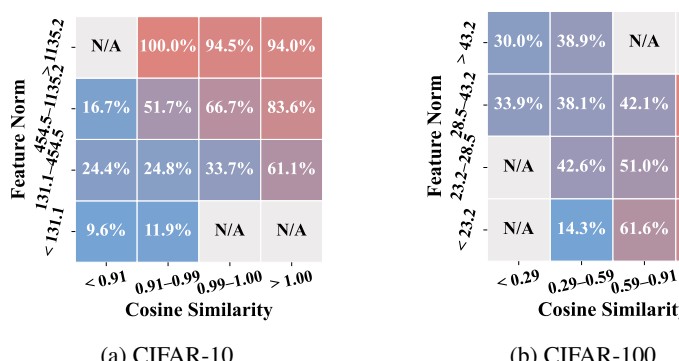

(a) CIFAR-10        (b) CIFAR-100

Figure 23: Heatmap of sample distribution for SphereFace in Mahalanobis space.

straint may reducing the feature space's capacity to distinguish OOD samples. The Cosine Classifier demonstrates the benefits of angular optimization, particularly for near-OOD detection.

To further analyze the performance of SphereFace, we visualize the sample distribution in the Mahalanobis space (Fig. 23), following the same protocol as in previous sections. The results reveal that the rigid margin constraint induces two pathological geometric behaviors: 1) On the CIFAR-10 dataset, SphereFace achieves extreme intra-class compactness, where ID samples are compressed into a narrow angular cone (with a cosine similarity lower quartile of 0.91). While this maximizes ID concentration, it resembles feature collapse and reduces the angular margin available for OOD rejection. Consequently, OOD samples are easily mapped into these high-similarity regions. 2) Extreme angular compression leads to covariance collapse, where intra-class variance in tangential directions is minimal. In the Mahalanobis space, the precision matrix ($\hat{\Sigma}^{-1}$) becomes ill-conditioned, acting as an unstable amplifier. ID samples are consequently projected to pathologically high norms. In contrast, PALM and ASL allow for appropriate intra-class variance, ensuring $\hat{\Sigma}$ remains well-conditioned.

## F  LIMITATIONS AND FUTURE WORK

Although ASL learns a well-structured feature space that enables strong performance with simple distance-based scoring, one aspect that merits further investigation is the exploration of post-hoc scoring methods on hyperspheres. Our current evaluation primarily relies on the normalized Mahalanobis distance, which has already yielded significant improvements. However, the $\ell_2$-normalized, angle-centered feature space learned by ASL exhibits distinctive geometric properties including hyperspherical structure and potential non-Gaussian intra-class distributions that remain underutilized. Future work could develop specialized scoring mechanisms that better leverage the high separability and hyperspherical characteristics of ASL features, potentially leading to further performance gains.

## G  DECLARATION OF LLM USAGE

During the writing process of this article, the large language models (Gemini 2.5 Pro, DeepSeek-R1) were used as a writing aid in a limited manner. All usage behaviors were limited to text-level optimization and polishing, and did not involve the core aspects of the research (including but not limited to core method design, experimental plan formulation, data collection and analysis, result demonstration, etc.), which does not affect the scientific rigor, method originality and conclusion reliability of this study.

