# OpenReview forum: "Revisiting Out-of-Distribution Detection: Angular Separation Learning as a Powerful and Simple Baseline"
_ICLR.cc/2026/Conference — Submitted to ICLR 2026_

### Official Review · Reviewer_puLH · 2025-10-20

**Soundness:** 3
**Presentation:** 3
**Contribution:** 2
**Rating:** 2
**Confidence:** 4

**Summary:**

## Summary
The paper proposes Angular Separation Learning (ASL) as a simple, training-time technique for OOD detection: normalize feature representations before the classification head to emphasize angular decisions. The authors hypothesize that feature-norm is an overlooked factor; they analyze Mahalanobis distance, highlight failure cases with low-norm OOD features and show strong performance on multiple benchmarks. They also compare with post-hoc normalization methods (e.g., Mahalanobis++) underperform ASL.

**Strengths:**

- **Simplicity.** Very simple algorithm and presented in a clear way.
- **Low overhead, strong results.** A lightweight add-on achieving strong performance.
- **Broad evaluation.** Covers many cases; Layerwise analysis done on Figure 5 is particularly a nice add.
- **Fair comparisons.** Side-by-side with state-of-the-art post-hoc scoring methods, highlighting gains attributable to the training scheme.

**Weaknesses:**

- **Overstated claims.**
  - Calling other training-based methods “complex” (line 15) / “exotic” (line 24) is too strong. Please tone down language unless quantitatively/qualitatively supported.
  - Line 111: [1], [2], [3] are works that already studied Mahalanobis and its corrections. Calling it “under-studied” is not accurate.
  - Line 407: Real-world practicality is an overstatement. One would not use a model with FPR95 ~ 20% on safety-critical applications.

- **Figure 1 clarity.**
  Low-norm OOD representations are closer to ID is a strong observation, but I find Figure 1 unclear: how exactly computed, how thresholds selected, which backbone/dataset used, and whether repeated across datasets/backbones.

- **Motivation vs. cosine head.**
  If the goal is angular learning, a cosine-based classifier (normalized features and normalized weights) is a straightforward alternative. Authors mention applying weight decay to bound weight norms, but do not justify not normalizing weights. Please add an ablation to evaluate:
  - Whether norm cancellation and a truly cosine classifier improve OOD detection.
  - Whether cosine model feature representations align with Section 3.2’s intuition.
  - If cosine performs worse, provide reasoning and supporting experiments if it does not agree with the intuition provided.

- **Low-norm failure analysis.**
  Beyond Fig. 1, selecting the 95% ID validation threshold, collecting false-positive OODs, and plotting their norm distribution alongside ID validation norms would be an easy validation for the main argument that we need to solve the low norm OOD sample symptom.

- **Related work (post-hoc).**
  It is a rather crowded line of research. Therefore, the coverage lags the literature. Please consider [4] for strong and simple baselines and extensive evaluation and also [5], [6], [7].

- **Norm as signal vs. hypothesis.**
  Line 150 cites a work using norm as a discriminative signal, which seems at odds with the main hypothesis. Either show [8] does not apply consistently or cite works aligned with your stance (e.g., [2], [5], [9]).

- **Intuition around Eq. (1).**
  The discussion around line 198 is confusing. In ideal CE, ID feature and class mean coincide (therefore, distance becomes 0). Interpreting the equation "holding everything else constant" may mislead; many configurations allow high class separation and tight clusters with large norms. The issue seems more about ambiguous CE feature assignments to ID/OOD than "MDS sensitivity to norm" per se; consider analysis grounded in training dynamics (possibly with synthetic settings).

- **Training efficiency.**
  Include confidence intervals from repeated runs after warmup. The table is concerning as there’s no clear reason LogitNorm should be slower than ASL. Control all hyperparameters (especially batch size) if reporting wall-time.

**Questions:**

- Figure 3 is aesthetically nice but uses large space; consider rearranging to save space.
- In Figure 1, what metric defines "nearest class mean"? L2 or cosine?
- In imbalanced classes, weight/feature norms may contain priors probabilities if we interpret CE trained models as posterior predictors (~ $p(y) \cdot p(x|y)$). How does normalization behave in this setting?
- Table 3 (CIFAR-100): PALM has the lowest FPR, yet both ASL and PALM are bolded.
- Can ASL be straightforwardly combined with activation shaping (e.g., ReAct, ASH)? It would be interesting to see interactions.
- Since training/optimization is involved, please report confidence intervals with repeated experiments over different random seeds. Any reason this was not done?


## References
[1] Ren et al., 2021. *A simple fix to Mahalanobis distance for improving near-OOD detection*. arXiv:2106.09022.
[2] Mueller & Hein, 2025. *Mahalanobis++: Improving OOD Detection via Feature Normalization*. arXiv:2505.18032.
[3] Shi et al., 2013. *Improved relative-transformation PCA based on Mahalanobis distance*. Acta Automatica Sinica.
[4] Bitterwolf, Mueller, Hein, 2023. *In or out? Fixing ImageNet OOD detection evaluation*. arXiv:2306.00826.
[5] Demirel, Fumero, Locatello, 2024. *Out-of-Distribution Detection with Relative Angles*. arXiv:2410.04525.
[6] Liu & Qin, 2025. *Detecting OOD through the lens of neural collapse*. CVPR.
[7] Ammar et al., 2023. *NECO: Neural collapse based OOD detection*. arXiv:2310.06823.
[8] Zhang & Xiang, 2023. *Decoupling maxlogit for OOD detection*. CVPR.
[9] Sun et al., 2022. *OOD detection with deep nearest neighbors*. ICML.

---

> ### Author Response · Authors · 2025-11-17
>
> We sincerely appreciate the time and effort put into reviewing our paper and providing valuable feedback. We would like to address your questions below.
>
> > Q1: Overstated claims.
>
> Thank you for your valuable advice. We have revised and improved the relevant statements in the revised version.
>
> > Q2: Figure 1
>
> We hereby concise clarify Fig. 1 details:
>
> 1. **Mahalanobis Space Computation**: Original features are whitened via $g(x)=\hat{\sum}^{-1/2}f(x)$ (feature) and $\hat{u}_c=\hat{\sum}^{-1/2}\hat{\mu}_c$ (class mean), where $\hat{\sum}$ is ID training covariance.
> 2. **"Nearest Class Mean" Metric**: The "nearest class mean" in Fig. 1 is quantified using cosine similarity between $g(x)$ and $\hat{u}_c$, which aligns with our focus on angular separation in the method.
> 3. **Threshold Selection**: Thresholds are determined independently for the two dimensions (feature $\ell_2$-norm of $g(x)$ and cosine similarity to $\hat{u}_c$), for each dimension, we calculate statistical thresholds that divide the distribution of ID samples into 4 equal parts, these thresholds partition the Mahalanobis space into a 4×4 grid, and each cell in the heatmap represents the proportion of ID samples falling into that region.
> 4. **Backbone/Dataset**: The model uses ResNet-18, which is consistent with our standard experimental setup for CIFAR datasets. Fig. 1 presents results on the CIFAR-10 dataset, with the heatmap reflecting the average performance on CIFAR-10’s Near-OOD datasets. We have repeated this analysis across other datasets (CIFAR-100, ImageNet-200) and provided the corresponding heatmaps in the Appendix.
>
> > Q3: Motivation vs. cosine head.
>
> We have experimentally compared the performance of ASL and the Cosine Classifier (Tables A).
>
> **Alignment with Section 3.2 Intuition.** Both ASL and the Cosine Classifier outperform CE, validating Section 3.2’s core intuition: "reducing feature norm interference improves OOD detection". ASL’s superiority stems from its dynamic weight constraint (weight decay) (bounds $\left\|w_j\right\|_2$ adaptively) vs. the Cosine Classifier’s rigid weight normalization. The adaptive "temperature" from weight decay enables finer angular tuning for OOD samples.
>
>
> Table A: Comparison of ASL with Cosine Classifier and SphereFace for OOD Detection
> | Method           | CIFAR-10 |||| CIFAR-100 ||||
> |:------------------|:-----:|:-----|:-----:|:-----|:-----:|:-----|:-----:|:-----|
> |                  | Near || Far || Near || Far |
> |                  | FP↓  | AU↑  | FP↓   | AU↑  | FP↓   | AU↑  | FP↓   | AU↑  |
> | CE               | 40.2       | 88.8       | 30.6       | 91.0       | 67.8       | 74.8       | 51.9       | 82.6       |
> | SpereFace        | 46.1       | 88.3       | 27.9       | 92.5       | 73.4       | 72.4       | 68.6       | 69.7       |
> | Cosine Classifier| 26.6       | 93.0       | 15.6       | 96.3       | 57.7       | 80.0       | 54.4       | 81.1       |
> | ASL              | **25.1**       | **93.3**       | **12.0**       | **97.2**       | **56.2**       | **80.9**       | **47.8**       | **85.0**       |
>
>
> > Q4: Low-norm failure analysis.
>
> We have added the requested analysis in Appendix D.4 of the revised manuscript.
> Following your recommendation, we selected the 95% TPR threshold from ID validation data, collected false-positive OOD samples, and plotted their feature norm distributions alongside ID norms for MDS, MD++, and ASL (Figures 15-17).
>
> > Q5: It is a rather crowded line of research. Therefore, the coverage lags the literature. Please consider [1] for strong and simple baselines and extensive evaluation and also [2], [3], [4].
>
> Thank you for highlighting these relevant works.
> - The NINCO[1] dataset is already integrated into our evaluation, and we report its performance as a key near-OOD benchmark for ImageNet-200 and ImageNet.
> - We have added discussions of ORA [2], NCI [3], and NECO [4] in the revised Related Work (Section 2). While these works successfully leverage angular and Neural Collapse geometry as post-hoc scoring mechanisms, ASL is a training-based method that learns an angularly-separated feature space as its primary objective.

---

> ### Author Response · Authors · 2025-11-17
>
> > Q6: Norm as signal vs. hypothesis.
>
> Using feature norm as an OOD score is a valid and common practice. Our core argument aims to reveal that in Mahalanobis distance scoring, feature norm can mislead OOD detection. Indeed, for well-trained models, feature norm can also serve as a reasonably good OOD score. To clarify this point, we briefly compare the performance of using only the feature norm as the score. As shown in Table B, under ASL training (where we compute the feature norm before feature normalization), using only the feature norm can also achieve reasonably good performance; whereas under standard CE training, the discriminative power of the feature norm drops significantly. This precisely explains why some works focus on optimizing or circumventing the interference of feature norms, or on enhancing the discriminativity of the feature norm itself.
>
> Table B: OOD Detection Performance using Feature Norm as the Score
> | Method           | CIFAR-10 |||| CIFAR-100 ||||
> |:------------------|:-----:|:-----|:-----:|:-----|:-----:|:-----|:-----:|:-----|
> |                  | Near || Far || Near || Far |
> |                  | FP↓  | AU↑  | FP↓   | AU↑  | FP↓   | AU↑  | FP↓   | AU↑  |
> | CE              | 90.9 | 67.3 | 67.5 | 78.7 | 74.9 |  69.6 | 77.7 | 66.0 |
> | ASL             | **28.8** | **92.3** | **22.0** | **94.5** | **64.2** | **77.3** | **50.5** | **83.3** |
>
>
> > Q7: Intuition around Eq. (1).
>
> The discussion around Equation (1) aims to illustrate how the feature geometric structure optimized by cross-entropy (CE) loss degrades the performance of MDS. Our reference to "MDS sensitivity to feature norms" is intended to characterize the behavior of such poorly structured feature spaces. Specifically, as shown in Fig. 13, CE typically minimizes loss by increasing feature norms, which results in large variance in feature norms, and this is also why MD++ outperforms MDS. Ideally, CE should generate features centered around class means with minimal intra-class dispersion; however, in practice, this ideal state is rarely achievable, which is why the OOD field has continuously developed methods to address this flaw.
>
> ASL mitigates this issue by applying $\ell_2$-normalization during training, which shifts the optimization focus toward angular separation. ASL rapidly stabilizes feature norms and reduces their variance (Fig. 13), ultimately leading to a well-structured feature space (Fig. 12).
>
>
> > Q8: Training efficiency & confidence intervals
>
> We have added confidence intervals (mean ± standard deviation) for key experiments in the appendix. For training efficiency, we report the average time across multiple epochs with unified hyperparameters, especially batch size (set to 128). Compared to LogitNorm, ASL removes the bias term from the linear layer, simplifying the structure, this is why ASL achieves faster training.
>
>
> > Q9: Figure 3 is aesthetically nice but uses large space
>
> Thank you for your valuable advice. We have modified the horizontal layout in the revised version to reduce space usage.

---

> ### Author Response · Authors · 2025-11-17
>
> > Q10: Imbalanced classes
>
> In class-imbalanced settings, the standard Cross-Entropy (CE) model indeed encodes class prior information into both weight norms and feature norms, and ASL effectively removes the class prior information from the feature norms, thereby confining it only to the weight norms.
> As experimental results on CIFAR-10-LT and CIFAR-100-LT demonstrate (see Tables C and D below), where the imbalance ratio $\rho$ is defined as the number of samples in the largest class divided by that in the smallest class, this characteristic of decoupling the prior from the feature representation allows ASL to achieve strong and comparable performance across all imbalance ratios ($\rho$=10 and $\rho$=100). The advantage is particularly pronounced for the most challenging Near-OOD detection under the more extreme $\rho$=100 setting.
>
> Table C: OOD Detection Performance on CIFAR-10-LT.
> | Method           | $\rho = 10$ |||| $\rho = 100$ ||||
> |:------------------|:-----:|:-----|:-----:|:-----|:-----:|:-----|:-----:|:-----|
> |                  | Near || Far || Near || Far |
> |                  | FP↓  | AU↑  | FP↓   | AU↑  | FP↓   | AU↑  | FP↓   | AU↑  |
> | CE               | 57.2 | 80.0 | 41.4  | 87.5 | 71.7  | 74.7 | 51.1  | 82.6 |
> | PALM             | 84.4 | 75.6 | 46.9  | 87.2 | 94.2  | 55.1 | 80.0  | 65.3 |
> | T2FNorm          | 37.8 | 88.5 | 23.8  | 94.4 | 62.6  | 78.9 | 42.7  | 88.0 |
> | cosine Classifier| 37.7 | **89.0** | 26.7  | 93.2 | 65.2  | 79.6 | 51.6  | 88.0 |
> | ASL              | **37.6** | 88.8 | **21.6**  | **95.0** | **52.9**  | **82.8** | **33.6**  | **91.4** |
>
> Table D: OOD Detection Performance on CIFAR-100-LT.
> | Method           | $\rho = 10$ |||| $\rho = 100$ ||||
> |:------------------|:-----:|:-----|:-----:|:-----|:-----:|:-----|:-----:|:-----|
> |                  | Near || Far || Near || Far |
> |                  | FP↓  | AU↑  | FP↓   | AU↑  | FP↓   | AU↑  | FP↓   | AU↑  |
> | CE               | 83.4 | 62.4 | 72.2  | 72.0 | 88.6  | 56.9 | 72.7  | 69.9 |
> | PALM             | 82.8 | 64.4 | **43.4**  | **86.8** | 90.6  | 57.7 | 71.9  | **74.3** |
> | T2FNorm          | 71.7 | 72.6 | 61.4  | 76.3 | 81.8  | 64.5 | 67.6  | 67.8 |
> | Cosine Classifier| 68.7 | 73.0 | 62.7  | 76.3 | 80.8  | 64.8 | 76.0  | 67.7 |
> | ASL              | **67.5** | **74.2** | 62.6  | 77.2 | **79.2**  | **66.6** | **62.5**  | 74.0 |
>
>
> > Q11: Table 3 (CIFAR-100): PALM has the lowest FPR, yet both ASL and PALM are bolded
>
> Thank you for your valuable advice. We have corrected this typo in the revised version.
>
> > Q12: Combined with activation shaping
>
> We conducted experiments on combining ASL with activation shaping methods (ReAct, ASH), and the results are shown in Table E. The results indicate that ASL can be combined with activation shaping methods such as ReAct and ASH to effectively improve overall OOD detection performance on CIFAR-10, as well as Far-OOD performance on CIFAR-100 and Imagenet-200.
>
> Table E: OOD Detection Performance of ReAct/ASH Combined with ASL
> | Method | CIFAR-10 |||| CIFAR-100 |||| IN-200 ||||
> |:------------|:-----:|:-----|:-----:|:-----|:-----:|:-----|:-----:|:-----|:-----:|:-----|:-----:|:-----|
> |            | Near || Far || Near || Far || Near || Far |
> |            | FP↓ | AU↑ | FP↓ | AU↑ | FP↓ | AU↑ | FP↓ | AU↑ | FP↓ | AU↑ | FP↓ | AU↑ |
> | ReAct | 62.0 | 86.7 | 42.0 | 91.1 | 58.8 | 80.2 | 53.6 | 80.6 | 65.3 | 80.4 | 27.7 | 93.0 |
> | +ASL | 25.4 | 93.3 | 13.5 | 96.7 | 65.5 | 77.8 | 46.9 | 85.1 | 66.3 | 80.2 | 25.4 | 93.7 |
> | ASH | 89.2 | 73.8 | 76.9 | 77.9 | 65.2 | 78.4 | 60.4 | 79.5 | 65.5 | 82.1 | 26.4 | 94.2 |
> | +ASL | 27.4 | 92.8 | 12.7 | 97.1 | 64.8 | 77.1 | 48.5 | 85.3 | 55.5 | 82.8 | 24.5 | 93.6 |
>
> [1] Julian Bitterwolf, Maximilian Mueller, and Matthias Hein. In or out? fixing imagenet out-ofdistribution detection evaluation. arXiv preprint arXiv:2306.00826, 2023.
>
> [2] Berker Demirel, Marco Fumero, and Francesco Locatello. OOD detection with relative angles. In The Thirty-ninth Annual Conference on Neural Information Processing Systems, 2025.
>
> [3] Litian Liu and Yao Qin. Detecting out-of-distribution through the lens of neural collapse. In Proceedings of the Computer Vision and Pattern Recognition Conference, pp. 15424–15433, 2025.
>
> [4] Mouïn Ben Ammar, Nacim Belkhir, Sebastian Popescu, Antoine Manzanera, and Gianni Franchi. NECO: NEural collapse based out-of-distribution detection. In The Twelfth International Conference on Learning Representations, 2024.

---

> > ### Comment · Reviewer_puLH · 2025-11-22
> >
> > I would like to thank the authors for their effort answering my questions. I think the authors did a really good job executing the actionable items and most of my concerns are addressed. You can see my follow up remarks below.
> >
> > > Q2: Figure 1
> >
> > Thank you so much for the clarification. I believe caption should stress that 4x4 grid is split keeping the number of samples equal over bins.
> >
> > > Q3: Motivation vs. cosine head.
> >
> > I appreciate the new experiments. The performance of cosine classifier closely follows ASL and the benefits that can be attributed to ASL are mostly from training dynamics. Including cosine head's ImageNet performance can further support the claims.
> >
> > > Q6: Norm as signal vs. hypothesis.
> >
> > My main point was "feature norm does not come free as a ID/OOD separating factor". The native CE trained models do not show a significant separation as seen in new experiments Figures 15 and 16. It is more about the writing style and strategy as your model wants to discard norm effect while cited work claims otherwise. I understand that the authors' take (and MD++) applies when mahalanobis score is used but when there are plenty of works suggesting the same, it would flow more naturally as a starting point. Yet, I leave the decision to the authors as it was not a central concern.
> >
> > > Q7: Intuition around Eq. (1).
> >
> > I agree that the ideal case is often not practically achieved but the arguments in the paragraph are not given the correct way. It is presenting the different parts of the loss as contradictive. When the objective is minimized the loss does not internally contradict with MDS score but the problems are arising from practical scenarios (optimizing one term is apparently easier than the other and this is the main flaw of plain CE models). It is mainly a presentation issue.
> >
> >
> > Finally, since the potential explanation boils down to the training dynamics (if the authors agree), along the training trajectory saving multiple checkpoints of the models and computing their OOD performance as well as ID accuracy (for instance CE vs ASL) would be insightful.

---

> > > ### Author Response · Authors · 2025-11-24
> > >
> > > Dear Reviewer,
> > >
> > > We sincerely appreciate your valuable comments! Below are our detailed responses to address your remaining concerns.
> > >
> > > > Q2: Figure 1
> > >
> > > Thank you for the clarification. You are correct that the grid partitions are designed based on an equal-sample strategy.
> > >
> > > Due to the correlation between Feature Norm and Cosine Similarity, strict equality across all 2D bins is geometrically infeasible. Therefore, we applied the "equal sample" constraint to the marginal distributions (using quartiles) for the axis thresholds. This ensures that each row and column contains 25% of the data, effectively highlighting the distributional properties.
> > >
> > > > Q3: Motivation vs. Cosine Classifier
> > >
> > > We appreciate the suggestion to include ImageNet results to further support our claims. We have conducted additional experiments on ImageNet-200 comparing ASL with the Cosine Classifier on Table A.
> > >
> > > Table A: Comparison of ASL with Cosine Classifier for OOD Detection on ImageNet-200
> > >
> > > | Method | Near-OOD || Far-OOD ||
> > > |:---|:---:|:---:|:---:|:---:|
> > > | | FP↓ |AU↑ | FP↓ | AU↑ |
> > > | CE | 59.3 | 82.8 | 24.4 | 94.0 |
> > > | Cosine Classifier | 55.1 | 84.3 | 23.9 | 93.9 |
> > > | ASL (Ours) | **52.4** | **85.1** | **20.4** | **95.0** |
> > >
> > > > Q6: Norm as signal vs. hypothesis
> > >
> > > We sincerely appreciate your thoughtful suggestion regarding the narrative flow. In the final version, we will refine the introduction and related work to present a more balanced view. We will first acknowledge existing literature that leverages feature norm as a signal, and then clarify that without specific constraints (like ASL), it remains unreliable in standard CE training due to optimization shortcuts. This will provide a smoother narrative flow.
> > >
> > > > Q7: Intuition around Eq. (1)
> > >
> > > We understand your point that theoretically, strictly minimizing the CE objective implies optimizing both angular alignment and magnitude, so the terms are not mathematically "contradictory" in the objective function itself. Our intended argument, which aligns with your observation about "practical scenarios," is that the optimization dynamics create an imbalance.
> > > In the final revision, we will rewrite the relevant paragraph in Section 3.2  to explicitly frame this as an "optimization bias" rather than a theoretical conflict, ensuring the precise interpretation of the loss function.
> > >
> > > We also greatly appreciate your suggestion regarding the analysis of training dynamics. Tracking the evolution of ID accuracy and OOD performance along the training trajectory indeed provides deeper insight into the model's behavior. We will include a figure illustrating these dynamics in the Appendix of the final version.
> > >
> > > Best regards

---

> > > > ### Comment · Reviewer_puLH · 2025-11-25
> > > >
> > > > I would like to thank the authors for providing additional insights.
> > > >
> > > > If the authors could at least show a preliminary table on the performance of a few checkpoints when ASL and CE training used, and evaluate their OOD performance I will update my score accordingly.

---

> > > > > ### Author Response · Authors · 2025-11-26
> > > > >
> > > > > We sincerely appreciate your constructive suggestions. Throughout the training process, we monitored both In-Distribution (ID) accuracy and Out-of-Distribution (OOD) detection performance at intervals of 10 epochs. Table 1 below details the performance metrics for these checkpoints on CIFAR-10.
> > > > >
> > > > > Table 1: Training dynamics comparison on CIFAR-10. (Values are presented as "CE / ASL (Ours)")
> > > > >
> > > > > | Epoch | ID Acc↑ | Near-OOD AU↑ | Far-OOD AU↑ |
> > > > > | :---: | :---: | :---: | :---: |
> > > > > | 10 | 75.4 / 80.9 | 71.3 / 77.7 | 79.4 / 90.2 |
> > > > > | 20 | 81.4 / 77.8 | 77.7 / 76.7 | 84.5 / 90.6 |
> > > > > | 30 | 84.6 / 83.4 | 79.7 / 83.3 | 86.3 / 92.7 |
> > > > > | 40 | 85.4 / 85.6 | 79.1 / 80.4 | 85.9 / 90.8 |
> > > > > | 50 | 87.9 / 88.1 | 83.0 / 85.4 | 88.3 / 93.0 |
> > > > > | 60 | 90.7 / 90.9 | 82.3 / 88.9 | 88.7 / 96.1 |
> > > > > | 70 | 91.6 / 91.9 | 83.7 / 89.8 | 89.4 / 95.8 |
> > > > > | 80 | 93.3 / 93.7 | 85.4 / 91.8 | 89.4 / 96.8 |
> > > > > | 90 | 94.9 / 94.9 | 88.3 / 92.9 | 91.0 / 97.1 |
> > > > > | 100 | 95.1 / 94.9 | 88.8 / 93.2 | 91.1 / 97.2 |
> > > > >
> > > > >
> > > > > To visualize these dynamics intuitively, we have added Figure 19 (CIFAR-10) and Figure 20 (CIFAR-100) in the revised Appendix E.2. We have incorporated the discussions regarding Related Work and Section 3.2 into the revised version.
> > > > >
> > > > > Thank you again for your valuable feedback, which has significantly strengthened our paper.

---

> > > > > > ### Comment · Reviewer_puLH · 2025-11-26
> > > > > >
> > > > > > To me, these results are somewhat surprising. Typically, one would expect CE models to show an AUROC curve that first increases and then degrades in later epochs, given their tendency toward overconfidence as training progresses.
> > > > > >
> > > > > > However, this pattern does not appear here. Possibly due to the regularization choices or simply because CIFAR as a toy dataset may not fully expose this behavior.
> > > > > >
> > > > > > I thank the authors once again for providing these detailed dynamics, and I am updating my score accordingly.

---

> > > > > > > ### Author Response · Authors · 2025-11-26
> > > > > > >
> > > > > > > We sincerely thank you for your time in reviewing our response and for increasing your score.
> > > > > > >
> > > > > > > Regarding the training dynamics, we agree that the regularization choices likely contribute to the observed stability. As illustrated in Fig. 18 of our paper, the mean of feature norms for the CE model increases during the early training stages. We observe that as training progresses, the variance of the feature norms tends to decrease or stabilize in the later stages. Consequently, the impact of optimization on the final model is primarily reflected in angular separation.

---

### Official Review · Reviewer_zrv2 · 2025-10-22

**Soundness:** 3
**Presentation:** 3
**Contribution:** 2
**Rating:** 4
**Confidence:** 4

**Summary:**

This paper proposes Angular Separation Learning (ASL), a simple modification to standard classification training for Out-of-Distribution (OOD) detection. By applying normalization to feature vectors before the final linear layer, the method enforces an angular decision geometry that alleviates the well-known sensitivity of distance-based OOD detectors (in particular of Mahalanobis detection score) to feature magnitude. The authors demonstrate that this minimalist change yields state-of-the-art or superior performance on several OOD benchmarks, especially for near-OOD cases. The paper argues that small architectural choices can achieve competitive OOD robustness without added complexity.

**Strengths:**

1.	The method is extremely lightweight and well-motivated by a geometric analysis.
2.	ASL consistently matches or outperforms more complex OOD detection approaches, particularly in the challenging near-OOD regime.
3.	The work prompts a healthy reconsideration of whether the field’s increasing methodological complexity is always justified.
4.	The method aligns the loss used during training with the score employed for InD/OoD separation.

**Weaknesses:**

1.	The idea of feature normalization and angular margin learning is not new (similar principles exist in SphereFace for example). The main novelty lies in reinterpreting and applying them to OOD detection.
2.	The method shows clear improvements in near-OOD detection; however, a more thorough analysis is needed to assess its effectiveness on far-OOD scenarios and its impact on overall classification accuracy.
3.	The impact of normalization on confidence calibration and threshold is not discussed.

**Questions:**

1.	Angular separation learning has been previously explored in the literature, notably in *SphereFace: Deep Hypersphere Embedding for Face Recognition* (Liu et al., CVPR 2017). Although SphereFace was not designed for OOD detection, its underlying rationale is closely related, as it also enforces angular-based feature learning on a hypersphere and discusses the Mahalanobis distance for classification. SphereFace should therefore be cited and discussed in relation to the proposed method. In particular, the loss functions of ASL and SphereFace should be explicitly compared and analyzed, as both promote hyperspherical feature learning but differ in how they enforce angular separation (standard normalized cross-entropy vs. multiplicative angular-margin loss).

2.	The mechanics of MDS failure are well-motivated for near-OOD samples. However, losing access to feature norm information may negatively impact both classification accuracy and far-OOD detection. While the accuracies of models trained on CIFAR-10 and CIFAR-100 are reported in the appendix, they should also be clearly mentioned alongside all experimental results, especially those presented in the main body of the paper. Regarding far-OoD detection, the results presented in Fig. 4 show a strong instability with respect to parameter $\lambda$ (on individual datasets) meaning that the method could exhibit poor performance in some particular situations. Could you elaborate on the strategy to choose the value of $\lambda$ that should not be guided by experimental results on particular datasets ?

3.	Could you provide an analysis of the features distribution to visually verify the claimed hyperspherical geometry and the relative positioning of InD and OOD samples?

4.	The estimation of $\mu_c$ and $\Sigma$ is a key factor in the effectiveness of MDS. Could you clarify how these estimates were computed (e.g., number of samples, and whether training, validation, or test data were used)?

5.	Neural Collapse (mentionned in the paper) is typically observed for in-distribution features, where class features collapse toward their centroids on a hypersphere. Given that Angular Separation Learning enforces $\ell_2$-normalization, did you observe a similar collapse behavior around class centroids for in-distribution samples, and how does this interact with the positioning of out-of-distribution features?

Angular Separation Learning (ASL) is a lightweight and well-motivated approach that demonstrates strong empirical performance, particularly for near-OOD detection. While the method is effective and clearly presented, its novelty is limited, with prior work (e.g., SphereFace) exploring similar angular feature normalization. The analysis of far-OOD performance, feature-space visualization, and hyperparameter sensitivity is incomplete. Overall, the paper provides a valuable incremental contribution to OOD detection, but it primarily reinforces existing intuitions rather than introducing a fundamentally new methodology.

---

> ### Author Response · Authors · 2025-11-17
>
> We sincerely appreciate the time and effort put into reviewing our paper and providing valuable feedback. We would like to address your questions below.
>
> > Q1: Relationship with SphereFace and Comparative Analysis
>
> We have cited SphereFace[1] in the revised version and added a comparative analysis in Appendix E.9. While SphereFace introduces a multiplicative angular margin to enhance inter-class separation, our experiments demonstrate that directly applying it to OOD detection leads to performance degradation (as shown in Table A, SphereFace even underperforms the CE baseline on most metrics). Similarly, the Cosine Classifier, despite also using angular learning, is outperformed by ASL because our weight decay mechanism on unconstrained weights provides an adaptive inverse temperature, enabling finer-grained angular tuning for OOD samples. In contrast, ASL, through its minimalist $\ell_2$-normalization design, focuses more on improving angular discriminability required for OOD detection rather than classification margin, thereby achieving superior performance.
>
> Table A: Comparison of ASL with Cosine Classifier and SphereFace for OOD Detection
> | Method           | CIFAR-10 |||| CIFAR-100 ||||
> |:------------------|:-----:|:-----|:-----:|:-----|:-----:|:-----|:-----:|:-----|
> |                  | Near || Far || Near || Far |
> |                  | FP↓  | AU↑  | FP↓   | AU↑  | FP↓   | AU↑  | FP↓   | AU↑  |
> | CE               | 40.2       | 88.8       | 30.6       | 91.0       | 67.8       | 74.8       | 51.9       | 82.6       |
> | SpereFace        | 46.1       | 88.3       | 27.9       | 92.5       | 73.4       | 72.4       | 68.6       | 69.7       |
> | Cosine Classifier| 26.6       | 93.0       | 15.6       | 96.3       | 57.7       | 80.0       | 54.4       | 81.1       |
> | ASL              | **25.1**       | **93.3**       | **12.0**       | **97.2**       | **56.2**       | **80.9**       | **47.8**       | **85.0**       |
>
> > Q2: ID accuracy, Far-OOD detection, and $\lambda$ parameter selection strategy
>
> 1. Supplementing ID Classification Accuracy in the Main Text
>
> We have added the accuracy to the main paper in the revised version.
>
> 2. Response to Concerns About Feature Norm Abandonment
>
> Classification Accuracy: As shown in Appendix Table 14 and revised main paper, ASL maintains competitive ID classification performance.
>
> Far-OOD Detection: Our extensive experiments demonstrate that ASL does not compromise far-OOD performance, in fact, it enhances it. On CIFAR-10, performance remains stable across the entire $\lambda$ range. In contrast, CIFAR-100 contains 100 fine-grained categories that require learning multi-granular feature representations at both coarse and fine levels, posing a significant challenge for OOD detection. The weight decay $\lambda$ directly controls the "temperature" of angular contrastive learning, and this regulation is inherently non-linear—different $\lambda$ values optimize feature learning at different granularities, leading to non-linear performance characteristics. It is important to emphasize that, despite these fluctuations, ASL consistently outperforms the CE baseline at all $\lambda$ values.
>
> 3. Hyperparameter $\lambda$
>
> We set $\lambda = 5 \times 10^{-4}$ as the default value and uniformly applied it across all experimental datasets (CIFAR-10, CIFAR-100, ImageNet-200, ImageNet, and BIMCV). It consistently achieved stable out-of-distribution (OOD) detection performance, verifying its broad applicability. There is no need to adjust this parameter according to dataset types (natural images/medical images) or scales (small-sample CIFAR/large-scale ImageNet), and it can well adapt to different scenarios.

---

> > ### Comment · Reviewer_zrv2 · 2025-11-21
> >
> > The authors added a citation and a quantitative comparison, which is appreciated. However, the conceptual discussion is still shallow.
> > My question asked for an explanation of why SphereFace—despite similar angular geometry—performs poorly for OOD and a clear argument for what ASL contributes beyond existing angular-normalization methods.
> > The current response mainly states that “SphereFace degrades performance,” but does not provide: theoretical intuition, geometric reasoning or a principled explanation of the differences in optimization dynamics.

---

> > > ### Author Response · Authors · 2025-11-24
> > >
> > > Dear Reviewer,
> > >
> > > We sincerely appreciate your valuable comments!
> > >
> > > **Quantitative Evidence of Hyperspherical Geometry & Normalization Effect**: Our claim that ASL enforces hyperspherical discriminability is explicitly quantified by the sample distribution densities shown in Fig. 1 (Page 2), these heatmaps serve as 2D histograms.
> > > Fig. 1(a): Raw Features.
> > > Fig. 1(b): Applying normalization (MD++) fails to achieve effective angular separation. While it constrains the feature norm, it does not force ID samples to cluster tightly in the angular dimension.
> > > Fig. 1(c): ASL achieves effective separation between ID and OOD samples in the feature space by explicitly optimizing angular discriminability.
> > >
> > > **SphereFace** enforces a fixed, multiplicative angular margin ($m\theta_{y}$). This imposes a rigid geometric constraint that forces the model to over-compress easy samples while potentially distorting the feature manifold for hard samples to satisfy the margin, harming the global distribution structure needed for OOD detection. In contrast, ASL avoids artificial, rigid margins. It utilizes $\ell_2$ normalization and weight decay to establish a dynamic temperature regulation mechanism, while allowing for appropriate intra-class variance.
> > >
> > > Best regards

---

> > > > ### Author Response · Authors · 2025-11-26
> > > >
> > > > Thank you again for your insightful comment regarding the conceptual comparison between ASL and SphereFace.
> > > >
> > > > We have updated the paper to include a new visualization and analysis in Appendix E.9 (Fig. 23).
> > > >
> > > > Our results show that SphereFace's rigid multiplicative margin induces two pathological behaviors that degrade OOD detection performance:
> > > >
> > > > 1.  Feature Collapse: It compresses ID samples into narrow cones, reducing the angular margin available for rejecting OOD samples (especially in CIFAR-10).
> > > > 2.  Covariance Collapse: The extreme angular compression minimizes intra-class variance, leading to an ill-conditioned precision matrix ($\Sigma^{-1}$), which acts as an unstable amplifier in the Mahalanobis distance calculation. ID samples are consequently projected to pathologically high norms.
> > > >
> > > > We sincerely appreciate the time you have dedicated to reviewing our work and helping us improve its quality.

---

> ### Author Response · Authors · 2025-11-17
>
> > Q3: Neural Collapse Phenomenon and Feature Distribution Visualization Analysis
>
> ASL's $\ell_2$-normalization simultaneously constructs a hyperspherical feature space and exhibits class-center clustering behavior consistent with Neural Collapse, jointly enhancing ID/OOD separation capability.
>
> Feature Distribution Visualization.
> We have added UMAP visualizations in Figs. 13 (CIFAR-10) and 14 (CIFAR-100) in the revised version. ASL’s feature space confirms the claimed hyperspherical geometry: $\ell_2$-normalization leads to compact, direction-aligned ID clusters.
> Notably, ASL achieves better ID/OOD separation: OOD samples lie more in gaps between ID clusters (especially for CIFAR-100). On CIFAR-10, ASL shows structured geometry similar to PALM (contrastive learning), validating its implicit contrastive property. These visuals validate ASL’s angular optimization.
>
> Neural Collapse.
> Under ASL training, the clustering of ID features aligns with Neural Collapse—ID samples cluster compactly toward class centers. This property directly enhances OOD detection performance: the tight clustering of ID features leaves obvious low-density regions on the hypersphere, and OOD samples naturally fall into these gaps due to angular deviation, thereby facilitating effective ID/OOD separation.
>
> > Q4: The estimation of $\mu_c$ and $\Sigma$ in MDS.
>
> The estimation of class means $\mu_c$ and the shared covariance matrix $\Sigma$ is as follows:
>
> $\mu_c = \frac{1}{N_c} \sum_{\substack{(x_i, y_i) \in D_{in} , y_i = c}} f(x_i)$, where $N_c$ is the number of training samples in class $c$, and $f(x_i)$ is the feature embedding of $x_i$.
>
> $\Sigma = \frac{1}{N-1} \sum_{\substack{(x_i, y_i) \in D_{in} , y_i = c}} (f(x_i)-\mu_c)(f(x_i)-\mu_c)^\top$, where $N = \sum_{c=1}^K N_c$ is the total number of training samples.
>
> No test set data is involved in these estimations to avoid information leakage.
>
> > Q5: The impact of normalization on confidence calibration and threshold is not discussed.
>
> 1. Confidence Calibration：
> In out-of-distribution (OOD) detection, the primary focus is on the separability between in-distribution (ID) and OOD samples. Regarding confidence calibration, we present the results in Table A, where "TS" indicates whether Temperature Scaling is used. We adopt the optimal temperature value searched on the validation set (searched independently for each method). After applying Temperature Scaling, ASL maintains favorable confidence calibration metrics.
>
> Table A: Confidence Calibration Metrics of Different Methods
> | Dataset   |        | Cross Entropy | LogitNorm | ASL (ours) |
> | :-------- | :----- | :-----------: | :-------: | :--------: |
> | CIFAR-10  | w/o TS | 2.76          | 59.3      | 2.84       |
> | CIFAR-10  | w/ TS  | 0.98          | 0.82      | 0.81       |
> | CIFAR-100 | w/o TS | 6.20          | 74.1      | 11.4       |
> | CIFAR-100 | w/ TS  | 3.32          | 21.0      | 3.07       |
>
>
> 2. Threshold:
> As shown in Fig. 10-12 in the appendix, for ASL, the ID distribution forms a sharp, high peak, resulting in clearer separation between the two peaks; this provides a more robust decision threshold. In our experiments, the lower FPR95 values signify that when 95% of ID samples are correctly identified, fewer OOD samples are falsely accepted. This demonstrates that ASL remains effective with a threshold fixed at this standard operating point (95% TPR).
>
> [1] Weiyang Liu, Yandong Wen, Zhiding Yu, Ming Li, Bhiksha Raj, and Le Song. Sphereface: Deep hypersphere embedding for face recognition. In Proceedings of the IEEE conference on computer vision and pattern recognition, pp. 212–220, 2017.

---

> > ### Comment · Reviewer_zrv2 · 2025-11-21
> >
> > My question asked for evidence supporting: the claimed hyperspherical geometry, the relative positioning of ID and OOD samples, the effect of normalization on separability.
> > The response provides descriptive commentary rather than measurable or quantitative evidence. The visualizations help, but they do not constitute an analysis.

---

### Official Review · Reviewer_RM6N · 2025-11-03

**Soundness:** 2
**Presentation:** 2
**Contribution:** 2
**Rating:** 4
**Confidence:** 3

**Summary:**

This paper revisits the foundation of out-of-distribution (OOD) detection and identifies a critical weakness of Mahalanobis-based methods — their sensitivity to feature norms. The authors propose Angular Separation Learning (ASL), a minimalist yet effective training strategy that applies $\ell_2$-normalization to features before the final classification layer. This simple modification encourages angular discrimination between classes, effectively mitigating the failure mode of distance-based detectors.

**Strengths:**

1. The proposed Angular Separation Learning (ASL) involves only a minimal modification—$\ell_2$ normalization of features before the cross-entropy loss—yet consistently surpasses much more complex training schemes (contrastive, synthetic outlier generation, or regularization-heavy methods). This reinforces the key message that complexity is not always correlated with OOD robustness, positioning ASL as a strong and practical new baseline.

2. The paper provides a unifying perspective connecting normalization, angular margin optimization, and contrastive representation learning. Proposition 1 formally shows that ASL implicitly performs prototype-based contrastive learning in angular space, without explicit pair mining or temperature tuning. This theoretical lens could inspire a rethinking of numerous normalization and contrastive frameworks.

3. The experiments are extensive, covering both convolutional and transformer backbones, and include near-/far-OOD as well as covariate shift scenarios.

**Weaknesses:**

1. The novelty of the proposed method appears somewhat limited, as normalization-based mechanisms for OOD robustness have already been explored in prior works such as LogitNorm (Wei et al., 2022), T2FNorm (Regmi et al., 2024a), and MD++ (Müller & Hein, 2025). It would strengthen the paper if the authors could further highlight the unique aspects and contributions of their approach compared to these existing methods.

2. Proposition 1 provides a qualitative statement about angular separation, but its proof sketch is not rigorous and lacks an explicit link to OOD generalization.

3. In the proposed ASL framework, features are projected onto the angular space via $\ell_2$-normalization before classification. Could the authors further clarify why discarding the feature magnitude and relying solely on angular information is beneficial for OOD detection?

**Questions:**

Please carefully check the weaknesses.

---

> ### Author Response · Authors · 2025-11-17
>
> We sincerely appreciate the time and effort put into reviewing our paper and providing valuable feedback. We would like to address your questions below.
>
> > Q1: Limited Novelty
>
> The main contributions of our paper include: 1) an in-depth analysis of the limitations of feature magnitude in distance-based OOD detection; 2) a systematic formulation and theoretical demonstration that applying cross-entropy to normalized features enables implicit prototype-based contrastive learning in the angular space, along with revealing the optimization dynamics of this process; and 3) a simple yet effective training framework that achieves robust experimental results.
>
> Regarding prior works: LogitNorm alleviates the overconfidence issue of OOD samples by normalizing output logits, with its core motivation being the regularization of the output space. Building on LogitNorm, T2FNorm enhances the discriminability of logits during testing. As a post-hoc method, MD++ yields limited effectiveness with mere late-stage normalization if the angular structure of the original feature space is poor. Unlike these works, our primary focus is on the role of feature normalization in shaping the feature space during training, which allows us to obtain a more discriminative feature space.
>
> > Q2: Proposition 1 Rigor
>
> Thank you for your valuable comments. We hereby provide the following clarifications and additions:
>
> 1. Positioning of Proposition 1: Characterizing Optimization Dynamics
>
> The core purpose of Proposition 1 is to qualitatively elucidate the optimization dynamics during ASL training, namely: the model is forced to optimize solely in the angular space. This process naturally induces an angular space structure characterized by intra-class compactness and inter-class separation.
>
> Regarding the specific representation of the final model's feature space, many works have conducted in-depth studies, such as [1]. Our optimization function can be viewed as a special form of these (with $\|H\|_F=1, b=0$). Utilizing the symmetry assumptions and convex analysis therein, similar conclusions can be drawn: at the global optimum, $H$ satisfies a simplex ETF structure. However, in practical training, models often struggle to reach the theoretical global optimum. Therefore, we focus more on how the optimization process itself shapes the feature geometry.
>
>
> 2. Direct Link Between Angular Space Optimization and OOD Generalization
>
> The standard cross-entropy (CE) loss can increase confidence by unrestrictively increasing feature norms, but this leads to a  loose feature geometric structure and high sensitivity to feature norm anomalies (especially low-norm OOD samples). ASL forces the model, under the same training error, to learn a more compact and separable angular representation.
>
> Our experimental results also demonstrate that: The feature space learned by ASL exhibits stronger angular separation, making ID and OOD samples easier to distinguish in the angular space; and it reduces misjudgments of low-norm OOD samples (Fig. 15, 16, 17 in the revised version).
>
>
>
> > Q3: Why Discarding Feature Norms is Beneficial
>
> We project features into the angular space via $\ell_2$-normalization, with the primary goal of addressing the sensitivity of distance-based OOD detectors (such as MDS) to feature magnitude. Specifically, our analysis shows that under standard cross-entropy training, variations in feature magnitude can severely interfere with the judgment of distance-based scores like MDS.
>
> Through ASL training, we achieve benefits at two levels:
>
> 1. It learns a highly discriminative feature space. This directly improves the performance of angle-based scores (such as cosine similarity).
>
> 2. It reshapes the distribution of feature magnitudes. We can briefly compare the performance of using only feature norm as the score, as shown in Table A. Under ASL training, using only feature normalization also achieves strong performance, whereas under standard CE training, the discriminative ability of feature norm decreases significantly.
>
> Table A: OOD Detection Performance using Feature Norm as the Score
> | Method           | CIFAR-10 |||| CIFAR-100 ||||
> |:------------------|:-----:|:-----|:-----:|:-----|:-----:|:-----|:-----:|:-----|
> |                  | Near || Far || Near || Far |
> |                  | FP↓  | AU↑  | FP↓   | AU↑  | FP↓   | AU↑  | FP↓   | AU↑  |
> | CE              | 90.9 | 67.3 | 67.5 | 78.7 | 74.9 |  69.6 | 77.7 | 66.0 |
> | ASL             | **28.8** | **92.3** | **22.0** | **94.5** | **64.2** | **77.3** | **50.5** | **83.3** |
>
> [1] Zhu Z, Ding T, Zhou J, et al. A geometric analysis of neural collapse with unconstrained features[J]. Advances in Neural Information Processing Systems, 2021, 34: 29820-29834.

---

> > ### Comment · Reviewer_RM6N · 2025-11-21
> >
> > Thank you for the additional experiments and the extended explanations. I still feel that the response does not fully clarify the paper’s conceptual contribution. In particular, the explanation of the link between angular-space optimization and OOD generalization remains mostly empirical. I would encourage the authors to provide a deeper theoretical or mechanistic analysis.

---

> > > ### Author Response · Authors · 2025-11-24
> > >
> > > Dear Reviewer,
> > >
> > > We sincerely appreciate your valuable comments! We agree that a deeper mechanistic explanation is crucial. To this end, we clarify that our method is grounded in a clear mechanistic analysis of feature space dynamics and gradient optimization.
> > >
> > > We address the theoretical link between Angular Separation Learning (ASL) and OOD generalization from three mechanistic perspectives:
> > >
> > > 1. Mechanistic Constraint via Gradient Orthogonality
> > >
> > > The "lazy shortcut" of increasing feature norms is not merely an empirical observation but a direct consequence of the standard cross-entropy gradient. This gradient contains a radial component that drives features to grow in magnitude until the softmax function saturates. In contrast, the gradient of the ASL loss with respect to feature $f$, denoted as $\frac{\partial \mathcal{L}_{CE}}{\partial f}$, is strictly orthogonal to the feature vector $\tilde{f}$. This mathematically guarantees that the radial component of the optimization trajectory is zero.
> > >
> > > 2. Elimination of the "Norm Confounding" Failure Mode
> > >
> > > ASL provides a structural solution: by enforcing constant norms ($||\tilde{f}||_2 = s$) during both training and inference, we mathematically eliminate the variable responsible for this failure mode. This ensures that the detection score is derived solely from angular alignment, which is structurally robust to the "norm shrinkage" typical of OOD features.
> > >
> > > 3. Dynamic Regularization Preventing Feature Collapse
> > >
> > > We observe that excessive constraints (e.g., enforcing overly strict intra-class angular alignment) can disrupt the feature structure, causing all features (including both ID and OOD) to collapse into an overly narrow space, thereby reducing the separability necessary to distinguish them. We address this mechanistically via the weight decay hyperparameter $\lambda$. By tuning $\lambda$, we effectively regulate the "sharpness" of the distribution. This prevents the model from collapsing features into an overly narrow space while maintaining non-vanishing gradients that force the model to learn finer-grained angular distinctions, which is critical for distinguishing near-OOD samples.
> > >
> > > In this way, we achieve an effective balance through a natural improvement to the standard Cross-Entropy objective.
> > >
> > > We hope this explanation clarifies the mechanistic underpinnings of ASL beyond its empirical success.
> > >
> > > Best regards

---

### Official Review · Reviewer_82FN · 2025-11-04

**Soundness:** 3
**Presentation:** 3
**Contribution:** 3
**Rating:** 6
**Confidence:** 3

**Summary:**

This paper studies the out-of-distribution (OOD) detection problem. It proposes that recent work rely on complex training paradigms, which significantly limit the computational efficiency. Moreover, it discover that the existing methods neglect the influence of feature magnitude, which may lead to incorrect detection of OOD samples. To address this problem, it proposes a new method called ASL, which encourages robust feature learning by adding a L2-normalization before the classifier. The proposed method is evaluated on multiple datasets, where the performance achieves SOTA in most cases.

**Strengths:**

- The motivation of this paper is reasonable. It reveals the reason behind the failure of existing distance-based detection methods by empirical studies.
- The proposed method is simple while effective. This will improve the genealizability of the proposed method.
- The experimental results are strong. Also, the authors have conducted extensive ablation studies and additional analysis to make the proposed method more convincing.
- Theoretical analysis is provided to improve the solidness.

**Weaknesses:**

- The proposed method combines a feature normalization and a linear classifier. What about directly modifying the linear classifier into a cosine classifier or some other distance-based classifier? Using cosine classifier for OOD detection is already studied by previous works, so what is the difference between previous work and this study?
- The training efficiency is an important strength of this paper, which, however, is only mentioned in the section of experiments. Is it possible to add some explanations or analyses in the section of method to highlight this advantage?
- The font sizes of some figures are inharmonious (e.g. Figure 15). It is better to improve the fonts.

[1] Hyperparameter-free out-of-distribution detection using cosine similarity.

**Questions:**

See weaknesses.

---

> ### Author Response · Authors · 2025-11-17
>
> We sincerely appreciate the time and effort put into reviewing our paper and providing valuable feedback. We would like to address your questions below.
>
> > Q1: How does ASL differ from existing cosine classifier methods for OOD detection?
>
> Thank you for your valuable suggestion. We have experimentally compared the performance of ASL and the Cosine Classifier (Table A).
>
> ASL outperforms the cosine classifier on both CIFAR-10 and CIFAR-100. By dynamically constraining the classifier weight norm through weight decay $\lambda$, ASL forms an adaptive inverse temperature parameter, enabling it to better balance near-OOD and far-OOD performance compared to the cosine classifier.
>
> Table A: Comparison of ASL with Cosine Classifier and SphereFace for OOD Detection
> | Method           | CIFAR-10 |||| CIFAR-100 ||||
> |:------------------|:-----:|:-----|:-----:|:-----|:-----:|:-----|:-----:|:-----|
> |                  | Near || Far || Near || Far |
> |                  | FP↓  | AU↑  | FP↓   | AU↑  | FP↓   | AU↑  | FP↓   | AU↑  |
> | CE               | 40.2       | 88.8       | 30.6       | 91.0       | 67.8       | 74.8       | 51.9       | 82.6       |
> | SpereFace        | 46.1       | 88.3       | 27.9       | 92.5       | 73.4       | 72.4       | 68.6       | 69.7       |
> | Cosine Classifier| 26.6       | 93.0       | 15.6       | 96.3       | 57.7       | 80.0       | 54.4       | 81.1       |
> | ASL              | **25.1**       | **93.3**       | **12.0**       | **97.2**       | **56.2**       | **80.9**       | **47.8**       | **85.0**       |
>
> Furthermore, the prior work [1] adopts a cosine classifier approach. During the training phase, this method requires the design of an additional branch to predict scaling parameters for adjusting the cosine similarity output, and this branch is optimized simultaneously. Essentially, it modifies the output layer to adapt to cosine-based modeling, which involves extra parameters and design complexity. In contrast, ASL adopts a minimalist design that requires no additional branches or parameters. It only leverages feature $\ell_2$-normalization combined with weight decay to constrain the classifier norm, thereby achieving adaptive angular contrastive learning.
>
> > Q2: Can you emphasize the training efficiency advantage in the Method section?
>
> Thank you for your valuable comment regarding highlighting the training efficiency of ASL in the method section. As you recommended, we have supplemented explanations of training efficiency in the Method section (3.3 ASL: Training for Optimal Angular Separation), where we elaborate on the link between ASL’s core design and its high efficiency.
>
>
> > Q3:The font sizes of some figures are inharmonious (e.g. Figure 15). It is better to improve the fonts.
>
> We have retypeset figures with inconsistent font sizes, such as Fig. 15. While enhancing visual harmony, we have also maintained the clarity and readability of the chart information. The specific content of the relevant updated materials has been fully included in the revised version.
>
>
> [1] Engkarat Techapanurak, Masanori Suganuma, and Takayuki Okatani. Hyperparameter-free out-ofdistribution detection using cosine similarity. In Proceedings of the Asian conference on computer vision, 2020.

---

### Meta-Review · Area_Chair_B8PU · 2026-01-07

**Summary:**

This paper revisits out-of-distribution (OOD) detection and identifies a key weakness in distance-based detectors: their sensitivity to feature magnitude, which allows low-norm OOD samples to be misclassified as in-distribution. The authors propose Angular Separation Learning (ASL), a minimalist training modification that applies ℓ2-normalization to features before the classifier, forcing learning to focus on angular rather than radial separation. They show that this implicitly induces prototype-based contrastive learning in angular space without requiring synthetic outliers, contrastive losses, or additional model components. Extensive experiments across CIFAR, ImageNet variants, transformers, and near-/far-OOD settings demonstrate that ASL matches or outperforms many more complex state-of-the-art methods while remaining computationally efficient. Overall, the work argues that rethinking basic training geometry can yield strong OOD robustness without added complexity.

**Reviewer Concerns:**

irst, several reviewers argue that feature normalization and angular learning are not new ideas, citing prior work such as SphereFace, cosine classifiers, LogitNorm, and MD++, and question whether ASL constitutes a fundamentally new contribution or mainly a repackaging for OOD detection. Second, while the paper provides intuition and empirical evidence, reviewers repeatedly request a deeper and more rigorous theoretical or mechanistic explanation linking angular optimization to OOD generalization, beyond qualitative arguments and visualizations. Third, concerns were raised about overstated claims and incomplete analysis, including sensitivity to hyperparameters, far-OOD stability, confidence calibration, and clearer positioning relative to existing angular-margin and cosine-based approaches.

**Reviewer Scores:**

6,4,4,2

---

### Decision · Program_Chairs · 2026-01-26

Reject